# Succinate receptor 1 restricts hematopoiesis and prevents acute myeloid leukemia progression

Vincent Cuminetti [1,2,3], Emeline Boet[4], Marcel Heugel[1,2,17],
Joanna Konieczny [3,17], Aurora Bernal [3], Manuel J. Gomez[5], Franco Grimolizzi[3],
Nuria Vilaplana-Lopera [6], Marc Ferré[1,2,3], Alicia Villatoro[3], Deo P. Pandey[7],
Carlos Torroja [5], Hagar Taman[8], Ruth H. Paulssen [8,9], Thomas Vogl [10],
Caroline A. Heckman [11], Anders Vik[12,13], Giovanna Giovinazzo [14],
Nick van Gastel [15], Paloma García [6], Fátima Sánchez-Cabo [5],
Jean-Emmanuel Sarry [4] & Lorena Arranz [1,2,3,16] ✉

Despite intriguing roles for the Succinate receptor (Sucnr1) in inflammation, few studies have explored its role in hematopoiesis. Here, we show that low *SUCNR1* represents a marker for reduced overall and progression-free survival in acute myeloid leukemia (AML) patients. Succinic acid, which displays Sucnr1-dependent and independent effects, promotes disease in mouse models of pre-leukemic myelopoiesis, AML and AML xenografts, expressing low *SUCNR1*. In vivo global or hematopoietic deletion of Sucnr1 induces expansion of hematopoietic stem and progenitor cells (HSPC) and hematopoiesis, whilst Sucnr1-tomato⁺ HSPC display restricted engraftment potential. Mechanistically, activation of Sucnr1 counterbalances the stimulatory effect of intracellular succinate in HSPC and preserves HSPC transcriptional programs via control of S100a8/S100a9. Blocking S100a9 with tasquinimod rescues the defects of Sucnr1 knock-out mice, and combined with a potent Sucnr1 agonist shows therapeutic value in AML mice. In AML xenografts, single-cell RNA-sequencing reanalyses confirm *SUCNR1* as a therapeutic vulnerability in patients. Together, Sucnr1 signaling restricts hematopoiesis at least partially through HSPC and via control of S100a8/S100a9. Its dysregulation emerges as contributor to malignancy that opens therapeutic avenues for AML patients.

Hematopoietic stem cell (HSC) ability to self-renew and retain its identity depends on intrinsic features and on the microenvironment provided by non-HSC. Thus, both intrinsic and extrinsic factors support the great heterogeneity of the HSC compartment in terms of transcriptional and functional lineage priming, as demonstrated by the varied outcomes of transplanted individual HSC[1]. The HSC niche refers to cell-to-cell interactions, secreted factors, inflammation, and metabolic signals such as hypoxia, among others[2]. The hypoxic niche allows the fine-tuned metabolic control required to maintain HSC long-term quiescence and self-renewal, with participation of hypoxia-inducible factor-1 (HIF-1)[2,3]. HSC generate energy mainly via glycolysis[3], while leukemia stem cells (LSC) in AML contain greater mitochondrial mass, have higher oxygen consumption but display a low reactive oxygen species (ROS) content, and are sensitive to inhibition of mitochondrial translation[4], indicating that they are dependent on mitochondrial metabolism including the tricarboxylic acid (TCA) cycle and oxidative phosphorylation (OxPHOS). Reduced OxPHOS through BCL-2 inhibition selectively eradicated human quiescent LSC in xenografts[5].

Under physiological hypoxia, low oxygen causes the intermediate metabolite of the TCA cycle succinate to accumulate. Succinate enables ATP generation by OxPHOS[6] and it can be released from mitochondria into cytoplasm, nucleus and extracellular space, where it exerts pleiotropic roles[7–9]. In myeloid cells, cytoplasmic succinate results in stabilization of HIF-1α subunit[10,11], which then induces interleukin-1β (IL-1β) expression[11]. One of the first evidences for a role of succinate in cancer development was the discovery of pseudohypoxia; activation of hypoxia signaling under normal oxygen levels[12]. This can happen for example as a result of succinate dehydrogenase (Sdh) inhibiting mutations and leads to increased expression of genes that facilitate angiogenesis, metastasis, and glycolysis, ultimately leading to tumor progression mediated through HIF-1α[10]. Global inducible deletion of the mitochondrial *Sdhd* led to hematopoietic defects, including depletion of bone marrow (BM) progenitors and differentiated cells[13], using a mouse model that did not allow distinction of potential hematopoietic and stromal effects[8]. Succinate is a metabolic signal that links hypoxia to IL-1β and cell responses in myeloid cells through intracellular mechanisms, but also autocrine/paracrine processes through Sucnr1[8,11,14–16]. Sucnr1 seems to have different functions depending on the context, and has been proposed as inflammatory in dendritic cells but anti-inflammatory in neural stem cells and both inflammatory or anti-inflammatory in macrophages[14–17]. Cancer cells release succinate and locally polarize macrophages into tumor-associated macrophages through Sucnr1 activation[18]. Despite these intriguing precedents, succinate has been scarcely explored in hematopoiesis. Human CD34+ progenitors express SUCNR1 and its activation has been suggested to induce proliferation in vitro[19].

Here, we investigate the in vivo role and the cellular and molecular mechanisms of succinate and Sucnr1 signaling in normal and malignant hematopoiesis. Low *SUCNR1* represents a marker for poor prognosis in AML patients. In mouse models of pre-leukemic myelopoiesis, AML and AML xenografts that express low *SUCNR1*, treatment with succinic acid results in disease progression. To study the effect of low Sucnr1 in the hematopoietic system, we used global or hematopoietic deletion of Sucnr1, which induce expansion of HSPC and hematopoiesis. Conversely, we generated a *Sucnr1-tdTomato* mouse line which informed that Sucnr1-tomato+ HSPC form a subset of HSPC with restricted engraftment potential. Importantly, activation of Sucnr1 counterbalances the stimulatory effect of succinate in HSPC and preserves transcriptional programs characteristic of HSPC via control of S100a8/S100a9 under steady-state. We further show therapeutic value for a drug combination blocking S100a9 and activating Sucnr1 in AML mice, and confirm *SUCNR1* as a potential therapeutic vulnerability in human AML. Thus, our data support dysregulation of Sucnr1 as a contributor to AML pathogenesis that show promise for future therapeutic exploitation.

## Results

### Low *SUCNR1* predicts acute myeloid leukemia progression

We first studied the prognostic value of *SUCNR1* in publicly available arrays from purified AML blasts in a published cohort of AML patients (GSE14468[20–22]). Low levels of expression of *SUCNR1* were associated with reduced overall survival (Fig. 1A) and progression-free survival (Fig. 1B). We performed subsequent Cox regression analysis, which uncovered *SUCNR1* expression as an independent predictor of overall survival and progression-free survival after adjusting for age, gender and French-American-Bristish (FAB) classification (Supplementary Table S1–S2). We separately analyzed less differentiated AML profiles according to FAB and found that low *SUCNR1* is a marker of poor prognosis in M0–M3 AML patients ($n = 224$) (Supplementary Fig. 1A). In parallel, we identified M4–M5 patients according to FAB as those patients with lowest *SUCNR1* expression, versus M0–M3 (Fig. 1C). In the same cohort, the expression of *SUCNR1* was heterogeneous among AML patients with various molecular signatures. CEBPA, FLT3-ITD and

NPM1 mutations were associated with higher *SUCNR1* expression in AML blasts, IDH1, IDH2 and FLT3-TKD had no impact on *SUCNR1* expression, and NRAS mutations were linked to a trend towards reduced levels of *SUCNR1* expression (Supplementary Fig. 1B). We then used the dataset TARGET AML at BloodSpot (GSE42519[23] and GSE13159[24,25]) that confirmed great heterogeneity in *SUCNR1* expression in primary AML and uncovered overall lower expression values compared to controls (Fig. 1D). To understand how low expression of *SUCNR1* is functionally linked to poor survival in AML patients in vivo, we used succinic acid injections in AML patient xenografts, as succinic acid takes the form of the endogenous anion succinate in living organisms. Using CD34+ progenitors obtained from one M1 AML patient with adverse risk and no detectable expression of *SUCNR1* by digital droplet (dd)PCR (Supplementary Fig. 1C), we found that in vivo treatment with succinic acid increased the leukemic output, as measured by numbers of human CD33+ myeloid progenitors, in the BM of xenograft recipients versus vehicle-treated mouse recipients (Fig. 1E and Supplementary 1D).

We then used the *Mx1-Cre NRAS^G12D* mouse model of aberrant pre-leukemic myelopoiesis[26,27], from now on referred to as NRAS-G12D+. Primary NRAS-G12D+ mice exhibit a mild hematopoietic phenotype, characterized as chronic myelomonocytic leukemia or myeloproliferative neoplasm (MPN)[28,29]. BM nucleated cells from NRAS-G12D+ expressed less *Sucnr1* than their control counterparts (Fig. 2A). Early stages of NRAS-G12D+-driven disease are characterized by a selective expansion of BM HSPC in the absence of other remarkable hematopoietic events, including unchanged monocytes and BM cellularity (Supplementary Fig. 2A). Intracellular succinate was similar in NRAS-G12D+ Lin−Sca-1+c-Kit+ (LSK) and monocytes versus control cells, but LSK cells showed 3-fold higher succinate content than monocytes (Fig. 2B). At early stages of disease, we found higher content of succinate in the BM extracellular fluid (BMEF) of pre-leukemic NRAS-G12D+ mice compared to control mice (Fig. 2C). In turn, succinic acid injection in vivo increased succinate content in BMEF (Supplementary Fig. 2B) and aggravated disease in NRAS-G12D+ mice, promoting myeloid bias and B cell development deficiency (Fig. 2D and Supplementary Fig. 2C) together with expansion of the LSK cell compartment in the BM analyzed by fluorescence-activated cell sorting (FACS) (Fig. 2E and Supplementary Fig. 2D). Analysis of the stem and progenitor cell subsets corresponding to HSC and MPP1-MPP6[30–32] revealed a selective expansion of MPP3 and MPP4 HSPC (Fig. 2F and Supplementary Fig. 2D). Within lineage-negative progenitors, we found altered fractions of Lin−c-Kit+Sca-1− (LK) cells which were increased and common lymphoid progenitors (CLP) that were reduced after succinic acid treatment (Supplementary Fig. 2E and F). LK subsets were slightly biased towards common myeloid progenitors (CMP) whereas megakaryocyte erythroid progenitors (MEP) and granulocyte monocyte progenitors (GMP) were unchanged after succinic acid treatment (Supplementary Fig. 2E and G).

We then used an aggressive leukemic mouse model heterozygous for *Mll-AF9* knock-in (KI) fusion transgene or wild-type (WT) control male littermates[33]. Expression of the MLL-AF9 fusion protein results in development of leukemia beginning around six months of age, mainly AML. BM nucleated cells from mice transplanted with BM of MLL-AF9+ leukemic donors expressed less *Sucnr1* than their healthy counterparts (Fig. 2G). Succinic acid administration reduced survival in mice transplanted with BM of MLL-AF9+ leukemic donors versus vehicle-treated mice (Fig. 2H). MLL-AF9+ mice had overt AML and the lineage-negative compartment was expanded in the BM of succinic acid- versus vehicle-treated mice (Fig. 2I), with an observed trend toward increased LK fraction and LSK cells that did not reach statistical significance (Supplementary Fig. 2H). The leukemic burden by means of increased counts of abnormal monocytes in extramedullary sites was higher in leukemic mice treated with succinic acid (Fig. 2J).

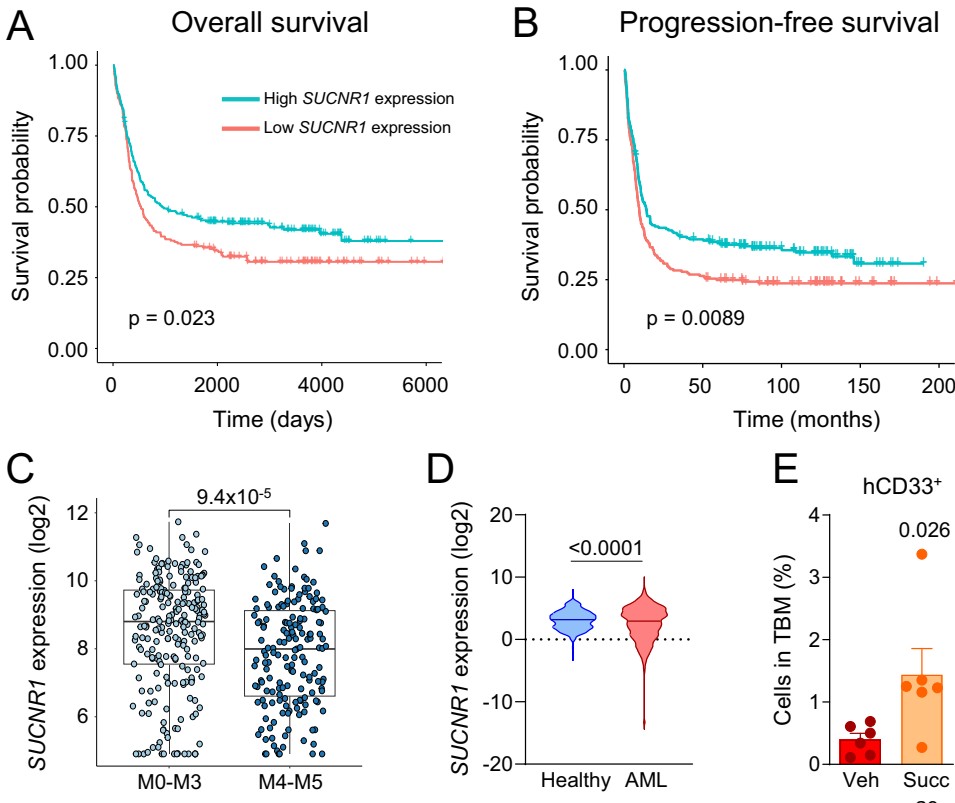

**Fig. 1 | Low *SUCNR1* predicts acute myeloid leukemia progression.**
**A–B** Kaplan–Meier analysis of acute myeloid leukemia (AML) patient survival data analyzed as a function of *SUCNR1* expression in AML blasts (GSE14468, *n* = 426). *SUCNR1* expression levels were discretized as high or low based on the median *SUCNR1* expression level for the whole cohort. *p* value indicates likelihood ratio test. **A** Analysis of overall survival. **B** Analysis of progression-free survival. **C** *SUCNR1* expression in patients categorized as M0-M3 (*n* = 224) and M4-M5 (*n* = 171) in GSE14468. **D** *SUCNR1* expression in total bone marrow (TBM) of healthy donors or AML patients, data were extracted from Bloodspot repository using TARGET AML database, normal cells are from GSE42519 and AML cells are from GSE13159.

**E** Treatment with succinic acid or vehicle of NSG mice for 10 days, starting 20 weeks after the transplant with CD34⁺ progenitors isolated from the BM of one M1 AML patient. TBM fraction of human CD33⁺ (hCD33⁺) blasts analyzed by fluorescence-activated cell sorting (*n* = 6 per group). Statistical analyses were performed with log-rank Mantel-Cox test (**A**, **B**), two-tailed Student's *t* test (**C**, **D**) or two-tailed Mann–Whitney *U* test (**E**). Significant *p*-values are reported. Data are biologically independent human samples or animals, and means ± SEM for bar plots or medians for violin plots. Box plots in (**C**) show median as center line; upper and lower quartiles as box limits; 1.5x interquartile range as whiskers. Source data are provided as a Source Data file.

To sum up, AML patients show heterogeneous expression of *SUCNR1* and low *SUCNR1* represents a prognostic marker for reduced overall and progression-free survival. Conversely, in AML mouse models and patient xenografts with low *SUCNR1* expression, succinate accumulates in BM and contributes to leukemia progression.

## Sucnr1 restricts HSPC and hematopoiesis under steady-state

To understand the uncovered protective effect of succinate signaling via Sucnr1 on hematopoiesis, we next characterized the immunophenotype of Sucnr1 knock-out (KO) mice, which have not been used before in studies of hematopoiesis. Young adult Sucnr1-KO mice had increased circulating red blood cells, hemoglobin and hematocrit, but no other remarkable events were observed compared to WT mice in peripheral blood (PB), BM or spleen (SP) (Supplementary Fig. 3A–D). Sucnr1-KO mice were analyzed at later middle-aged stages and then showed an increase in circulating leukocytes, particularly monocytes and B cells (Fig. 3A), in addition to expanded reticulocytes (Supplementary Fig. 3E). In the BM, Sucnr1 deletion induced hypercellularity with expansion of myeloid cells, B cells and HSPC identified as LSK cells (Fig. 3B–D). The fractions of LSK cells were altered in Sucnr1-KO versus WT mice with expansion of long-term (LT)-HSC, short-term (ST)-HSC and multipotent progenitors (MPP), analyzed by FACS (Supplementary Fig. 4A). Further analysis of the stem and progenitor cell subsets corresponding to HSC and MPP1-MPP6[30–32] revealed absolute expansion

of HSC, MPP1, MPP4, MPP5 and MPP6, and a trend towards expansion of MPP3 (Fig. 3E). We also analyzed the population percentages of LSK cells, which was increased in the BM of Sucnr1-KO versus WT mice (Supplementary Fig. 4B). Analysis of HSPC subsets corresponding to HSC and MPP1-MPP6 uncovered increased relative frequencies of HSC, MPP1, MPP4, MPP5 and MPP6 (Supplementary Fig. 4B), supporting for an HSPC origin of the stimulated hematopoiesis in the BM of Sucnr1-KO mice. Numbers of colonies formed ex vivo by HSPC were higher in Sucnr1-KO mice (Fig. 3F). We performed serial replatings and found that HSPC from Sucnr1-KO mice generated higher numbers of secondary colonies in a non-significant trend, and similar numbers of tertiary colonies compared to WT (Fig. 3F). Numbers of LK progenitors, particularly MEP and GMP, as well as CLP were increased in the BM of Sucnr1-KO versus WT mice (Fig. 3G and H).

Consistent with the increased numbers of HSPC, analysis of the LSK cell compartment in the BM of Sucnr1-KO mice showed reduced numbers of early apoptotic BM LSK and a trend towards increased numbers of live cells (Supplementary Fig. 4C). Conversely, we analyzed LSK cell cycle and the proliferative fraction in the G1 phase was slightly reduced (Supplementary Fig. 4D).

Succinate levels were reduced in the BMEF of Sucnr1-KO versus WT mice (Supplementary Fig. 4E), whilst intracellular succinate levels in HSC were increased (Supplementary Fig. 4F). Given that succinate is an intermediate metabolite of the TCA cycle, we studied its potentially

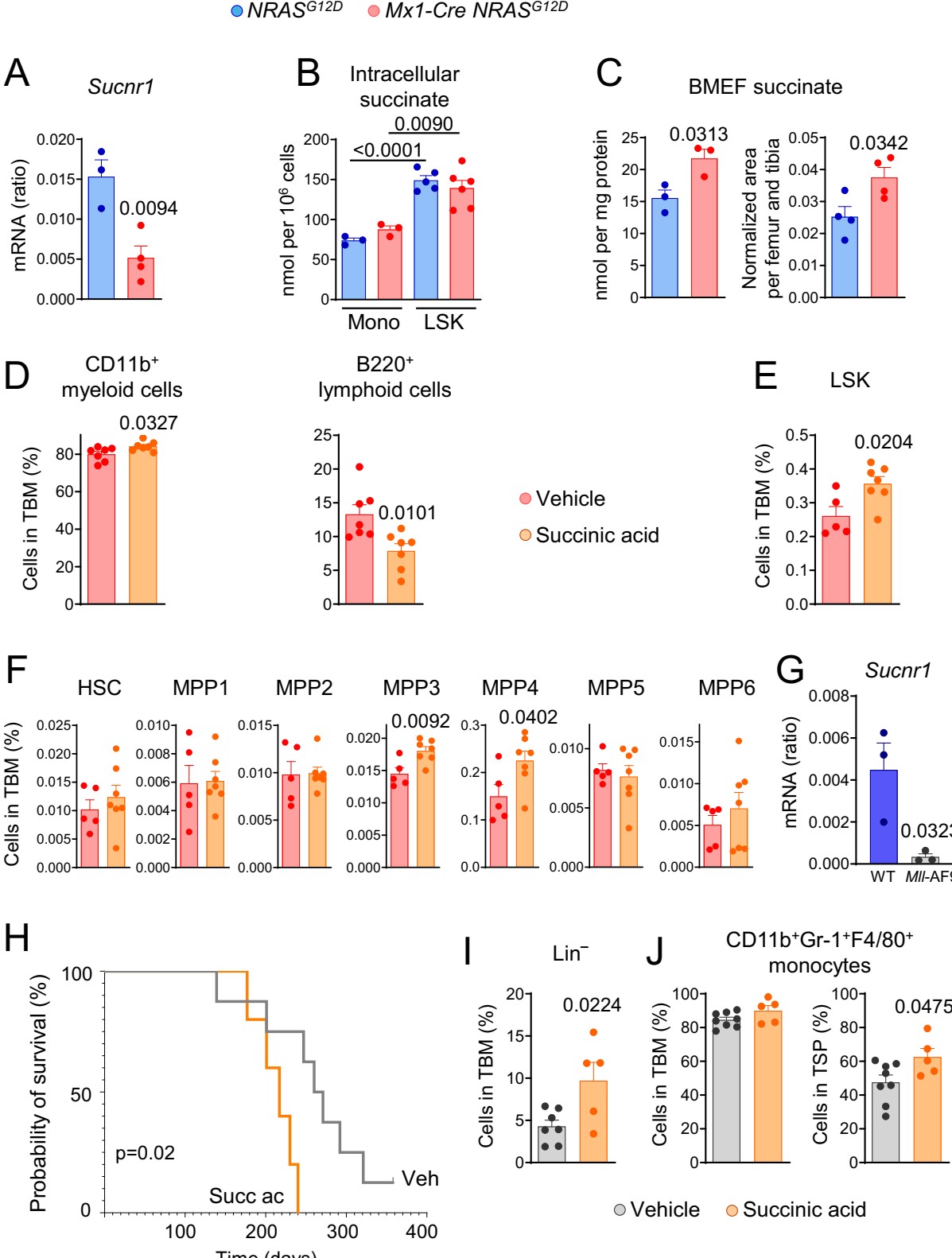

increased cycling which would result in metabolic changes. Sucnr1 deletion resulted in increased baseline oxygen consumption rate in lineage-negative progenitors (Supplementary Fig. 4G). This effect was specific and reversible because the maximal oxygen consumption rate and the glycolytic flux remained unchanged (Supplementary Fig. 4G). Consistent with these data, both total and mitochondrial ROS levels were higher in LSK cells from Sucnr1-KO mice (Supplementary Fig. 4H).

In view of the stimulated hematopoiesis in the BM of Sucnr1-KO mice, we also analyzed the SP as an extramedullary site of hematopoiesis. The total numbers of nucleated cells in SP were increased in Sucnr1-KO mice, due to higher numbers of myeloid cells, including both granulocytes and monocytes, and B cells (Supplementary Fig. 4I). Numbers of secondary colonies generated ex vivo by HSPC were higher in Sucnr1-KO mice (Supplementary Fig. 4J). Conversely, in vivo

**Fig. 2 | Succinate promotes acute myeloid leukemia progression in mouse models with low *SUCNR1* expression. A** Droplet digital (dd)PCR expression of *Sucnr1* relative to *Gapdh* in bone marrow nucleated cells (BMNC) from control (*Mx1-Cre⁻ NRAS^G12D*, *n* = 3) and NRAS-G12D⁺ (*Mx1-Cre⁺ NRAS^G12D*, *n* = 4) mice, 11 weeks after poly-inosine:poly-cytosine (pIpC). **B** Intracellular succinate levels in total BM (TBM) Lin⁻Sca-1⁺c-Kit⁺ (LSK) cells and CD11b⁺Gr-1⁺F4/80⁺ monocytes obtained by fluorescence-activated cell sorting (FACS)-sorting from control (*n* = 3 Mono, *n* = 5 LSK) and NRAS-G12D⁺ (*n* = 3 Mono, *n* = 6 LSK) mice, 5 weeks after pIpC. **C** Succinate levels in BM extracellular fluid (BMEF) from control and NRAS-G12D⁺ mice measured by colorimetric assay 5 weeks after pIpC induction (*n* = 3 per group, left) and by nuclear magnetic resonance 6 weeks after pIpC induction (*n* = 4 per group, right). **D–F** Treatment with succinic acid (Succ. ac.) or vehicle (Veh) of NRAS-G12D⁺ mice for 27 weeks, starting 7 weeks after pIpC induction. TBM fractions analyzed by FACS of **D** CD11b⁺ myeloid cells and B220⁺ lymphocytes (*n* = 7 per group), (**E**) LSK cells (*n* = 5 Veh, *n* = 7 Succ. ac.), **F** LSK cell subsets: LSK CD34⁻Flt3⁻CD150⁺CD48⁻, hematopoietic stem cells (HSC); LSK CD34⁺Flt3⁻CD150⁺CD48⁻, multipotent

progenitors 1 (MPP1); LSK CD34⁺Flt3⁻CD150⁺CD48⁺ (MPP2); LSK CD34⁺Flt3⁻CD150⁻CD48⁺ (MPP3); LSK CD34⁺Flt3⁺CD150⁻CD48⁺ (MPP4); LSK CD34⁺Flt3⁻CD150⁻CD48⁻ (MPP5); LSK CD34⁻Flt3⁻CD150⁻CD48⁻ (MPP6) (*n* = 5 Veh, *n* = 7 Succ. ac.). **G** ddPCR expression of *Sucnr1* relative to *Gapdh* in BMNC from wild-type (WT) and *Mll*-AF9 mice (*n* = 3 per group). **H–J** Treatment with succinic acid or vehicle of C57BL6/J mice transplanted with BMNC from leukemic *Mll*-AF9 mice for a maximum of 42 weeks, starting 4 weeks after transplant. **H** Probability of survival of recipient mice treated with succinic acid (*n* = 5) or vehicle (*n* = 8) with leukemia-confirmed deaths, calculated by Kaplan–Meier method. **I** TBM fractions of Lin⁻ cells analyzed by FACS at endpoint (*n* = 7 Veh, *n* = 5 Succ. ac.). **J** TBM and total spleen (TSP) cell fractions of monocytes analyzed by FACS (*n* = 8 Veh, *n* = 5 Succ. ac.). Statistical analyses were performed with two-tailed Student's *t*-test (**A–G, I, J**) or log-rank Mantel-Cox test (**H**). Significant *p*-values are reported. Data are biologically independent animals, and means ± SEM for bar plots. Source data and non-significant *p*-values are provided as a Source Data file.

treatment of mice with one single injection of succinic acid did not affect mobilization of progenitors from BM (Supplementary Fig. 5A).

To dissect the role of Sucnr1 on healthy hematopoiesis via the hematopoietic compartment, we transplanted *Sucnr1^fl/fl* control or *Mx1-Cre Sucnr1^fl/fl* experimental BM cells into C57BL/6 J WT mice, which were later injected with poly-inosine:poly-cytosine (pIpC; Fig. 4A) to induce Sucnr1 deletion from hematopoietic cells (Supplementary Fig. 6A). The end-point analysis was performed 47 weeks (about 11 months) after pIpC induction, to make it a time-point comparable to the global deletion of Sucnr1. In vivo deletion of Sucnr1 from hematopoietic cells resulted in early expansion of the numbers of circulating total leukocytes, i.e., myeloid cells, monocytes and B lymphocytes (Fig. 4B), which was maintained over the course of the experiment (Fig. 4C). In BM, in vivo deletion of Sucnr1 from hematopoietic cells led to an expansion of MPP versus intact recipients, particularly MPP4 (Fig. 4D). Thus, Sucnr1 effects on hematopoiesis can be explained greatly by intrinsic effects to hematopoietic cells.

Together, these results indicate that Sucnr1 restricts HSPC and hematopoiesis in vivo, mainly cell-autonomously.

## Sucnr1 is expressed in HSPC and its activation restricts their functional response

In view of the lack of commercially available mouse Sucnr1-specific antibody, we generated a *Sucnr1-tdTomato* reporter mouse line (Supplementary Fig. 7A–D) and used it to study the expression of Sucnr1 protein by BM HSPC subsets. FACS-sorted LSK cells from *Sucnr1-tdTomato* and WT mice were imaged by imaging flow cytometry, which revealed presence of SUCNR1 in a subset of LSK cells (Fig. 5A). HSPC subsets, committed hematopoietic progenitors and differentiated cells obtained from the mouse BM were further analyzed by spectral flow cytometry (Supplementary Fig. 8A). After spectral unmixing, we quantified the Sucnr1-tdTomato⁺ cell fractions (Fig. 5B and Supplementary Fig. 9A–E). Our analysis revealed that all HSPC subsets show a varying fraction of Sucnr1⁺ cells and the majority of Sucnr1⁺ HSPC were identified as MPP3, with an average of 0.191% of Sucnr1⁺ cells, followed by MPP4 (Fig. 5B and Supplementary Fig. 9C). Importantly, although Sucnr1⁺ cells were found across all hematopoietic cell subsets analyzed, including all lineage-positive cell types (Supplementary Fig. 9C and D), the majority of Sucnr1⁺ cells in the hematopoietic system were identified as lineage-negative progenitors, including all LK progenitors particularly GMP, LSK cells and CLP (Supplementary Fig. 9E). In parallel, we confirmed *SUCNR1* mRNA expression across all hematopoietic cell subsets analyzed in normal human hematopoiesis, using expression data from the HemaExplorer collection at BloodSpot (GSE17054[34], GSE19599[35], GSE11864[36] and E-MEXP-1242[37]), including HSC and HSPC (Supplementary Fig. 9F). In humans, CMP, GMP and differentiating promyelocytes and myelocytes display remarkably high levels of *SUCNR1* mRNA expression (Supplementary Fig. 9F).

We then tested mouse HSPC functional response to cell-permeable succinate ex vivo by adding monomethyl succinate, inactive on Sucnr1[38], which enhanced the colony-forming potential in total BM (TBM) in a dose-response manner (Fig. 5C). Conversely, succinic acid, which displays both Sucnr1-dependent and independent effects[38,39], resulted in unchanged colony-forming potential (Fig. 5C). Taken together, these results suggest that Sucnr1 activation represses the intracellular effect of succinate. We further explored the effect of succinic acid and monomethyl succinate combination, which resulted in a partial increase in colony-forming potential in TBM versus succinic acid alone and a partial reduction versus monomethyl succinate alone, confirming that Sucnr1 activation represses the intracellular effect of succinate (Fig. 5D).

To dissect a potentially direct effect of Sucnr1 activation on HSC function, FACS-sorted HSC immunophenotypically defined as Lin⁻c-Kit⁺Sca-1⁺CD34⁻Flt3⁻CD150⁺CD48⁻ were tested for their colony-forming potential in serial replatings in the presence of monomethyl succinate or cis-epoxy succinate, 10- to 20-fold more potent than succinic acid on Sucnr1[39]. In the second replating, monomethyl succinate showed a trend towards stimulation but cis-epoxy succinate reduced the colony-forming potential of HSC, with no effect on long-term colony forming potential measured after the third plating of the cells (Fig. 5E).

To confirm a direct role of Sucnr1 on HSPC function, FACS-sorted HSC from Sucnr1-KO mice were tested for their colony-forming potential in serial replatings in the presence of monomethyl succinate or cis-epoxy succinate. In the first and second replating, both monomethyl succinate and cis-epoxy succinate stimulated or showed a trend towards stimulation of the colony-forming potential of Sucnr1-KO HSC, suggesting loss of colony-forming potential repression by Sucnr1 activation in the absence of Sucnr1 (Fig. 5F). No changes were observed at third plating, indicative of no impact on long-term colony forming potential.

We then performed competitive repopulation assays of BM nucleated cells isolated from WT or Sucnr1-KO mice (Ly5.2) into lethally irradiated Ly5.1 congenic recipients. Surprisingly, peripheral blood analysis from recipient mice 16 weeks after transplantation revealed a mild reduction in HSPC engraftment in competition, in recipients transplanted with Sucnr1-KO cells compared to recipients transplanted with WT control cells (Fig. 5G and Supplementary Fig. 9G). To substantiate the role of Sucnr1 in HSPC function in vivo, we performed competitive repopulation assays using 200 Sucnr1-tdTomato⁺ or Sucnr1-tdTomato⁻ FACS-sorted HSPC (containing a mix of 40 HSC, 80 MPP3 and 80 MPP4) isolated from Ly5.2 *Sucnr1-tdTomato* mice into lethally irradiated Ly5.1 congenic recipients. Peripheral blood analysis from recipient mice 10 weeks after the transplant showed a reduction of circulating leukocytes and donor-derived monocytes in recipients of Sucnr1⁺ cells compared to recipients of Sucnr1⁻ cells (Fig. 5H and Supplementary Fig. 9H).

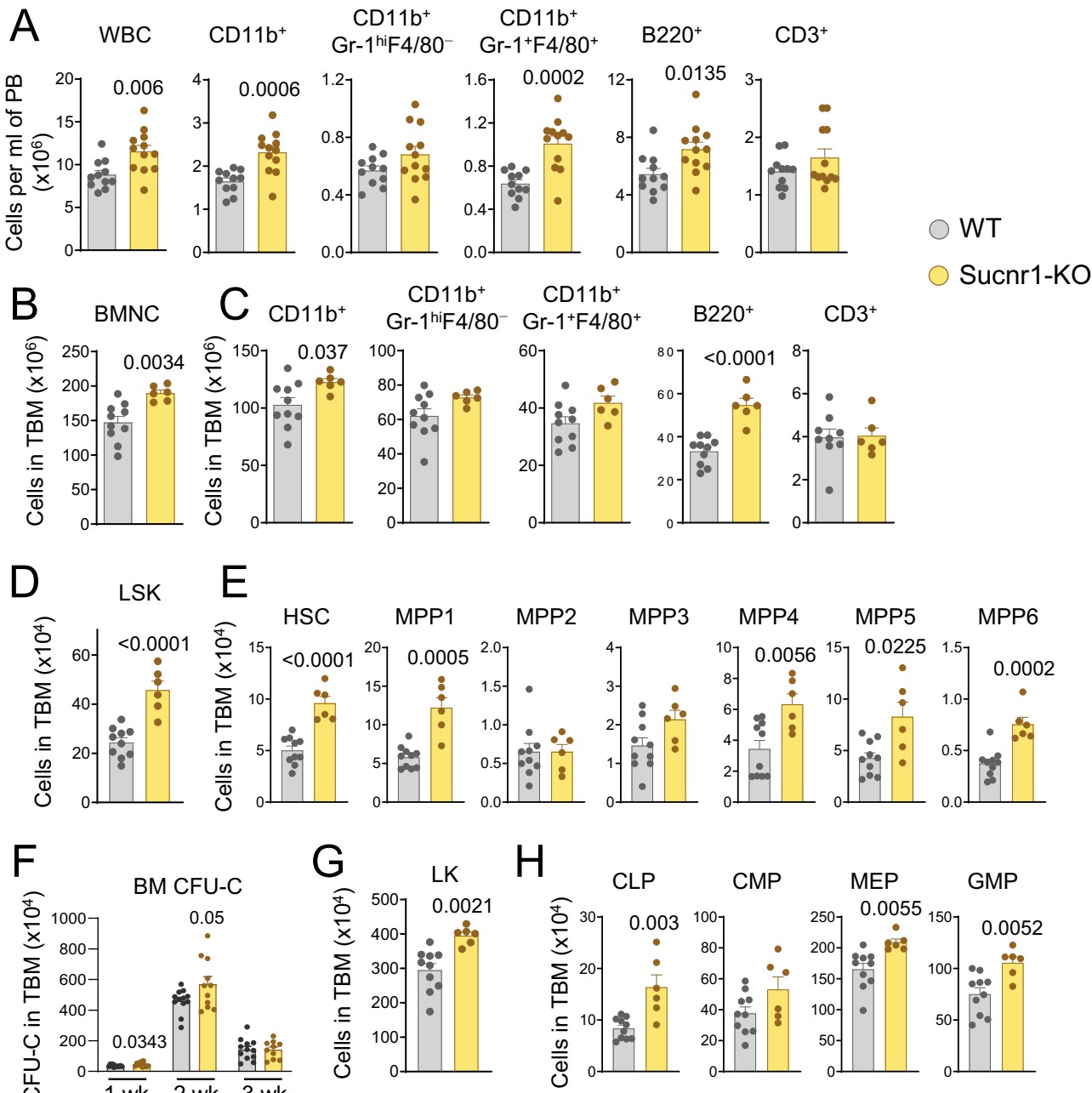

**Fig. 3 | Sucnr1 restricts HSPC and hematopoiesis under steady-state.** Analysis of hematopoiesis in C57BL6/J wild-type (WT) and Sucnr1 knock-out (Sucnr1-KO) mice (n = 10 WT, n = 6 Sucnr1-KO, 53-54 weeks old, unless indicated otherwise). **A** Numbers of white blood cells (WBC) measured with hematological analyzer and CD11b⁺ myeloid cells, CD11b⁺Gr-1hiF4/80⁻ granulocytes, CD11b⁺Gr-1⁺F4/80⁺ monocytes, B220⁺ B lymphocytes and CD3⁺ T lymphocytes per ml of peripheral blood (PB) analyzed by fluorescence-activated cell sorting (FACS) (n = 11 WT, n = 12 Sucnr1-KO, 38-41 weeks old). **B** Total bone marrow (TBM) nucleated cells measured with cell counter. **C** CD11b⁺ myeloid cells, CD11b⁺Gr-1hiF4/80⁻ granulocytes, CD11b⁺Gr-1⁺F4/80⁺ monocytes, B220⁺ B lymphocytes and CD3⁺ T lymphocytes analyzed by FACS. **D** TBM number of Lin⁻Sca-1⁺c-Kit⁺ (LSK) cells analyzed by FACS. **E** TBM numbers of LSK cell subsets: LSK CD34⁻Flt3⁻CD150⁺CD48⁻, hematopoietic stem cells (HSC); LSK CD34⁺Flt3⁻CD150⁺CD48⁻, multipotent progenitors 1 (MPP1); LSK

CD34⁺Flt3⁻CD150⁺CD48⁺ (MPP2); LSK CD34⁺Flt3⁻CD150⁻CD48⁺ (MPP3); LSK CD34⁺Flt3⁺CD150⁻CD48⁺ (MPP4); LSK CD34⁺Flt3⁺CD150⁻CD48⁻ (MPP5); LSK CD34⁻Flt3⁻CD150⁻CD48⁻ (MPP6) analyzed by FACS. **F** TBM numbers of colony-forming unit cells (CFU-C) ex vivo after serial replating (n = 12 WT, n = 11 Sucnr1-KO 1–2 wk, n = 10 Sucnr1-KO 3 wk; wk, weeks). **G** TBM number of Lin⁻c-Kit⁺Sca-1⁻ (LK) progenitors analyzed by FACS. **H** TBM numbers of common lymphoid progenitors (CLP, Lin⁻c-KitlowSca-1lowCD127⁺), common myeloid progenitors (CMP, LK CD34⁺FcRγ⁻), megakaryocyte erythroid progenitors (MEP, LK CD34⁻FcRγ⁻) and granulocyte-monocyte progenitors (GMP, LK CD34⁺FcRγ⁺) analyzed by FACS. Statistical analyses were performed with two-tailed Student's *t* test (**A–D, E** except MPP1 and MPP5, **F, G, H** except CLP) or two-tailed Mann−Whitney *U* test (**E** MPP1 and MPP5, **H** CLP). Significant *p*-values are reported. Data are biologically independent animals, and means ± SEM. Source data and non-significant *p*-values are provided as a Source Data file.

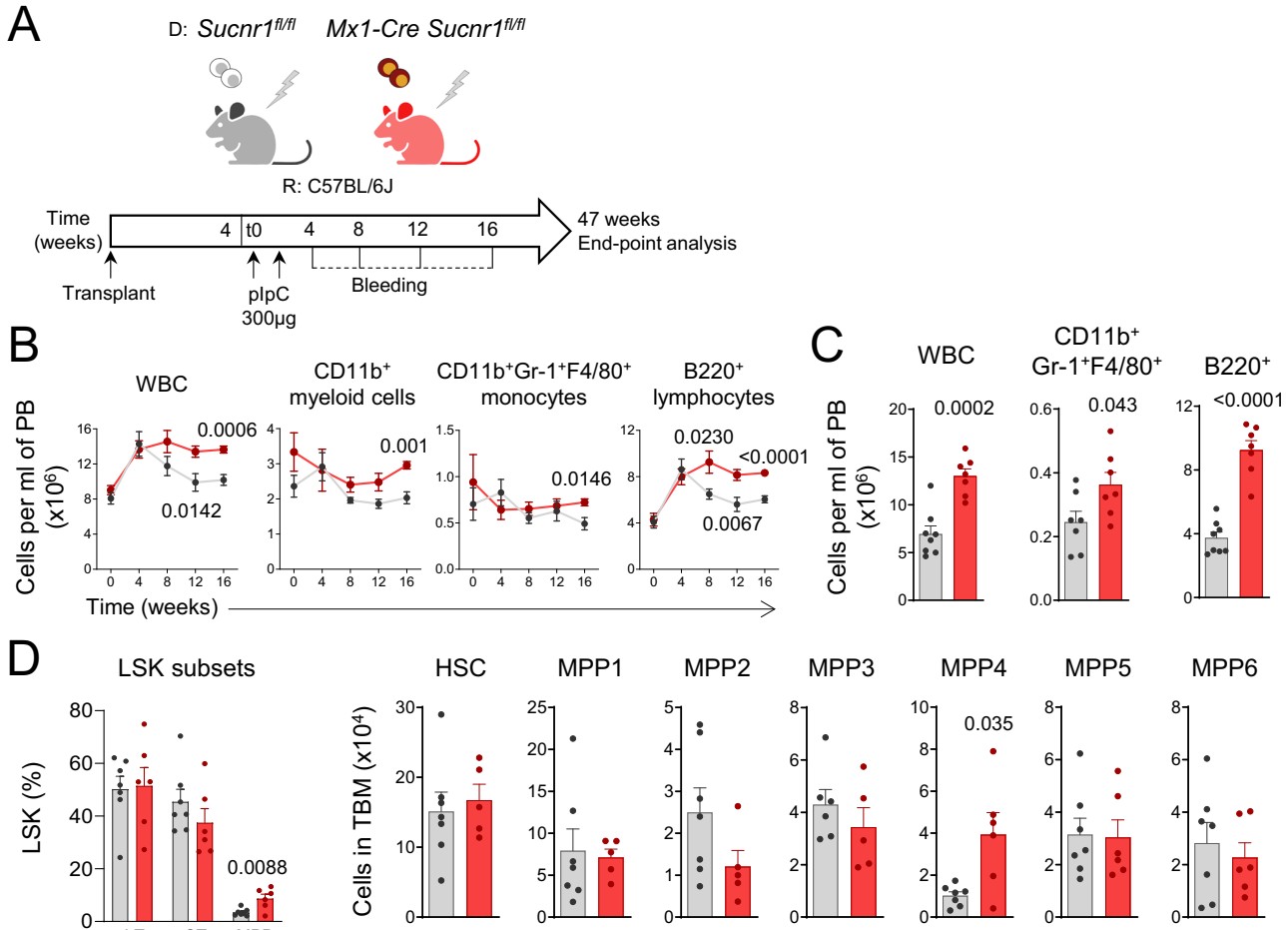

**Fig. 4 | Sucnr1 restricts HSPC and hematopoiesis cell-autonomously.** Transplant of C57BL6/J wild-type (WT) mice with bone marrow nucleated cells (BMNC) from *Sucnr1^fl/fl* or *Mx1-Cre Sucnr1^fl/fl* and induction 4 weeks after with poly-inosine:poly-cytosine (pIpC). **A** Illustration of the experimental design. R: recipient. D: donor cells. The mouse, cell and beam icons: Created in BioRender. Cuminetti, V. (2026) https://BioRender.com/9bm56nx. **B** Evolution in number of white blood cells (WBC) per ml of peripheral blood (PB) measured with hematological analyzer (*n* = 8 *Sucnr1^fl/fl*, *n* = 8 weeks 0–4 and *n* = 7 weeks 8–16 *Mx1-Cre Sucnr1^fl/fl*), and in numbers of CD11b^+ myeloid cells (*n* = 8 weeks 0–4, *n* = 7 weeks 8, 16 and *n* = 6 week 12 *Sucnr1^fl/fl*; *n* = 8 weeks 0–4, *n* = 7 week 8 and *n* = 6 weeks 12–16 *Mx1-Cre Sucnr1^fl/fl*), CD11b^+Gr-1^+F4/80^+ monocytes (*n* = 8 weeks 0–12 and *n* = 7 week 16 *Sucnr1^fl/fl*, *n* = 8 weeks 0–4, *n* = 7 week 8 and *n* = 6 weeks 12–16 *Mx1-Cre Sucnr1^fl/fl*) and B220^+ B lymphocytes (*n* = 8 weeks 0–4, 12–16 and *n* = 7 week 8 *Sucnr1^fl/fl*, *n* = 8 weeks 0–4, *n* = 7 weeks 8, 16 and *n* = 6 week 12 *Mx1-Cre Sucnr1^fl/fl*) per ml of PB analyzed by fluorescence-activated cell sorting (FACS). **C** Number of WBC (*n* = 8 *Sucnr1^fl/fl*, *n* = 7 *Mx1-Cre Sucnr1^fl/fl*), and numbers of CD11b^+Gr-1^+F4/80^+ monocytes (*n* = 7 per group) and B220^+ B lymphocytes (*n* = 8 *Sucnr1^fl/fl*, *n* = 7 *Mx1-Cre Sucnr1^fl/fl*) per ml of PB,

47 weeks after pIpC. **D** (Left) Fractions of Lin^–Sca-1^+c-Kit^+ (LSK) subsets; long-term hematopoietic stem cells (LT, LSK CD34^–Flt3^–), short-term hematopoietic stem cells (ST, LSK CD34^+Flt3^–) and multipotent progenitors (MPP, LSK CD34^+Flt3^+) within the LSK cell compartment analyzed by FACS (*n* = 7 *Sucnr1^fl/fl*, *n* = 6 *Mx1-Cre Sucnr1^fl/fl*). (Right) Total BM (TBM) numbers of LSK cell subsets: LSK CD34^–Flt3^–CD150^+CD48^–, hematopoietic stem cells (HSC); LSK CD34^+Flt3^–CD150^+CD48^–, multipotent progenitors 1 (MPP1); LSK CD34^+Flt3^–CD150^+CD48^+ (MPP2); LSK CD34^+Flt3^–CD150^–CD48^+ (MPP3); LSK CD34^+Flt3^+CD150^+CD48^+ (MPP4); LSK CD34^+Flt3^+CD150^–CD48^+ (MPP5); LSK CD34^–Flt3^–CD150^–CD48^– (MPP6) analyzed by FACS (*n* = 7 all except *n* = 6 MPP3 *Sucnr1^fl/fl*, *n* = 5 HSC, MPP1-3 and *n* = 6 MPP4-6 *Mx1-Cre Sucnr1^fl/fl*). Statistical analyses were performed with two-tailed Student's *t*-test (**B**, **C**, **D** except MPP4) or two-tailed Mann–Whitney *U* test (**D** MPP4). Significant *p*-values are reported. Data are biologically independent animals, and means ± SEM for bar plots. Data are mean ± SEM in (**B**) for better visualization. Source data and non-significant *p*-values are provided as a Source Data file.

Thus, Sucnr1 is expressed in HSPC mainly in MPP3 and its activation restricts their functional response to succinate ex vivo. Notably, although BM cells from Sucnr1-KO mice are less fit in competition with WT cells, HSPC transplantation experiments uncovered that Sucnr1 expression traces a subset of HSPC with restricted engraftment potential in vivo versus Sucnr1^– HSPC.

**Sucnr1 preserves transcriptional programs characteristic of HSPC**

To identify specific changes in HSPC driven by Sucnr1 deletion, we next performed gene expression profiling by bulk RNA sequencing (RNA-Seq) of WT and Sucnr1-KO LSK cells. We identified a significant impact of Sucnr1 genetic abrogation on the transcriptional program of the LSK

cell compartment, with 4585 differentially expressed genes (adjusted *p* < 0.05) (Fig. 6A). Gene Set Enrichment Analysis (GSEA) according to two different databases, Chemical and Genetic Perturbations (CGP) and Gene Ontology Biological Processes (GOBP) revealed coordinated changes in a variety of genes associated with reduced stemness and primitive HSC, and instead enrichment in functional categories related to myeloid activation, inflammatory response and ROS as well as signatures characteristic of late progenitors and mature hematopoietic cells, in Sucnr1-KO HSPC compared to WT HSPC (Fig. 6B and Supplementary Fig. 10A and B). Various RNA-Seq hits involving some of the most differentially expressed genes in LSK cells from Sucnr1-KO versus WT mice and related to enhanced inflammation (*Il1b*, *S100a8*, *S100a9*)[40] and reduced self-renewal (*Procr*, *Hoxa9*) were confirmed by

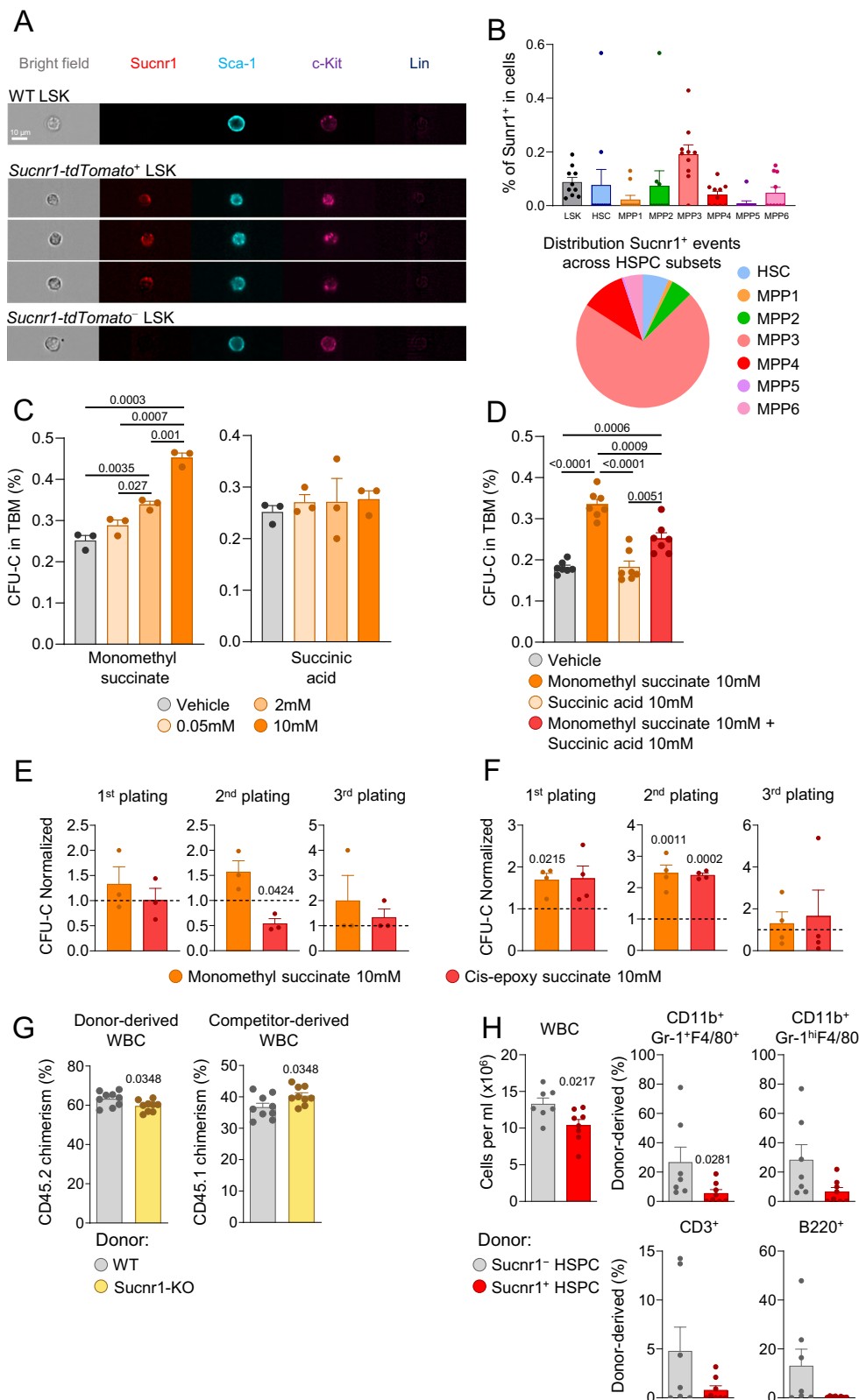

qRT-PCR (Fig. 6C). We then analyzed the levels of expression of *Procr* and *Hoxa9* in different HSPC subsets by ddPCR, and found that they were significantly reduced only in the MPP subset, but not in the LT-HSC and ST-HSC compartments (Fig. 6D).

Given the increased mRNA expression of *Il1b* in Sucnr1-KO versus WT mice, we analyzed the intracellular levels of pro-IL-1β protein and found them slightly but significantly increased in Sucnr1-KO compared to WT mice (Fig. 6E). This effect seemed to be specific of the HSPC compartment, as we observed no differences in the intracellular levels of pro-IL-1β in total live cells, lineage-positive cells, lineage-negative progenitors or LK progenitors (Supplementary Fig. 11A).

In fact, consistent with the reduced levels of succinate, we found that the protein levels of IL-1β were lower in the BM extracellular fluid of Sucnr1-KO versus WT mice, with no changes in IL-1α (Fig. 6F). To

**Fig. 5 | Sucnr1 is expressed in HSPC and its activation restricts their functional response. A** Representative images of Lin⁻Sca-1⁺c-Kit⁺ (LSK) from C57BL6/J wild-type (WT) mice, and of tdTomato⁺ LSK and tdTomato⁻ LSK from *Sucnr1-tdTomato* mice, obtained with Amnis ImageStream X (scale bar, 10 μm). **B** (Top) Proportion of Sucnr1⁺ cells in LSK subsets (*n* = 10): LSK CD34⁻Flt3⁻CD150⁺CD48⁻ hematopoietic stem cells (HSC); LSK CD34⁺Flt3⁻CD150⁺CD48⁻, multipotent progenitors 1 (MPP1); LSK CD34⁺Flt3⁻CD150⁺CD48⁺ (MPP2); LSK CD34⁺Flt3⁻CD150⁻CD48⁺ (MPP3); LSK CD34⁺Flt3⁺CD150⁻CD48⁺ (MPP4); LSK CD34⁺Flt3⁻CD150⁻CD48⁻ (MPP5); LSK CD34⁻Flt3⁻CD150⁻CD48⁻ (MPP6) analyzed by spectral flow cytometry and (bottom) pie chart representing proportions of each LSK cell subset in LSK Sucnr1⁺ events. **C** Total bone marrow (TBM) fractions of colony-forming unit cells (CFU-C) ex vivo treated for one week with succinic acid, monomethyl succinate, or PBS as vehicle (*n* = 3). **D** TBM fractions of CFU-C ex vivo treated for one week with succinic acid, monomethyl succinate, or succinic acid and monomethyl succinate or PBS as vehicle (*n* = 7). **E, F** Serial CFU-C assay with FACS-sorted HSC from WT (**E**) (*n* = 3) and Sucnr1-KO (**F**) (*n* = 4) treated with monomethyl succinate, cis-epoxy succinate or

PBS as vehicle. Data are normalized to vehicle control of each animal (dashed line). **G** Peripheral blood (PB) chimerism 16 weeks after competitive repopulation with CD45.2⁺ WT or Sucnr1-KO BM nucleated cells (BMNC) and CD45.1⁺ B6.SJL BMNC (1:1) in B6.SJL mice analyzed by FACS (*n* = 9 per group). White blood cells, WBC. **H** Number of WBC per ml of PB measured with hematological analyzer and PB chimerism in CD11b⁺Gr-1⁺F4/80⁺ monocytes, CD11b⁺Gr-1ʰⁱF4/80⁻ granulocytes, CD3⁺ T lymphocytes and B220⁺ B lymphocytes analyzed by FACS, 10 weeks after competitive repopulation with a mix of 200 Sucnr1-tdTomato⁺ or Sucnr1-tdTomato⁻ CD45.2⁺ HSC (80), MPP3 (86) and MPP4 (74) and 2.5 × 10⁵ CD45.1⁺ B6.SJL BMNC in B6.SJL mice (*n* = 7 Sucnr1⁻, *n* = 8 Sucnr1⁺, HSC and progenitor cell (HSPC) recipient mice). Statistical analyses were performed with two-tailed Student's *t* test (**C, D, G, H** WBC), two-tailed one-sample *t* test versus theoretical value of 1 (**E, F**) or two-tailed Mann–Whitney *U* test (**H** except WBC). Significant *p*-values are reported. Data are biologically independent animals, and means ± SEM for bar plots. Source data and non-significant *p*-values are provided as a Source Data file.

capture the global inflammatory status in the BM of Sucnr1-KO mice, we then studied additional pro-inflammatory cytokines, including IL-6 which showed no changes and chemokines like MIP-1β, MCP-1 and RANTES, and growth factors like GM-CSF, which were all reduced compared to WT (Fig. 6F). In turn, anti-inflammatory cytokines such as IL-9 and IL-13 were increased in the BM of Sucnr1-KO versus WT mice (Fig. 6G).

Conversely, consistent with the increased mRNA expression of *S100a8*, *S100a9* in HSPC from Sucnr1-KO versus WT mice, the protein levels of the alarmin complex S100A8/S100A9 were increased in the BMEF (Fig. 6H).

Taken together, in vivo deletion of Sucnr1 impacts gene expression programs related to impaired function and promotion of inflammation in HSPC, and particularly in the MPP compartment. Despite complex changes taking place in the global cytokine status of the BM in Sucnr1-KO mice, S100a8 and S100a9 are consistently increased in expression in HSPC and in levels in BMEF.

## Sucnr1 restricts HSPC and hematopoiesis via control of S100a8/S100a9

We next asked the potential functional link between Sucnr1 and S100a8/S100a9 by in vivo treatment of Sucnr1-KO mice with tasquinimod (Fig. 7A), a small-molecule that inhibits S100a9 signaling[41]. Tasquinimod treatment in Sucnr1-KO mice reduced the expanded numbers of circulating monocytes as early as 3 weeks after the start of the treatment. Further reductions in circulating leukocytes, including both overall myeloid cells as well as B cells, followed at 6 weeks after the start of tasquinimod administration (Fig. 7B). Consistent with these observations, tasquinimod administration reduced the numbers of monocytes and B cells in BM and SP of Sucnr1-KO mice, as well as overall myeloid cells in SP (Fig. 7C and D). The effect on both the myeloid and lymphoid lineages seemed to originate from the reduction of the expanded HSC, MPP3 and MPP6 subsets in Sucnr1-KO mice after treatment with tasquinimod as compared to vehicle-treated mice (Fig. 7E). HSPC from Sucnr1-KO mice treated with tasquinimod generated less primary and secondary colonies, but similar numbers of tertiary colonies, compared to vehicle-treated mice (Fig. 7F and G). These data suggest that Sucnr1 restricts HSPC function and myeloid and lymphoid development through inhibition of S100a8/S100a9 signaling.

To substantiate a selective role of S100a8/S100a9 in the hematopoietic system via Sucnr1 deletion, we performed an independent experiment treating both Sucnr1-KO and WT mice with tasquinimod or vehicle in parallel (Supplementary Fig. 12A). Confirming our previous results, tasquinimod treatment in Sucnr1-KO mice reduced the expanded numbers of circulating leukocytes to the numbers observed in WT mice, including both overall myeloid cells as well as B cells and with no significant effect observed in WT mice

(Supplementary Fig. 12B). Tasquinimod administration reduced the expanded numbers of monocytes in BM of Sucnr1-KO mice to the numbers found in WT mice, whilst no effect was observed in WT mice (Supplementary Fig. 12C and D). Conversely, treatment with tasquinimod reduced the numbers of BM B cells to similar levels in both Sucnr1-KO and WT mice (Supplementary Fig. 12C and D). The expansion of BM HSC in Sucnr1-KO mice was reduced by treatment with tasquinimod, but a minor reduction in HSC numbers was also observed in WT mice, bringing HSC values in both tasquinimod-treated groups to similar level (Supplementary Fig. 12C and E). In turn, the increased colony-forming potential in Sucnr1-KO mice was reduced to WT values, with no effect seen in samples obtained from WT mice (Supplementary Fig. 12F). Of note, this assay is optimal for the growth of primitive erythroid progenitor cells, granulocyte-macrophage progenitor cells, and multi-potent granulocyte, erythroid, macrophage, megakaryocyte progenitor cells, but not for the growth of lymphoid progenitor cells. Thus, these data suggest that tasquinimod effect originates in the HSC compartment for both lineages, and whereas the effect on the myeloid lineage is fully dependent on Sucnr1 deletion, the effect on the B cell lineage is only partially dependent on the loss of Sucnr1 signaling.

To confirm a direct role of Sucnr1 and S100a8/S100a9 on HSPC function, FACS-sorted HSC from Sucnr1-KO mice were tested for their colony-forming potential in serial replatings in the presence of tasquinimod. Tasquinimod inhibited the colony-forming potential of Sucnr1-KO HSC at first plating compared to vehicle, confirming the functional role of elevated S100a8/S100a9 expression in Sucnr1-KO HSPC (Fig. 7H). No changes were observed at second and third plating, suggesting no impact on long-term colony forming potential.

Together, Sucnr1 restricts haematopoiesis through inhibition of S100a8/S100a9 signaling in HSPC.

## Therapeutic potential of Sucnr1 and S100a8/S100a9 targeting in acute myeloid leukemia

To understand the effect of Sucnr1 via control of S100/a8 and S100/a9 in malignant hematopoiesis, we first analyzed the expression level of *S100a8* and *S100a9* mRNA in mouse NRAS-G12D⁺ HSPC. In pre-leukemic NRAS-G12D⁺ HSC, MPP1 and MPP2-3 versus control cells, low *Sucnr1* expression correlated with *S100a8* and *S100a9* upregulation (Supplementary Fig. 13A). To study the therapeutic potential of targeting Sucnr1 signaling pathway in AML, we turned to the more aggressive *Mll*-AF9 mouse model, which expressed very low levels of *Sucnr1* (Fig. 2G). We transplanted MLL-AF9⁺ BM into lethally irradiated recipients and treated them with tasquinimod, tasquinimod in combination with cis-epoxy succinate, or the corresponding vehicles, for a maximum of 24 weeks (Fig. 8A). The combination of tasquinimod and cis-epoxy succinate reduced the numbers of circulating leukocytes (Fig. 8B) and Gr-1ʰⁱ blasts in spleen (Fig. 8C),

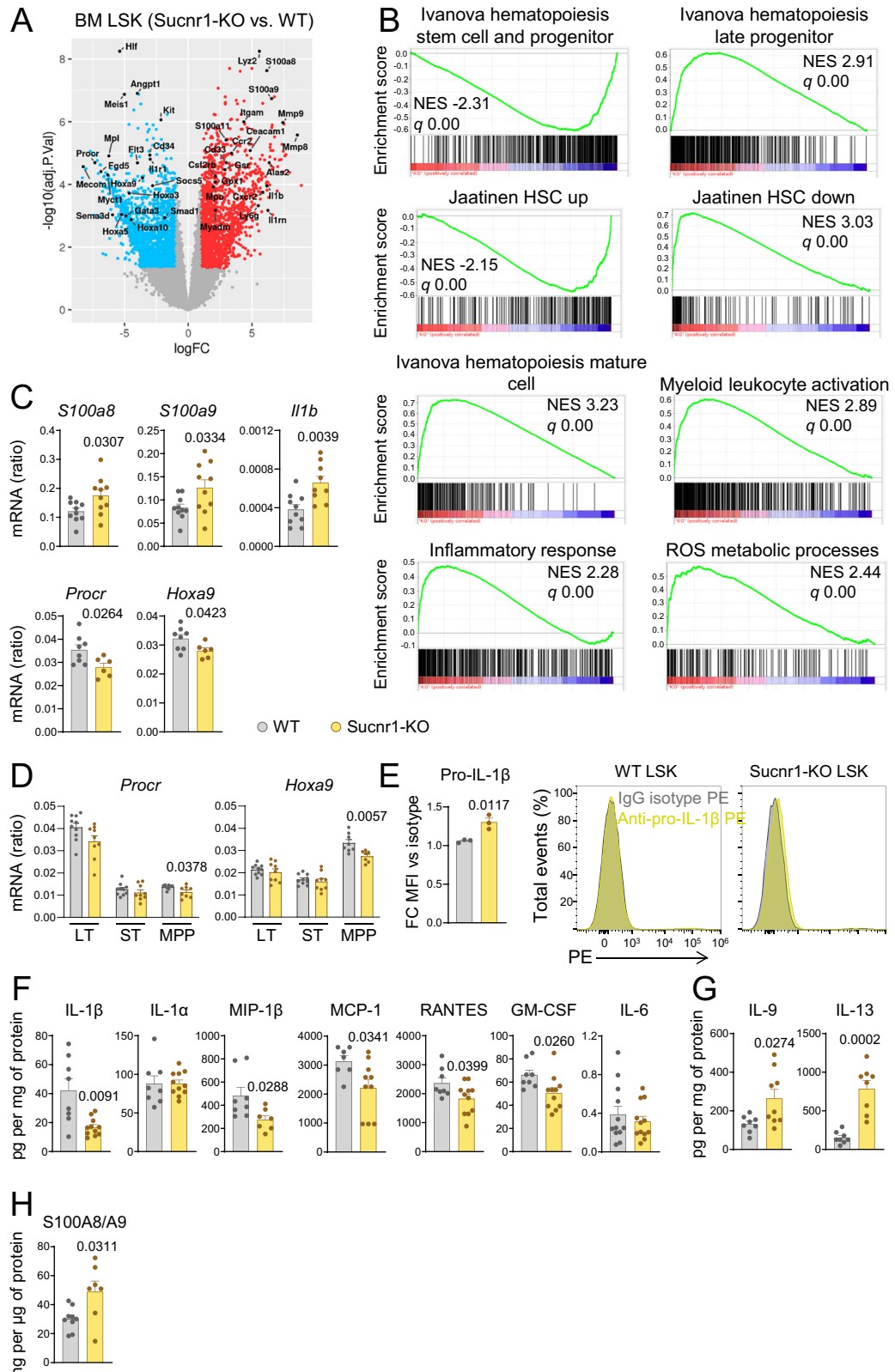

compared to mice receiving the vehicle for both drugs. In the BM, we found that tasquinimod alone decreased the numbers of c-Kit⁺FcRγ⁺ GMP-like leukemic blasts[42] versus mice treated with vehicle (Fig. 8D and E). Further, tasquinimod reduced the number of BM c-Kit⁺FcRγ⁻CD34⁺CD48⁺ lymphoid-primed multipotent progenitor (LMPP) LSC[43,44] when administered alone or in combination with cis-epoxy succinate, versus the corresponding vehicle-treated mice

(Fig. 8D and E). These data indicate that activating Sucnr1 signaling with a potent Sucnr1 agonist and its downstream pathway through S100A9 targeting shows therapeutic value in MLL-AF9⁺ mice, reducing the numbers of immunophenotypically-defined LSC in the BM and leukemic burden in periphery.

In humans, low *SUCNR1* expression associated with high *S100A8* and *S100A9* expression in M4–M5 patients versus M0–M3, in publicly

**Fig. 6 | Sucnr1 preserves transcriptional programs characteristic of HSPC. A, B** RNA sequencing in fluorescence-activated cell sorting (FACS)-sorted Lin⁻Sca-1⁺c-Kit⁺ (LSK) from bone marrow (BM) of 27–30 week-old C57BL6/J wild-type (WT, $n = 4$) and Sucnr1 knock-out (Sucnr1-KO, $n = 3$) mice. **A** Volcano plot. Red and blue dots represent differentially expressed genes with adjusted p˂0.05 and absolute log₂FC > 1. Fold-change, FC. **B** Gene Set Enrichment Analysis (GSEA). HSC, hematopoietic stem cells; ROS, reactive oxygen species. **C** Digital droplet PCR (ddPCR) mRNA expression of *S100a8* ($n = 10$ per group), *S100a9* ($n = 10$ per group), *Il1b* ($n = 10$ WT, $n = 9$ Sucnr1-KO), *Procr* ($n = 8$ WT, $n = 6$ Sucnr1-KO) and *Hoxa9* ($n = 8$ WT, $n = 6$ Sucnr1-KO) relative to *Gapdh* in FACS-sorted LSK of 35–44 week-old mice. **D** ddPCR mRNA expression of *Procr* ($n = 10$ LT, ST and $n = 8$ MPP WT, $n = 9$ LT, ST and $n = 8$ MPP Sucnr1-KO) and *Hoxa9* ($n = 10$ LT, ST and $n = 8$ MPP WT, $n = 9$ LT, ST and $n = 7$ MPP Sucnr1-KO) relative to *Gapdh* in FACS-sorted LSK subsets of 30–45 week-old mice; long-term HSC (LT, LSK CD34⁻Flt3⁻), short-term HSC (ST, LSK CD34⁺Flt3⁻) and multipotent progenitors (MPP, LSK CD34⁺Flt3⁺). **E** (Left)

Intracellular pro-IL-1β in LSK-gated cells from total BM of 43 week-old WT and Sucnr1-KO mice ($n = 3$ per group) analyzed by FACS. Mean fluorescence intensity, MFI. (Right) Representative histograms in LSK-gated cells using spectral flow cytometry. **F** Pro-inflammatory IL-1β, IL-1α, MIP-1β, MCP-1, RANTES, GM-CSF levels in BM extracellular fluid (BMEF) from 40-45 week-old WT ($n = 8$) and Sucnr1-KO ($n = 11$) mice and IL-6 level in BMEF from 19 to 35 week-old WT ($n = 12$) and Sucnr1-KO ($n = 13$) male and female mice, analyzed by Bioplex. **G** Anti-inflammatory IL-9 and IL-13 levels in BMEF from 40-45 week-old WT ($n = 8$) and Sucnr1-KO ($n = 9$ IL-9, $n = 8$ IL-13) mice, analyzed by Bioplex. **H** Levels of S100A8 and S100A9 in BMEF of 41-46 week-old WT ($n = 9$) and Sucnr1-KO ($n = 7$) mice, analyzed by ELISA. Statistical analyses were performed with two-tailed Wald test with Benjamini-Hochberg correction for multiple comparisons (**A**), two-tailed Student's *t* test (**C–E**, **F** except IL-1β, **H**), or two-tailed Mann–Whitney *U* test (**F** IL-1β, **G**). Significant *p*-values are reported. Data are biologically independent animals, and means ± SEM for bar plots. Source data and non-significant *p*-values are provided as a Source Data file.

available arrays from purified AML blasts in a published cohort of AML patients (GSE14468; $n = 543$) (Fig. 9A). We then analyzed single-cell RNA-sequencing (scRNA-seq) data from 2 AML patient-derived xenografts (PDX) before and 8 days after the start of cytarabine (AraC), following a 5-day treatment starting at diagnosis (GSE178912[45]; Fig. 9B and Supplementary Fig. 14A). After integration of the data with the Harmony algorithm, we performed unbiased clustering using K-nearest neighbor analysis (Fig. 9B), which identified 7 clusters (0–6). Following AraC treatment, clusters 0 and 1 expanded markedly (Fig. 9B). Together with cluster 5, these clusters displayed a shared transcriptional program characterized both by the residual disease signature post-AraC (AraC_UP; Farge et al.)[46] and by an inflammatory/senescence–stress signature (CISG_UP; Duy et al.)[47] (Fig. 9C). Importantly, we found *S100A8* and *S100A9* greatly expressed in these clusters of persistent cells enriched post-AraC (clusters 0, 1, 5) (Fig. 9D). Their expression was significantly increased post-AraC compared to diagnosis (Fig. 9E and F) and correlated with the residual disease signature[46] (Fig. 9G). In contrast, *SUCNR1* was expressed at low levels in cluster 3, which decreased in numbers after AraC treatment (Fig. 9B) and showed low expression of *S100A8* and *S100A9* (Fig. 9D and E). Functional enrichment analysis revealed that cluster 3 was enriched for mitochondrial and metabolic processes (Supplementary Fig. 14B). This finding is consistent with previous studies showing that AraC treatment induces a metabolic rewiring characterized by enhanced mitochondrial activity, a key driver of chemoresistance[48]. Thus, the reduction of cluster 3 post-AraC suggests that *SUCNR1*-expressing cells represent a metabolically active population with a distinct role in shaping treatment response. In this context, *SUCNR1* expression could represent both an important determinant of chemotherapy response and a potential therapeutic vulnerability.

To test the role of SUCNR1 in chemotherapy response, we treated NOMO-1 or Kasumi-1 AML cell lines with AraC alone or in combination with cis-epoxy succinate. Cis-epoxy succinate increased AraC-induced apoptosis rate in both cell lines (Supplementary Fig. 15A and B). The combination of cis-epoxy succinate but not monomethyl succinate and AraC reduced the expression of *S100A9*, versus AraC alone (Supplementary Fig. 15C). In view of these data, we then asked the potential of SUCNR1 activation to induce apoptosis in human AML cell lines. To answer this question, we used a lentivector to overexpress *SUCNR1* in Kasumi-1 cells and treated the cells with cis-epoxy succinate. Notably, cis-epoxy succinate treatment alone was able to induce apoptosis, and the rate of apoptosis was significantly stronger in cells overexpressing *SUCNR1* versus cells transfected with an empty vector (Supplementary Fig. 15D and E).

Collectively, these data indicate that *SUCNR1* and *S100A8*/*S100A9* are linked and functionally relevant in human AML. Whereas *S100A8* and *S100A9* are markers of residual disease after chemotherapy in AML PDX models, *SUCNR1* selectively marks a metabolically active cell

subset, and its activation induces apoptosis and potentiates chemotherapy response in vitro.

## Discussion

Sucnr1 has been extensively studied in the context of inflammation, where it has been proposed to have both pro- and anti-inflammatory roles depending on cell type and condition[14–17], but here we uncover its role on hematopoiesis.

Functionally, our data shows that Sucnr1 signaling has a role in the restriction of hematopoiesis and the HSPC pool in both health and malignancy, with a major cell-autonomous effect. Under steady-state, loss of Sucnr1 in vivo induces excess B cell and myeloid lineage expansion and our results suggest that these effects are originated at least partially in the HSPC compartment, as a fraction of HSPC expresses Sucnr1 and HSC show ex vivo a differential functional response to cell-permeable succinate analogues active or inactive on Sucnr1. Absolute numbers and relative frequencies of immunophenotypically defined subsets of HSPC, with exception of MPP2, and numbers of primary and secondary colonies ex vivo were increased after Sucnr1 deletion, which was concomitant with a higher survival rate in HSPC, but no impact on tertiary colonies and a mild reduction in HSPC engraftment analyzed in in vivo BM competitive repopulating assays. The hematological consequences of in vivo deletion of Sucnr1 are further propagated through absolute expansion in committed lineage progenitors, including MEP, GMP and CLP. Conversely, genetic deletion of Sucnr1 resulted in a reduced fraction of cells in the G1 phase of the cell cycle, suggesting that activation of Sucnr1 promotes mouse HSPC proliferation, as previously demonstrated in human CD34⁺ progenitors in vitro[19].

These seemingly opposing effects of Sucnr1 on HSPC numbers/survival/colony-forming potential versus engraftment in competition/proliferation may be important factors underlying the chronic nature of the hematopoietic abnormalities displayed by Sucnr1-KO mice, which become symptomatic only at the middle-age. This paradox between numbers and fitness is common to other chronic conditions, like ageing and related chronic inflammation, where the HSPC pool is increased in numbers but reduced in function[49,50], representing a shift towards an abnormal, biased and less effective hematopoietic system. The potential role of Sucnr1 in the ageing hematopoietic system should be subject of future work. Notably, although Sucnr1-KO BM cells are less fit in competition with equal numbers of WT cells, primary transplants of *Mx1-Cre Sucnr1ᶠˡ/ᶠˡ* BM cells into C57BL/6 J WT mice, induced only after stable engraftment, greatly recapitulate the hematopoietic expansion observed in Sucnr1-KO mice. Further, fixed numbers of Sucnr1-tdTomato⁺ FACS-sorted HSPC displayed restricted engraftment in vivo compared to Sucnr1-tdTomato⁻ HSPC, providing direct evidence that Sucnr1 expression traces a subset of HSPC with restricted hematopoietic potential.

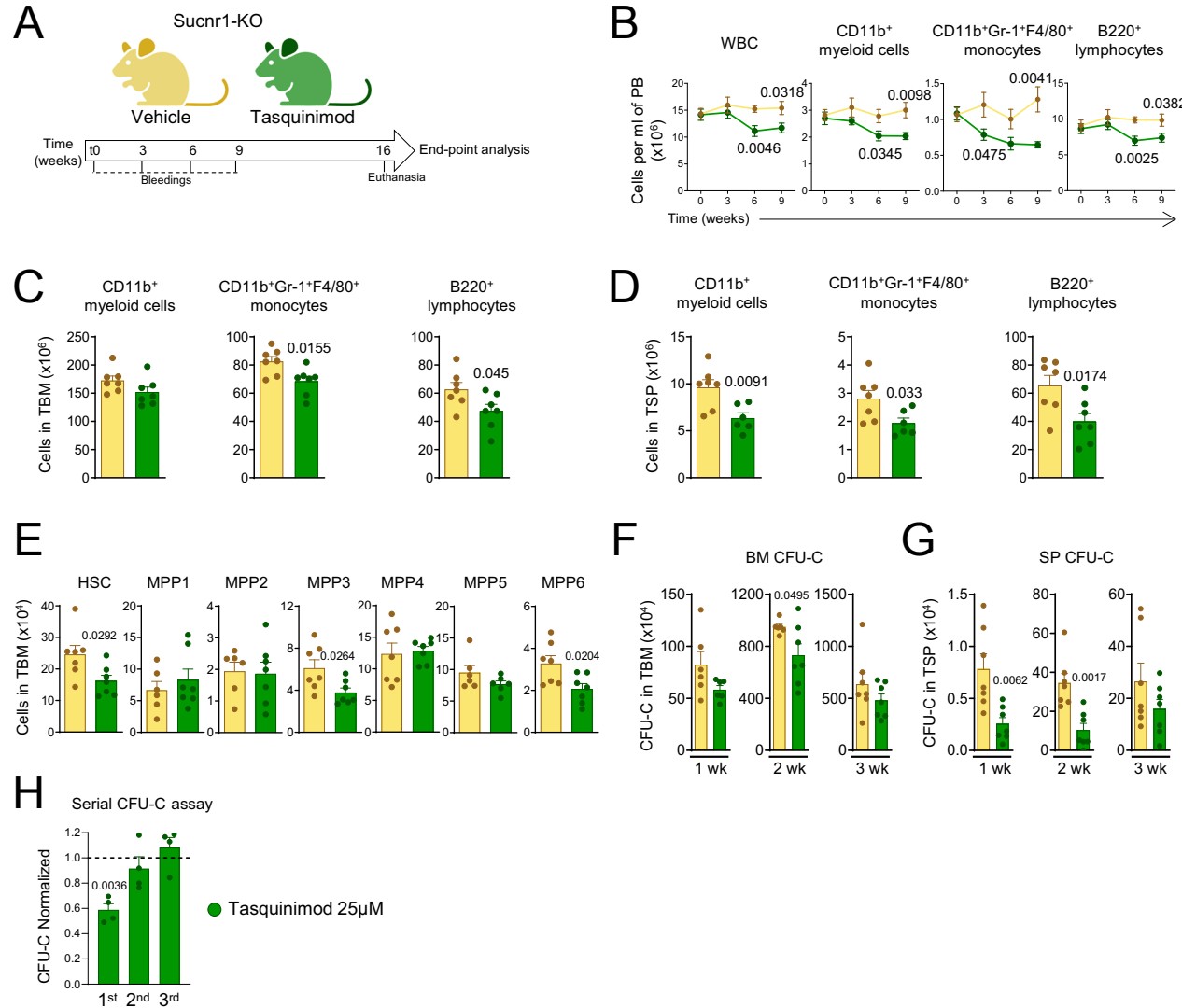

**Fig. 7 | Sucnr1 restricts HSPC and hematopoiesis via control of S100a8/S100a9.**
**A–G** Treatment of Sucnr1 knock-out (Sucnr1-KO) mice with S100A9 inhibitor tas-
quinimod or vehicle, starting at the age of 40–41 weeks and for 16 weeks.
**A** Illustration of the experimental design. The mouse icons: Created in BioRender.
Cuminetti, V. (2026) https://BioRender.com/9bm56nx. **B** Evolution in number of
white blood cells (WBC) per ml of peripheral blood (PB) measured with hemato-
logical analyzer, and in numbers of CD11b[+] myeloid cells, CD11b[+]Gr-1[+]F4/80[+]
monocytes and B220[+] B lymphocytes per ml of PB analyzed by fluorescence-
activated cell sorting (FACS) (*n* = 7 per group, except *n* = 6 for B lymphocytes
vehicle at week 3). **C** Total bone marrow (TBM) numbers of CD11b[+] myeloid cells,
CD11b[+]Gr-1[+]F4/80[+] monocytes and B220[+] B lymphocytes, analyzed by FACS (*n* = 7
per group). **D** Total spleen (TSP) number of CD11b[+] myeloid cells (*n* = 7 vehicle, *n* = 6
tasquinimod), CD11b[+]Gr-1[+]F4/80[+] monocytes (*n* = 7 vehicle, *n* = 6 tasquinimod) and
B220[+] B lymphocytes (*n* = 7 per group) in TSP cells, analyzed by FACS. **E** TBM
numbers of Lin[−]Sca-1[+]c-Kit[+] (LSK) cell subsets: LSK CD34[−]Flt3[−]CD150[+]CD48[−],
hematopoietic stem cells (HSC, *n* = 7 per group); LSK CD34[+]Flt3[−]CD150[+]CD48[−],

multipotent progenitors 1 (MPP1, *n* = 6 vehicle, *n* = 7 tasquinimod); LSK
CD34[+]Flt3[−]CD150[+]CD48[+] (MPP2, *n* = 6 vehicle, *n* = 7 tasquinimod); LSK
CD34[+]Flt3[−]CD150[−]CD48[+] (MPP3, *n* = 7 per group); LSK CD34[+]Flt3[+]CD150[−]CD48[+]
(MPP4, *n* = 7 per group); LSK CD34[+]Flt3[−]CD150[−]CD48[−] (MPP5, *n* = 6 per group); LSK
CD34[−]Flt3[−]CD150[−]CD48[−] (MPP6, *n* = 7 per group), analyzed by FACS. **F** TBM (*n* = 6
per group wk 1, *n* = 6 vehicle and *n* = 7 tasquinimod wk 2, *n* = 7 per group wk 3) and
**G** TSP (*n* = 7 per group) numbers of colony-forming unit cells (CFU-C) ex vivo after
serial replating (wk, week). **H** Serial CFU-C assay with FACS-sorted HSC from Sucnr1-
KO treated with tasquinimod or PBS as vehicle. Data are normalized to vehicle
control of each animal (dashed line). Statistical analyses were performed with two-
tailed Student's *t*-test (**A**, **B** except monocytes at 9 weeks, **C**–**G**), two-tailed
Mann−Whitney *U* test (**B** monocytes at 9 weeks) or two-tailed one-sample *t*-test
versus theoretical value of 1 (**H**). Significant *p*-values are reported. Data are biolo-
gically independent animals, and means ± SEM for bar plots. Data are mean ± SEM
in (**B**) for better visualization. Source data and non-significant *p*-values are provided
as a Source Data file.

We further provide data supporting a functional role of Sucnr1
restricting HSPC mobilization and extramedullary hematopoiesis in
SP. However, one single injection of succinic acid was unable to alter
HSPC mobilization from BM to periphery, consistent with the overall
chronic nature of Sucnr1 effect on the hematopoietic system.

As mentioned above, conditional deletion of Sucnr1 in the
hematopoietic compartment greatly recapitulates the effect of global
deletion of Sucnr1, supporting for a major intrinsic effect of Sucnr1 to
hematopoietic cells. *Sucnr1-tdTomato* reporter mice showed

expression of Sucnr1 to various degrees in lineage negative progeni-
tors and HSPC, where it is most expressed in MPP3 followed by MPP4.
Further, we revealed that activation of Sucnr1 represses the stimula-
tory effect of cell-permeable succinate in HSC colony-forming poten-
tial, an effect dependent on the presence of Sucnr1 in HSPC, whilst
Sucnr1-tdTomato[+] HSPC have restricted engraftment potential in vivo
compared to equal numbers of Sucnr1-tdTomato[−] HSPC. In addition,
deletion of Sucnr1 promotes a global anti-inflammatory environment
and low levels of succinate in the BM while inducing activation of HSPC

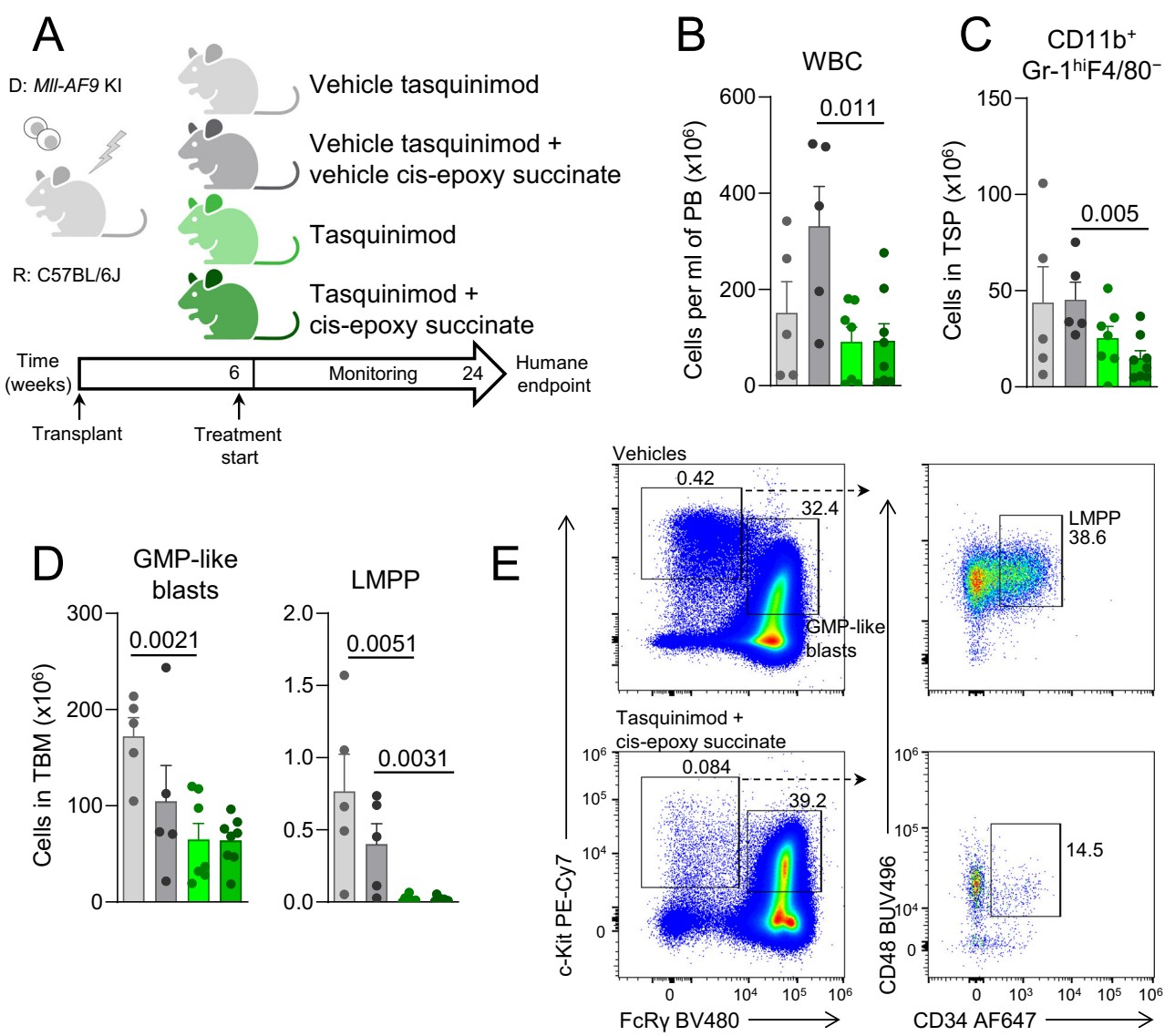

**Fig. 8 | Therapeutic potential of Sucnr1 and S100a9 targeting in mouse models of acute myeloid leukemia.** Treatment with S100a9 inhibitor tasquinimod ($n = 7$), tasquinimod together with cis-epoxy succinate ($n = 8$), or corresponding vehicle ($n = 5$ per group) in C57BL6/J wild-type (WT) mice transplanted with bone marrow nucleated cells (BMNC) from leukemic *Mll*-AF9 mice, treated for a maximum of 24 weeks starting 5 weeks after the transplant. **A** Illustration of the experimental design. The mouse, cell and beam icons: Created in BioRender. Cuminetti, V. (2026) https://BioRender.com/9bm56nx. **B** Number of white blood cells (WBC) measured with hematological analyzer at endpoint analysis. **C** Total spleen (TSP) numbers of CD11b⁺Gr-1ʰⁱF4/80⁻ granulocytes at endpoint analysis. **D** Total BM

(TBM) number of c-Kit⁺FcRγ⁺ granulocyte monocyte progenitors (GMP)-like leukemic blasts and c-Kit⁺FcRγ⁻CD34⁺CD48⁺ lymphoid-primed multipotent progenitor (LMPP) leukemic stem cells. **E** Representative gating strategy for FACS analysis and fractions of cells in TBM using spectral flow cytometry, of GMP-like blasts and LMPP in TBM. Statistical analyses were performed with two-tailed Student's *t* test (**B**–**D** except LMPP) or two-tailed Mann–Whitney *U* test (**D** LMPP). Significant *p*-values are reported. Data are biologically independent animals, and means ± SEM for bar plots. Source data and non-significant *p*-values are provided as a Source Data file.

---

with a pro-inflammatory gene signature and higher content of succinate in HSC. Although partial contribution of the BM stem cell microenvironment or metabolic systemic effects[51–53] with potential to influence hematopoiesis will require future investigation, we provide compelling evidence that supports for a direct, selective role of Sucnr1 on HSPC. Further, the regulation of succinate localization by Sucnr1 should be subject of future work. Whereas it could simply be a matter of increased succinate availability in the absence of Sucnr1 binding, active regulatory mechanisms could additionally take place. In addition, although most Sucnr1-expressing cells in the BM were identified as lineage-negative, we found expression of Sucnr1 across all the lineage-positive cells analyzed. Future studies should explore the potential role of Sucnr1 signaling in the regulation of their primary and effector functions.

Consistent with the mild reduction in HSPC performance in in vivo BM competitive repopulating assays, our genomic studies in HSPC further revealed transcriptional repression of gene signatures associated with stemness and primitive HSC, and instead activation of transcriptional programs characteristic of late progenitors and mature hematopoietic cells as well as functional categories related to myeloid activation, ROS and the previously mentioned inflammatory response, in Sucnr1-KO versus WT HSPC. We demonstrate that Sucnr1 restricts myeloid and lymphoid development through inhibition of S100a8/S100a9 signaling in healthy HSC and progenitors. We found increased expression of *S100a8* and *S100a9* in LSK cells from Sucnr1-KO versus WT mice and the protein levels of the complex S100A8/S100A9 were also increased in the BMEF after Sucnr1 deletion. Further, S100A9 pharmacological inhibition rescued the hematopoietic and HSPC

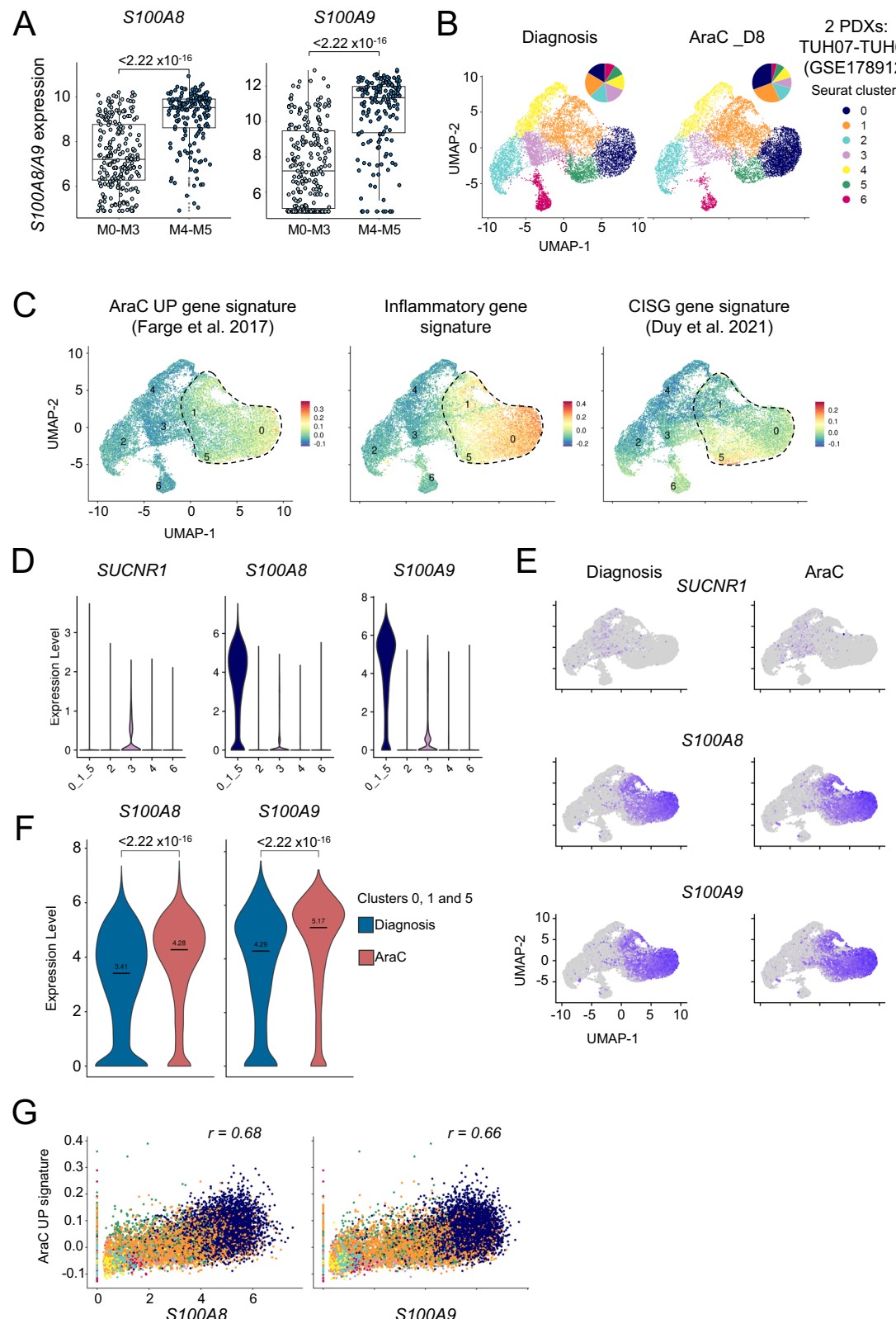

defects in Sucnr1-KO mice in vivo and ex vivo. Importantly, our data show that tasquinimod effect originates in the HSC compartment for both lineages, and whereas the effect on the myeloid lineage is fully dependent on Sucnr1 deletion, the effect on the B cell lineage is only partially dependent on Sucnr1 signaling loss.

Taken together, in vivo deletion of Sucnr1 promotes inflammation specifically in HSPC, with selective increases in IL-1β mRNA and pro-IL-

1β protein levels that are not reproduced in other hematopoietic cells or in the BMEF. The global cytokine status of the BM in Sucnr1-KO mice displayed complex changes, which involved reduced levels of IL-1β protein. These discrepancies may be caused by multiple factors which should be subject of future work. Despite this complexity, we were able to uncover the primary mechanistic role of S100A8/S100A9 in the hematopoietic changes displayed by Sucnr1-KO mice. The control of

**Fig. 9 | Therapeutic potential of SUCNR1 and S100A8/S100A9 targeting in human acute myeloid leukemia. A** *S100A8* and *S100A9* expression in patients categorized as M0-M3 (*n* = 224) and M4-M5 (*n* = 171) in GSE14468. **B–G** single-cell RNA-sequencing data analyses in fluorescence-activated cell sorting (FACS)-sorted human CD45⁺CD33⁺ cells from the bone marrow of two patient-derived xenograft (PDX) models (GSE178912, TUH07, TUH69) at day 0 (*n* = 2 samples derived from 2 patients and obtained from 4 to 5 animals, respectively) and at day 8 following a 5-day treatment with 30 mg/kg cytarabine (AraC, *n* = 2 samples derived from the same 2 patients and obtained from 5 animals, respectively). **B** Uniform manifold approximation and projection (UMAP) plot of human AML blasts (CD45⁺CD33⁺) using Seurat. Colors indicate k-means clusters (*k* = 7). Pie-chart inserts represent proportion of each cluster. **C** Feature plots show the expression specificity of the AraC UP, inflammatory and CISG gene signatures after AraC treatment. Expression level is indicated by a color gradient from light blue to red. Dashed line demarcates

the clusters that share these transcriptional programs. **D** Expression of *SUCNR1*, *S100A8* and *S100A9* genes after AraC in clusters 0, 1 and 5 combined and in other clusters. **E** Feature plots show the expression of *SUCNR1*, *S100A8* and *S100A9* gene at diagnosis and after AraC. **F** Expression of *S100A8* and *S100A9* genes in clusters 0, 1 and 5 combined before and after AraC. **G** Correlation between expression of *S100A8* or *S100A9* genes and AraC residual disease signature. Pearson's coefficient correlations are indicated. Statistical analyses were performed with two-tailed Student's *t*-test (**A**, **F**). Significant *p*-values are reported. Data are biologically independent human samples and box plots in (**A**) showing median as center line; upper and lower quartiles as box limits; 1.5x interquartile range as whiskers. Data are presented as violin plots for (**D**) and violin plots with median for (**F**) for better visualization. Source data are provided as a Source Data file and Supplementary Data 1.

Sucnr1 over the alarmin axis S100a8/S100a9 was mostly unknown to date, but interestingly both pathways are linked to inflammatory responses and Ca²⁺ [8,14,15,17,40]. Various studies have shown that Sucnr1 activation results in increased intracellular Ca²⁺ [8,54,55], whereas S100A8/S100A9 are considered Ca²⁺ biosensors that change their conformation in response to Ca²⁺ binding and thereby their affinity to bind other proteins[56]. Like succinate, S100A8 and S100A9 can be secreted to the extracellular space, via an active Ca²⁺-dependent mechanism, and signal through a variety of receptors[56]. Whether Sucnr1 controls S100a8/S100a9 signaling with participation of a Ca²⁺-dependent mechanism should be investigated in future work.

Overall, our findings support for a function of Sucnr1 signaling as a negative regulator over the intracellular effect of succinate on HSPC and hematopoiesis. Remarkably, monomethyl succinate, cell-permeable but inert on Sucnr1[38], stimulated the colony-forming potential of FACS-sorted HSC in second replating. Conversely, cis-epoxy succinate, powerful Sucnr1 agonist, reduced the colony-forming potential of HSC at the same stage. Further, monomethyl succinate enhances the colony-forming potential in total BM, in a dose-response manner. Conversely, succinic acid, which displays both Sucnr1-dependent and independent effects, results in unchanged colony-forming potential. When used in combination, the results showed a partial increase in colony-forming potential versus succinic acid alone and a partial reduction versus monomethyl succinate alone, providing robust evidence that Sucnr1 activation represses the intracellular effect of succinate. In turn, similar concentrations of monomethyl succinate and cis-epoxy succinate induced the same increased functional output in HSC in the absence of Sucnr1, but it is mostly unknown whether the underlying mechanisms are exactly the same for both compounds. Thus, in the absence of Sucnr1, cis-epoxy succinate enhances colony-forming potential through receptor-independent mechanisms, possibly with contribution to intracellular metabolic modulation. However, given the structural differences among different succinate analogues and the multi-layer intracellular functions of succinate, this question should be subject of future work.

Further, together with the increased intracellular levels of succinate in HSC, we found stimulation of basal oxygen consumption in lineage-negative progenitors and high total and mitochondrial ROS levels in LSK cells from Sucnr1-KO mice, suggesting increased fueling of the TCA cycle by succinate. Conversely, injection of succinic acid in mouse models of myeloid malignancies and human AML xenografts that display low *Sucnr1* expression promotes disease progression and mortality. In agreement with these data, succinate increases early in the BM extracellular space, and this is concomitant to the early expansion of HSPC in the NRAS-G12D⁺ mouse model. Notably, intracellular succinate activates metabolism and may have multiple additional effects, but treatment with tasquinimod, S100a9 blocker, rescued the hematopoietic defects of Sucnr1-KO HSPC in vivo and ex vivo. Further, tasquinimod combined with cis-epoxy succinate showed potent therapeutic value in AML mice. Importantly, the

combination of cis-epoxy succinate but not monomethyl succinate and AraC reduced the expression of *S100A9*, versus AraC alone, in human AML cell lines. Thus, S100a9 is an alternative pathway controlled by Sucnr1, independent of intracellular succinate, which counterbalances the stimulatory effect of intracellular succinate.

The functional role of Sucnr1 in hematopoiesis clinically translates into low *SUCNR1* representing a poor prognostic marker in AML patients whilst succinate contributing to leukemia progression in mouse models of myeloid malignancies that display low Sucnr1 expression. *SUCNR1* expression level varies across AML molecular signatures, but low *SUCNR1* expression was found in the dataset TARGET AML in primary AML versus control samples and it was associated with reduced overall and progression-free survival rate of AML patients in publicly available arrays from purified AML blasts (GSE14468)[20–22]. NRAS mutations were linked to a trend towards reduced levels of *SUCNR1* expression in human AML blasts and BM nucleated cells from NRAS-G12D⁺ and MLL-AF9⁺ mice expressed less *Sucnr1* than their healthy counterparts. Importantly, *S100A8* and *S100A9* are overexpressed in some AML subgroups, mainly M4 and M5 subtypes of the FAB classification[57], where the expression of *SUCNR1* is lowest. We also found overexpression of *S100a8* and *S100a9* in HSPC subsets from NRAS-G12D⁺ versus control mice, with exception of MPP4 where the expression of both genes is reduced. How the differential expression of *S100a8/S100a9* in selective HSPC subsets is regulated through potential *Sucnr1* expression variations and its contribution to the chronicity of NRAS-G12D-driven pre-leukemic disease should be subject of future work. S100a8 regulates S100a9 activity and sustains AML immature phenotype, whereas S100a9 induces differentiation and growth arrest of AML cells. Expression of *S100A8* predicts poor survival in AML patients[58].

Further, consistent with low *SUCNR1* expression, we found high *S100A8* and *S100A9* in publicly available single-cell RNA-sequencing data from persistent cells after cytarabine treatment in AML PDX. Clusters that expanded after cytarabine displayed a shared transcriptional program characterized both by the residual disease signature post-AraC (AraC_UP; Farge et al.)[46] and by an inflammatory/senescence–stress signature (CISG_UP; Duy et al.)[47], and they expressed high levels of *S100A8* and *S100A9*. The expression of *S100A8* and *S100A9* was increased after treatment and correlated with the residual disease signature[46]. *SUCNR1* was expressed at low levels in a different cluster, which constricted after treatment, showed low expression of *S100A8* and *S100A9* and was enriched for mitochondrial processes, key driver of chemoresistance[48]. In this context, *SUCNR1* expression could represent both an important determinant of chemotherapy response and a potential therapeutic vulnerability. In support of this idea, our work shows that *SUCNR1* activation in vitro stimulates apoptosis and chemotherapy response in human AML cell lines. These results align but deepen the understanding provided by previous reports showing that S100A8/A9⁺ AML cells display a distinct metabolic, differentiation, and chemoresistance profile, compared to

S100A8/A9⁻ AML cells which are therapy-sensitive and found mainly in periphery[59]. Using cells lines and in vitro cultures, stromal cell–derived IL-6 induced expression of S100A8/A9 in AML cells in a JAK/STAT3-dependent manner[59]. Our data in vivo supports Sucnr1 as an alternative pathway controlling S100A8/A9 independently of IL-6. Boosting SUCNR1 and blocking S100A9 thus may open potential therapeutic avenues in the treatment of myeloid malignancies.

Taken together, we provide robust evidence pointing to Sucnr1 signaling as a pathway restricting hematopoiesis at least partially through the HSPC compartment and via control of S100a8/S100a9, in both healthy conditions and myeloid neoplasias. In AML patients, low levels of expression of *SUCNR1* predict poor prognosis and could represent both an important determinant of chemotherapy response and a potential therapeutic vulnerability.

## Methods

### Ethic statement
Our research complies with all relevant ethical regulations. This human study was approved by the Regional Committee for Medical Research Ethics North Norway (REC North 2015/1082). Written informed consent was obtained in accordance with the Norwegian legislation and the Declaration of Helsinki. Participants received no compensation. Mouse experiments were conducted with the ethical approval of the Norwegian Food and Safety Authority (8660, 13407, 24739, 30992), the local Animal Care and Ethics Committees at UiT – The Arctic University of Norway (09/16 and 09/21) and University of Oslo (Norway; A022). The maximal leukemic burden permitted was leukocytosis 5-fold over normal healthy values and it was not exceeded. Criteria applied to define humane endpoints for mouse termination before the established endpoint were in accordance with the Norwegian Food and Safety Authority. Briefly, we used a cumulative score sheet to evaluate welfare that included parameters related to appearance, discomfort symptoms, eyes appearance, body weight, and injection site reaction when applicable. Human endpoints were applied under a total score > 9, and euthanasia was considered under total scores of 6–9.

### Humans
Two females, one healthy donor and one AML patient, aged 48–62 years old, were included in the study. The healthy donor donated peripheral blood and the AML patient donated peripheral blood and bone marrow. Sex was not considered in the study design. The AML patient was classified as M1 with adverse risk and was selected based on the low expression of *SUCNR1*. The healthy individual was selected according to similarity to the AML patient regarding age and sex. The diagnosis of AML was established according to the revised criteria of the World Health Organization. Cytogenetic risk group was established according to Dohner and colleagues. Myeloblasts were calculated by morphological evaluation of BM smears, complemented by regular flow cytometry.

### Mice
Age and sex matched C57BL/6 J WT (The Jackson Laboratory, strain 000664), immunodeficient NOD Scid Gamma (NSG)[60] (The Jackson Laboratory, strain 005557), *Mll-AF9*[60] (The Jackson Laboratory, strain 009079), *Mx1-Cre NRAS*^G12D[26,61] (gift from P. García), *Sucnr1-tdTomato* (generated in this paper), *Sucnr1*⁻ᐟ⁻[52] and *Sucnr1*^fl/fl (gift from A. Cervera), *Mx1-Cre*[62] (The Jackson Laboratory, strain 003556) and C57BL/6 J Ly5.1 (Charles River Laboratories, strain 708) were used in experiments. NSG mice are on a NOD/ShiLtJ background, *Mll-AF9* are on a mixed genetic background (C57BL/6, 129 P and possibly other contributions) and all other mice are on a C57BL6/J background. The sex of the mice used in each experiment is indicated. Briefly, either female or male mice were used in most experiments, and in experiments where both females and males were used, no sex-based analysis was

conducted to maintain statistically relevant experimental group sizes. Phenotyping of *Sucnr1* mice versus C57BL/6 J WT mice was performed in males of different age groups: 10-17 weeks and 38–54 weeks, unless otherwise indicated. *Cre* expression in *Mx1-Cre NRAS*^G12D was induced by intraperitoneal (i.p.) injection of two doses (in two consecutive days) of 300 μg of poly-inosine:poly-cytosine (pIpC; Sigma-Aldrich Merck, P9582). Experimental animals were housed with food and water ad libitum, with a 12 h day/night cycle, under specific and opportunistic pathogen free (SOPF) environment at 21.5 ± 1.5 °C and 55 ± 5% humidity at the Section of Comparative Medicine at University of Oslo or at 21 ± 1 °C and 55 ± 10% humidity at the Unit of Comparative Medicine at UiT – The Arctic University of Norway, where experiments were conducted. Ages and sex of animals in each experiment are provided in supplementary Data 2.

### Generation of *Sucnr1-tdTomato* mice
*Sucnr1-tdTomato* reporter mice were generated by gene targeting at the Pluripotent Cell Technology and Transgenic Units of CNIC. Targeting vector was assembled by Gene Bridges GmbH (The recombineering company, Germany). Briefly, a 5′-T2A-tdTomato-3′ followed by bGH polyadenylation signal was placed in front of the stop codon of *Sucnr1*, and an *FRT-Neo-FRT* cassette was inserted downstream. In the construct, right and left homologous arms were 7.0 Kb in length. The backbone of *Sucnr1-tdTomato-FRT-Neo-FRT* targeting vector was eliminated by AscI-MluI digestion and gene targeting performed as previously described[63], using hybrid mouse embryonic stem cells (mESC) 129×C57BL/6. Neomycin-resistant ES cell clones were screened for homologous recombination (HR) by Southern blot of BamHI digested DNA using an external probe 5′ to the left arm. Of 288 clones, 10 were confirmed as positive also for HR at 3′ side, using an external probe 3′ to the right arm and PstI digested DNA. Both probes were amplified by PCR using the following primer sequences: 5′-CAAACC-CACAAAAGACACAGTAAG-3′

and 5′-TTCCTTTTTCAGCCCATCTATATC-3′ (5′probe-BamHI);
5′-TCAAAAACTACCTCATTTCCAAGA-3′
and 5′-TAATCCTGCTATGTTTTCCTCTCC-3′ (3′probe-PstI).

Targeted clones were expanded and injected into C57BL/6 blastocysts, and microinjected embryos were transferred to pseudopregnant females using standard procedures. Germline transmission of *Sucnr1-tdTomato-FRT-Neo-FRT* allele was confirmed with 2 of the targeted clones by PCR using the following primer sequences:

5′-TAGTGAGACGTGCTACTTCCATTT-3′,
and 5′-TTTCAGATCCTAAGTGTCCAAACA-3′.

Then, the Neo resistance cassette was eliminated by breeding with a Flpe deleter line. The mice were backcrossed to C57BL/6 J for more than 10 generations, until a pure genetic background was achieved in the colony.

For colony genotyping we used the following primer sequences:
5′- TATATACACACTGACACGGCCTCT-3′ (**P1**)
and 5′-ATGAACCCATCTTCAGCATATTTT-3′ (**P2**), amplifying a region of size 900 bp for the WT allele;
5′-TAGTGAGACGTGCTACTTCCATTT-3′ (**P3**)
and 5′-TTTCAGATCCTAAGTGTCCAAACA-3′ (**P4**), amplifying a region of size 600 bp for the targeted allele;
5′-GGGGAGGATTGGGAAGACAATA-3′ (**P5**)
and 5′-GGTGGAAAGCTAAAGCCTAGAAATG-3′ (**P6**), amplifying a region of 250 bp for *Sucnr1-tdTomato* allele after deletion of the Neo resistance cassette.

### Mouse hematopoietic cell fraction extraction
Blood samples, bones and spleens were processed as previously described[64]. Briefly, peripheral blood was obtained from the saphenous vein and analyzed with either ABX Pentra XL 80, HORIBA (software v2.2.1), ProCyte Dx Hematology Analyzer, IDEXX (software v:00-34 Build 57) or HT5 element, Antech (software v01.16.00.5281), red

blood cells were afterwards lysed for flow cytometry analysis. Bone marrow hematopoietic cells fraction was obtained from hips, femurs, tibias, humerus, sternum and spine, pooled together from each mouse. Bone marrow nucleated cells (BMNC) were obtained by crushing bones in PBS and filtered on a 40 μm mesh. Spleens were weighed, crushed and filtered on a 40 μm mesh. Red blood cells from blood, BM and spleens were lysed with 0.15 M $NH_4Cl$ (Sigma-Aldrich Merck, A9434) for 10 min at 4 °C. Cells were counted on a CASY cell counter (OMNI Life Science) for subsequent methods.

## Fluorescence-activated cell sorting (FACS)

Immunophenotype of HSPC was defined as LSK. LSK were further defined as LT-HSC (LSK $CD34^-Flt3^-$), ST-HSC (LSK $CD34^+Flt3^-$), and MPP (LSK $CD34^+Flt3^+$). Detailed FACS analysis of the stem and progenitor cell subsets corresponding to HSC and MPP1-MPP6 was performed as previously described[30–32]. Briefly, LT-HSC were further defined as HSC (LT-HSC $CD150^+CD48^-$) and MPP6 (LT-HSC $CD150^-CD48^-$). ST-HSC were further defined as MPP1 (ST-HSC $CD150^+CD48^-$), MPP2 (ST-HSC $CD150^+CD48^+$), MPP3 (ST-HSC $CD150^-CD48^+$) and MPP5 (ST-HSC $CD150^-CD48^-$). MPP were further defined as MPP4 (MPP $CD150^-CD48^+$). $Lin^-$ hematopoietic progenitor subsets were defined as committed CLP; (c-$Kit^{low}Sca-1^{low}CD127^+$), CMP ($Lin^-$c-$Kit^+Sca-1^-$ (LK) $CD34^+FcRγ^-$), MEP (LK $CD34^-FcRγ^-$), and GMP (LK $CD34^+FcRγ^+$). Differentiated hematopoietic cells of $CD11b^+$ myeloid lineage were defined as $CD11b^+Gr1^{hi}F4/80^-$ granulocytes and $CD11b^+Gr1^{hi}F4/80^+$ monocytes, cells of lymphoid lineage were defined as $CD3^+$ T cells and $B220^+$ B cells. In *Mll*-AF9 leukemic mice, GMP-like leukemic blasts were defined as c-$Kit^+FcRγ^+$ and LMPP LSC were identified as c-$Kit^+FcRγ^-CD34^+CD48^+$. For pro-IL-1β intracellular staining, mouse TBM was stained for LSK, then fixed and permeabilized with eBioscience™ Intracellular Fixation & Permeabilization Buffer Set (ThermoFischer, 88-8824-00). Briefly, cells were fixed in 100 μL of intracellular fixation buffer for 40 min at room temperature (RT), then washed and incubated in 100 μL of permeabilization buffer with 4% of anti-IL1β (pro-form) monoclonal antibody (ThermoFischer, 12-7114-82) or corresponding isotype for 40 min at RT protected from light, then washed. Samples were analyzed on a Beckton Dickinson (BD) Canto or LSR Fortessa analyzer with DIVA software v6.1.3, v8.0.1 and v9.0, or a SONY ID7000 spectral flow cytometer with ID7000 software v2.02.17121. For transplant of HSPC from *Sucnr1-tdTomato* donors, cells were sorted on an Aria II in PBS supplemented with 2% heat-inactivated FBS. For colony-forming unit cell (CFU-C) assays, cells were sorted on a FACS Melody (BD) with FACS Chorus software v1.1.2 in PBS supplemented with 2% heat-inactivated fetal bovine serum (FBS; ThermoFisher, 10082147). For molecular biology analyses, cells were FACS-sorted on an Aria II and Aria III cell sorter (BD). For qRT-PCR and ddPCR, cells were sorted in Lysis/Binding Buffer from Dynabeads™ mRNA DIRECT™ Purification Kit (ThermoFischer, 61012) or PicoPure™ RNA Isolation Kit (ThermoFischer, KIT0204). For RNA-sequencing, cells were sorted in extraction buffer from PicoPure™ RNA Isolation Kit. Antibodies used are listed in Supplementary Data 3.

## Imaging flow cytometry

Bone marrow cells from *Sucnr1-tdTomato* mice or C57BL6/J WT littermates were magnetically enriched for $Lin^-$ progenitors using Mouse Hematopoietic Progenitor (Stem) Cell Enrichment Set – DM (Beckton Dickinson, 558451). LSK cells were FACS-sorted on an Aria III (BD) in PBS from $Lin^-$ enriched cells and concentrated in 20 μl of PBS. Samples were visualized and raw data recorded on an Amnis® ImageStream®X Mk II imaging cytometer using INSPIRE® software v200.1.620.0. Single-stained BMNC from C57BL6/J WT mice and non-stained BMNC from *Sucnr1-tdTomato* mice were used for compensation controls. Raw data were processed and analyzed with IDEAS® software version 6.2, single cells were gated using Area and Aspect ratio features followed by gating of Sucnr1-tdTomato⁺ LSK cells.

## Cell culture

Colony-forming unit cell (CFU-C) assay was performed as previously described[65]. Briefly, cultures were performed at 37 °C, 20% $O_2$ and 5% $CO_2$ water-jacketed incubator (Thermo Scientific). $20 \times 10^3$ and $200 \times 10^3$ cells were plated in duplicate in MethoCult™ GF M3434 (Stem Cell Technologies, 03434) for BM and spleen, respectively. CFU-C were counted once a week for 3 weeks, replating the first 2 weeks. For replating, CFU-C were resuspended in PBS at RT, methylcellulose was washed twice with PBS, and the cells were then seeded in fresh Methocult™ GF M3434. For in vitro treatment, succinic acid (Sigma-Aldrich Merck, S9512) or monomethyl succinate (Sigma-Aldrich Merck, M81101) were added to the methylcellulose at 0.05, 2, or 10 mM at each plating. Serial CFU-C assays from 300 FACS-sorted HSC from magnetically enriched bone marrow $Lin^-$ progenitors were performed as described previously[66], colonies were scored and 10,000 cells replated after 7, 12 and 17 days, 10 mM of monomethyl succinate or cis-epoxy succinate (Glixx Laboratories, GLXC-22816) or 25 μM tasquinimod (MedChemExpress, HY-10528) were added at each plating. For cell lines, Kasumi-1 (provided by Dr Paloma Garcia, DSMZ, ACC 220, authentication from supplier) and NOMO-1 (DSMZ, ACC 542, authentication from supplier) were seeded at $0.5 \times 10^6$ cells per ml in RPMI medium without phenol red (ThermoFisher, 11835030) supplemented with 10% heat-inactivated FBS and 1% Penicillin-Streptomycin-Glutamine (ThermoFischer, 10378016). Cells were passed every 2–3 days. $0.5 \times 10^6$ Kasumi-1 and NOMO-1 cells were treated for 48 h with 1.77 μM of cytarabine (AraC, MedChemExpress, HY-13605) or vehicle together with 10 mM of cis-epoxy succinate or vehicle. Kasumi-1 cells transduced with a lentivector overexpressing *SUCNR1* or by an empty vector were treated with 10 mM of cis-epoxy succinate or vehicle without AraC. After 48 h, cells were washed and pelleted for apoptosis assay.

## Lentivirus production and transduction

Lentiviruses encoding SUCNR1-GFP or empty vector (EV-GFP) were produced by transfecting HEK293FT (ThermoFischer, R70007, authentication from supplier) cells cultured in DMEM (ThermoFischer, 41966029) containing 1% Penicillin/Streptomycin (ThermoFischer, 15140122) and 10% FBS with VSV-G, psPAX2, and expression constructs designed and produced using VectorBuilder at a 1:2:2 ratio, using Lipofectamin 2000 (ThermoFischer, 11668019). SUCNR1-GFP construct was pLV[Exp]-Hygro-EF1A > hSUCNR1[NM_033050.6](ns):P2A:EGFP and EV-GFP construct was pLV[Exp]-Hygro-EF1A > EGFP. After 8 h, the cells were washed and cultured in Kasumi-1 medium. Virus-containing supernatant was harvested at 48 h, filtered through a 0.45 μm membrane, and used for transduction. Kasumi-1 cells were transduced by spinoculation; $1 \times 10^6$ cells in 2 ml were mixed with 2 ml of virus with polybrene (Sigma-Aldrich, H9268) at 8 μg per ml and centrifuged at 1000 g for 90 min at RT, then incubated overnight. Medium was refreshed the next day. Transduced cells were expanded and sorted with a BD FACS Melody for GFP expression after 2–3 passages. EV-GFP and SUCNR1-GFP vector summaries are provided in Supplementary Data 4.

## Cell cycle and apoptosis assays

For cell cycle analysis, freshly isolated total bone marrow cells were magnetically enriched for $Lin^-$ population. Cells were resuspended at $2 \times 10^6$ cells per ml of Hank's Balanced Salt Solution (HBSS; ThermoFisher, 14175095) supplemented with 10 % heat-inactivated FBS, 20 mM HEPES (ThermoFisher, 15630056), 50 μM Verapamil (Sigma-Aldrich, V4629) and 5 μg of Hoescht 33342 (Sigma-Aldrich Merck, 14533) per ml. After 45 min of incubation at 37 °C under orbital agitation, Pyronin Y (Sigma-Aldrich Merck, 83200) was added at 1 μg per ml and cells were incubated for 45 additional min at 37 °C under agitation. Analysis of apoptotic cells was performed as previously described[64] through Pacific Blue™ Annexin V/SYTOX™ AADvanced™ Apoptosis Kit

for flow cytometry (ThermoFischer, A35136), using BMNC to study apoptosis in LSK cells, or in AML cell lines after in vitro treatment with SUCNR1 agonist and/or cytarabine. Briefly, BMNC were stained for LSK, washed and incubated in 1X Annexin binding buffer (Thermo-Fisher, V13246) with Pacific Blue-conjugated anti-Annexin V antibody (ThermoFisher, A35122) and SYTOX™ AADvanced™ Dead Cell Stain Kit (ThermoFisher, S10274) for 15 min at RT, before flow cytometry analysis, according to the manufacturer instructions.

## Succinate measurements

FACS-sorted LSK and $CD11b^+Gr-1^+F4/80^+$ monocytes were analyzed using Succinate colorimetric assay kit (Sigma-Aldrich Merck, MAK 184-1KT) according to manufacturer instructions. Intracellular succinate measurement in HSC was done by liquid chromatography-coupled mass spectrometry (LC-MS) after FACS-sorting of 5000–10000 $Lin^-Sca-1^+c-Kit^+CD150^+CD48^-$ or 5000 $Lin^-Sca-1^+c-Kit^+CD34^-Flt3^-CD150^+CD48^-$ HSC after $c-Kit^+$ magnetic enrichment using CD117 microbeads (Miltenyi biotec, 130-091-224), directly into acetonitrile (Sigma-Aldrich, 1000291000) as previously described[67]. Briefly, for LC-MS cell lysates were centrifuged at 17,000 $g$ for 30 min at −9 °C, followed by transfer of supernatants to auto-sampler vials (SureSTART, ThermoFisher Scientific). The supernatants were analyzed on a Vanquish™ Neo UHPLC (ThermoFisher Scientific) with SeQuant ZIC-HILIC™ trap column (0.3×5 mm, 5 µm particle size; Merck, 1.50492) and SeQuant ZIC-HILIC™ capillary separation column (0.3 × 150 mm, 3.5 µm particle size; Merck, 1.50479). We injected 5 µl of sample, blank or succinic acid standard (Sigma-Aldrich, S9512) onto the trap column in 30 s, followed by gradient elution. HILIC gradient mobile phase A was 5% acetonitrile and 95% water containing 5 mM ammonium acetate; mobile phase B was 90% acetonitrile and 10% water. To elute nonpolar compounds, mobile phase B was kept at 100% from 0 to 6.5 min, then it was ramped linearly from 100 to 42.5% from 6.5 to 26.5 min, and decreased to 10% at 27.5 min. To remove salt from the column, mobile phase B was held at 10% from 27.5 to 37.5 min B and increased back to 100% at 37.5 min. From 37.5 to 41.5 min, mobile phase B was held at 100% to regenerate the water layer of the column. Solvent flow rate was set at 5 µl per min for the gradient and at 4 µl per min for the salt elution phase (waste). The temperature of the column was maintained at 45 °C. MS data were acquired on an Orbitrap Exploris™ 240 mass spectrometer (ThermoFisher Scientific) equipped with Easy-Spray™ source (ThermoFisher Scientific, ES081) with capillary emitter (ThermoFisher Scientific, ES994) in full scan MS mode with control software version 4.4. Negative polarity was acquired at a resolution of 60,000 (at m/z = 200); automatic gain control target and maximum injection time were set by default; microscans and scan range were set to 2 and 80–800 Daltons, respectively. MS source parameters were 2300 V (negative ion mode) for capillary voltage and 320 °C for the temperature of the ion transfer tube. Raw data were loaded into Thermo XCalibur Qual Browser and the total ion chromatogram (TIC) and succinic acid peak (m/z 117.0194 with 5ppm window, retention time 2.7–3.3 min; identity confirmed by standard) were integrated. Succinic acid peak areas were normalized to the TIC area and batch corrected.

Succinate was measured in BMEF by colorimetric assay with normalization to the amount of protein measured by Pierce™ BCA Protein Assay Kit (ThermoFischer, 23225); and by nuclear magnetic resonance (NMR) as previously described[68]. For the latter, BMEF samples were prepared by mixing 1:1 with phosphate buffer prepared in $D_2O$, containing the internal reference TMSP ((3-trimethylsilyl)propionic-(2,2,3,3-d4)-acid sodium salt) and $NaN_3$ to yield final concentrations of 0.1 M phosphate, 0.5 mM TMSP and 1.5 mM $NaN_3$ (all from Sigma-Aldrich Merck). Samples were then transferred into 3 mm NMR tubes and measured on a 600 MHz Bruker spectrometer equipped with an Avance-III console and a 5 mm triple resonance cryoprobe (TCI) z-axis pulsed field gradient (PFG) cryogenic probe, a cooled SampleJet autosampler (Bruker), and automated tuning and matching. Spectra were acquired at 300 K with a 1D ${}^1$H-NOESY (Nuclear Overhauser effect spectroscopy) pulse sequence with pre-saturation water suppression (noesygppr1d, Bruker standard pulse program), with a spectral width of 12 ppm, 32768 data points, and an interscan delay of 4 s. The ${}^1$H carrier frequency was set on water resonance, and the ${}^1$H 90 ° pulse was calibrated at a power of 0.256 W with a typical length of about 7–8 µs. A total of 64 transients and 8 steady state scans were acquired, resulting in a total experimental time of 7.5 min. Spectral processing was performed using MetaboLab[69] software v2023.0529. Free induction decays were apodized with a 0.3 Hz exponential line-broadening function and zero-filled to 131072 points prior to Fourier transformation. Chemical shifts were referenced to the TMSP signal at 0 ppm, and spectra were manually phase corrected. A spline baseline correction was applied, and the water and spectral edge regions were excluded prior to probabilistic quotient normalization (PQN). Succinate was assigned using Chenomx NMR Suite (v7.0, Chenomx Inc.) in conjunction with the Human Metabolome Database (HMDB). Succinate intensities were obtained directly from the spectra. Succinate intensities were normalized to the reference (TMSP) intensity, total spectral area of the sample and BMEF volume obtained.

## Cell energy phenotype

Freshly isolated total bone marrow cells from Sucnr1-KO and C57BL/6 J WT males aged 26–52 weeks were magnetically enriched for $Lin^-$ progenitors and $0.5 × 10^6$ $Lin^-$ cells were immobilized in triplicates to XF96 microplates (Agilent, 101085-004) pretreated with Cell-Tak (Corning; 354240). Measurement of intact cellular respiration was performed using the Seahorse XFe 96 analyzer (Agilent) with Wave controller software controller 2.4.2 and the XF Cell Energy Phenotype Test Kit (Agilent, Cat. 103325-100) according to manufacturer instructions. Medium assay consisted of XF Base Medium (Agilent, 102353-100) supplemented with 10 mM glucose, 1 mM pyruvate and 2 mM glutamine. Respiration was measured under basal conditions, and in response to a stressor mix of Oligomycin (ATP synthase inhibitor; 0.3 µM) and the electron transport chain accelerator ionophore Trifluorocarbonylcyanide Phenylhydrazone (FCCP; 0.15 µM).

## Reactive oxygen species

Quantification of reactive oxygen species (ROS) was performed as previously described[70]. Briefly, $10x10^6$ BMNC were stained for ROS levels (superoxide anion) with 5 µM hydroethidine (DHE; Thermo-Fisher D23107) for cellular ROS or 10 µM MitoSOX (ThermoFisher, M36008) for mitochondrial ROS, by 20 min incubation at 37 °C. Cells were then washed once with PBS and stained for LSK. Fluorescence of oxidized DHE or MitoSOX was determined by flow cytometry. Antibodies used are listed in Supplementary Data 3.

## RNA isolation, reverse transcription and droplet digital PCR

RNA was isolated using Tri Reagent (Sigma-Aldrich Merck, T9424) or RNeasy Plus Mini Kit (QIAGEN, 74134) according to manufacturer protocol for mouse BMNC, with Dynabeads™ mRNA DIRECT™ Purification Kit (ThermoFischer, 61012) for ddPCR, or using PicoPure™ RNA Isolation Kit (ThermoFischer, KIT0204) for RNA-seq on FACS-sorted mouse LSK cells. Reverse transcription was performed using the High-Capacity cDNA Reverse Transcription Kit (ThermoFischer, 4368814), following the manufacturer recommendations. ddPCR was performed with ddPCR™ Supermix for Probes (No dUTP) (Bio-Rad 186-3023) according to the manufacturer instruction, using custom-designed probes. Droplets were generated and read on a QX200 Droplet Digital™ PCR Systems with QuantaSoft software v1.7.4. The expression level of each gene was calculated by absolute quantification of copies. All values in mice were normalized with mouse *Gapdh* as endogenous housekeeping gene. Probes used are listed in Supplementary Data 3.

## Cytokine analyses

BMEF was obtained as previously described[64] and used for cytokine analyses[64]. Briefly, BMEF was isolated from one femur and one tibia of each mouse by flushing both bones together with 150 μL of PBS. The cell suspension was then centrifuged 10 min at 15,000 g and the supernatant collected. Multiplex immunoassays were performed using Bio-Plex Pro Mouse Cytokine 23-plex Assay kit (Bio-Rad, M60009RDPD) and Bio-Plex Mouse Cytokine IL-6 set (Bio-Rad, 171G5007M) on a Bio-Plex 200 System (Bio-Rad) with Bio-Plex Manager v6.2 software. Proteins were quantified with Pierce™ BCA Protein Assay Kit (ThermoFisher, 23225) using a CLARIOStar plate reader with CLARIOStar plus v6.20 software. Results were expressed as pg per mg of protein. S100A8/S100A9 were quantified by sandwich ELISA as previously described[71,72]. The ELISA system detects both active heterodimers and inactive tetramers formed by S100A8 and S100A9. Increased levels of S100A8/A9 complexes represent biological active S100A8/A9 heterodimers[73]. Briefly, 20 μl of BMEF were diluted 1/100, 1/200, and 1/400 and measured in duplicate. If all values coincided after recalculation, the value was accepted. If needed, another dilution range was run in the linear 1–30 ng per ml range to allow calculation of exact S100A8/A9 levels. Antibodies used for ELISA were prepared as previously described[74]. Each plate included a reference sample as positive control. Total protein amount was measured with Pierce™ Coomassie (Bradford) Protein Assay kit (ThermoFisher, 23200). Results were expressed as ng per μg of protein.

## In vivo pharmacological treatments

For the mobilization study, C57BL6/J mice aged 51 weeks were injected with 2 mg per kg of succinic acid and killed 1 h or 6 h after to study CFU-C proportion in BM, SP and PB. For S100A9 inhibitor tasquinimod, Sucnr1-KO male mice aged 28–41 weeks or C57BL6/J WT mice aged 26-39 weeks were administered with vehicle or tasquinimod at 30 mg per kg per day in drinking water for 9–16 weeks as previously described[41]. Briefly, tasquinimod was dissolved in DMSO, the required volume of tasquinimod solution was added to the bottle, then top up with DMSO to reach a 2% DMSO concentration in the final volume of drinking solution. The drinking solution was completed by adding polyethylene glycol 300 (PEG 300, Sigma-Aldrich Merck, 8.07484) for a final concentration of 2%, then drinking water containing 3% sucrose. The vehicle for tasquinimod was 2% DMSO, 2% PEG 300 and water containing 3% sucrose. Water bottles were changed twice per week. Effect of acute succinic acid administration in human AML xenografts was studied through short-term daily i.p. injections of vehicle (saline) or succinic acid at 2 mg per kg neutralized with NaOH to pH 7.3, for 10 days. Effect of chronic succinic acid administration in leukemia models was evaluated by long-term i.p. injections of succinic acid. C57BL6/J WT female mice transplanted with BMNC from leukemic *Mll-AF9* mice, were treated with vehicle or incrementing doses of succinic acid at 2, 20, 100, 200 and 400 mg per kg daily over intervals of 2–4 weeks, then at 400 mg per kg 3 times a week on alternate days for 8 additional weeks. The treatment started 4 weeks after the transplant. The total treatment duration was 42 weeks. For tasquinimod and cis-epoxy succinate in recipients of MLL-AF9+ bone marrow, treatment started 5 weeks after transplantation and mice were euthanized and used for endpoint analyses at the maximum period of 24 weeks of treatment or before, when they exhibited signs of frailty according to humane endpoints or reached a threshold of 100×10[6] WBC per ml in peripheral blood. Mice were treated with vehicle or tasquinimod at 30 mg per kg per day in drinking water for 17 weeks, then 20 mg per kg per day in drinking water for 7 weeks, as previously described[41]. Mice received daily i.p. injection of vehicle or 2 mg per kg of cis-epoxy succinate for a maximum of 24 weeks. 22–37 weeks old *Mx1-Cre NRAS^{G12D}* male mice were treated daily with vehicle or incrementing doses of succinic acid at 100, 200, 400, 800 and 1200 mg per kg over intervals of 2–4 weeks, starting 7 weeks after pIpC induction. The total treatment duration was 27 weeks.

## Transplantation assays

For human xenografts, myeloablation of 8–12-week-old female NSG mice was performed through i.p. injection of Busulfan (Busilvex, Pierre Fabre Pharmaceuticals, 130309) at 25 mg per kg. Mice were transplanted 24 h later with 2×10[6] CD34+ cells isolated from the BM of one AML patient. Animals showing human CD33 engraftment <0.1% in BM were considered negative and excluded from the study. *Mll-AF9* knock-in mice BM transplantation was performed through the tail vein after myeloablation of 12-week-old female mice for succinic acid treatment or 10-week-old male mice for tasquinimod and cis-epoxy succinate treatment, whole body irradiated with 9 Gy (in 2 doses separated by 3 h) using an X-RAY source (Rad Source's RS 2000 or Xstrahl RS 320, respectively). 4 h after, mice were transplanted intravenously (i.v.) with 2×10[6] BMNC. To study the function of Sucnr1 through the hematopoietic compartment, we transplanted 2×10[6] BMNC from *Mx1-Cre Sucnr1^{fl/fl}* or *Sucnr1^{fl/fl}* mice into 15-week-old male C57BL/6 J recipients, previously irradiated as described (Rad Source's RS 2000). Expression of Cre recombinase was induced by i.p. injection of two doses (in two consecutive days) of 300 μg of pIpC 4 weeks after the transplant. For competitive transplants, 16-week-old Ly5.1 C57BL6/J mice were whole body irradiated (Xstrahl RS 320) and injected i.v. in the tail vein with 1×10[6] BMNC from either Sucnr1-KO or C57BL6/J Ly5.2 mice aged 42–43 weeks, together with 1×10[6] BMNC from 16-week-old Ly5.1 C57BL6/J mice. For transplants of HSPC from *Sucnr1-tdTomato* donors, 8-week-old Ly5.1 C57BL6/J were whole body irradiated (Xstrahl RS 320) and injected in the tail vein with 200 Ly5.2 Sucnr1-tdTomato+ or Sucnr1-tdTomato− FACS-sorted mix of HSC, MPP3 and MPP4 (40 HSC, 86 MPP3 and 74 MPP4), together with 2.5×10[5] BMNC from Ly5.1 C57BL6/J mice.

## Human CD34+ cell enrichment

Peripheral blood mononuclear cells (PBMC) and BM nucleated cells from one AML patient and PBMC from one healthy donor, were obtained by density gradient centrifugation (Lympholyte, Cedarlane, CL5015) and used for CD34+ cell enrichment using immune magnetic technology (Stem Cell Technologies, 15086).

## RNA sequencing and bioinformatic data analysis

For calculation of survival probability, normalized gene expression was retrieved from GEO GSE14468 as previously described[75]. Briefly, probe "223939_at" specific to the *SUCNR1* transcript, NM_033050, was used in the analysis of expression for survival analysis of 426 AML patients, containing 224 M0-M3 and 171 M4-M5 AML patients, 216 females (50.7%) and 210 males (49.3%), aged 19–61 years old. Survival data were gently provided by contributor author Dr. P. J. M. Valk (Department of Hematology, Erasmus University Medical Center, Rotterdam, The Netherlands). The median expression of *SUCNR1* for the whole cohort (n = 426) was used to dichotomize its expression into High and Low levels. Then, Kaplan–Meier curves were produced for the overall and progression-free survival of the individuals. Log-rank p-value is reported. Two-tailed Wald test without Benjamini-Hochberg correction for multiple comparisons was used for Cox regression of human survival. *SUCNR1* expression in healthy individuals and AML patients were obtained in BloodSpot and retrieved from GSE42519 and GSE13159, respectively.

RNA-Seq data from FACS-sorted LSK cells obtained from the BM of Sucnr1-KO and C57BL/6 J WT male mice aged 27–30 weeks have accession number GSE215255. RNA amplification and RNA-Seq library production were performed at the Genomics Support Centre Tromsø (GSCT, UiT – The Arctic University of Norway). Total RNA was isolated using the Arcturus PicoPure™ RNA Isolation Kit (ThermoFischer, KIT0204) from small numbers of FACS-sorted cells (27000-75000).

RNA was amplified with the NEBNext Single cell/Low input RNA libray kit v. 3.0 (NEB#E6420) following the protocol to the fragmentation step. cDNA was fragmented to target size of 150 bp with Covaris M220. Libraries were prepared with TruSeq RNA library preparation kit v. 2.0 (RS#122-2001, Illumina) starting the protocol from "perform end repair" to final library. The quantity of the final library was determined by dsDNA - Qubit High Sensitivity and fragment size was determined by Agilent Technologies 2100 Bioanalyzer using the Agilent DNA 1000 chip. Libraries were sequenced with the NextSeq550 instrument (Illumina) with System Suite software v4.0 at the GSCT (UiT – The Arctic University of Norway). At least 3 biological replicates were used per experimental group (4 WT, 3 Sucnr1-KO).

Sequencing reads were pre-processed by means of a pipeline that used FastQC (http://www.bioinformatics.babraham.ac.uk/projects/fastqc/) with software v0.11.9, to assess read quality, and Cutadapt[76] v5.2 to eliminate Illumina adaptor remains, to trim the first 6 nucleotides, and to discard reads that were shorter than 30 bp after trimming. Resulting reads were mapped against reference transcriptome GRCm38.99 and quantified using RSEM[77]. Expected expression counts calculated with RSEM were then processed with an analysis pipeline that used the Bioconductor package Limma[78] v3.17 for normalization (using TMM method) and differential expression testing, for contrast Sucnr1-KO versus C57BL/6 J WT mice. The test focused on 15224 genes expressed with at least 1 count per million (CPM) in at least three samples, and used a random variable to define groups of samples that belonged to the same batch (date of FACS-sorting). Changes in gene expression were considered significant if associated to Benjamini and Hochberg adjusted $p < 0.05$. 4585 differentially expressed genes were detected. Functional enrichment analyses were performed with Ingenuity Pathway Analysis (IPA) from QIAGEN Inc. (https://www.qiagenbioinformatics.com/products/ingenuity-pathway-analysis) and GSEA[79]. IPA was performed on 3841 differentially expressed genes with Benjamini-Hochberg adjusted $p < 0.05$ and absolute $\log_2 FC > 1$. GSEA was performed against two different gene set collections from the Molecular Signatures Database (MSigDB)[80]: Chemical and Genetic Perturbations (CGP) and Gene Ontology Biological Processes (GOBP). FDR q values < 0.25 were used to filter enrichment results. Additional graphical representations (bar plots, volcano plots) were generated with R.

The scRNA-seq data from GSE178912 were aligned and quantified using the Cell Ranger Single-Cell toolkit (v.3.0.2) against the GRCh38 human reference genome. The pre-filtered output generated by Cell Ranger was used for all downstream analyses. Count matrices were imported into the R environment (v4.4.2) using the Seurat package (v5.3.0), and all visualizations were generated with ggplot2 (v4.0.0). A Seurat object was created using the CreateSeuratObject function with the min.cells parameter. Low-quality cells were removed based on quality control thresholds: fewer than 200 or more than 6000 detected genes, and a mitochondrial RNA content exceeding 15%. On average, 21650 cells were retained, with a median of 19065 detected genes per cell across all conditions (CTL and ARAC D8) from TUH07 and TUH69 PDX samples (GSE178912).

Data normalization was performed using the NormalizeData function with a scaling factor of 10,000 and logarithmic transformation. The 2000 most highly variable genes (HVGs) were identified using the FindVariableFeatures function with the selection method set to Variance Stabilizing Transformation (vst). The merged dataset was then scaled with ScaleData. Principal component analysis (PCA) was performed using the RunPCA function (Seurat) with 30 principal components (npcs=30), followed by batch correction and integration using the Harmony algorithm.

Dimensionality reduction was conducted with Uniform Manifold Approximation and Projection (UMAP) using RunUMAP with dims=1:30. A K-nearest neighbor (KNN) graph was constructed using FindNeighbors (dims=1:20), and clustering was performed using the

Louvain algorithm implemented in Seurat via FindClusters, with a resolution parameter of 0.1.

Gene signature scores were computed using the AddModuleScore function from Seurat and visualized through UMAP and violin plots. Functional enrichment analysis was performed with RunEnrichment function of the single-cell pipeline (scp) package v0.5.6. Source data from analysis of GSE178912 are provided as Supplementary Data 1.

## Statistics and reproducibility

Statistical analyses were performed using Prism 9 software (GraphPad). Data are biologically independent human samples or animals, and means ± SEM for bar plots or medians for violin plots. Box plots show median as center line; upper and lower quartiles as box limits; 1.5x interquartile range as whiskers. All biologically independent data points are shown unless otherwise indicated for better visualization. A p value of less than 0.05 was considered significant. Significant p values are reported in the figures and non-significant p values are reported in the Source Data file. Statistical significance was evaluated by unpaired two-tailed t-test or two-tailed Mann–Whitney U test where appropriate. Survival curve in mice was analyzed by log-rank (Mantel-Cox) test. Two-tailed Wald test with or without Benjamini-Hochberg correction for multiple comparisons was used for gene expression in bulk RNA-seq and Cox regression of human survival, respectively. For scRNA-seq data, the 2000 most HVGs were identified using the FindVariableFeatures function with the selection method set to vst. PCA was performed using the RunPCA function (Seurat) with 30 principal components (npcs=30), followed by batch correction and integration using the Harmony algorithm. Gene signature scores were computed using the AddModuleScore function from Seurat and visualized through UMAP and violin plots. Functional enrichment analysis was performed with RunEnrichment function of the scp package v0.5.6. The differential gene expression was analyzed using two-tailed t-test.

In most experiments and unless otherwise indicated, mice of the same sex and approximate similar age were used to control for covariates. Mice were randomized to treatment groups. The investigators were not blinded to allocation during experiments and outcome assessment. Animals that showed symptoms of disease or health issues unrelated to aberrant myelopoiesis were excluded from the study, i.e., obesity, tumor masses, etc. Criteria applied for mouse termination before the established endpoint were in accordance with the Norwegian Food and Safety Authority. Outliers in mouse studies were excluded using Grubbs test, and ROUT test was used when several outliers were suspected. For in vivo experiments, sample size was calculated based on estimation of the minimum number of animals required to obtain biologically meaningful results. Where appropriate, experimental group size was increased by adding results from different mice of similar characteristics analyzed on different days based on availability of new experimental and wild-type or control mice. Most in vivo experiments were performed over two or more days, to ensure replication of results. Experiments replicated independently confirmed previous results. Cohort size and numbers of cells transplanted in xenografts were informed by the total number of available cells. Animals showing human CD33 engraftment <0.1 % in BM were considered negative and excluded from the study. Data of ex vivo cultures, cytokine analyses, molecular biology assays, from each mouse are the mean result of at least two or three technical replicates, and most of these assays were performed over two or more days with samples from different mice, to ensure replication of results.

## Reporting summary

Further information on research design is available in the Nature Portfolio Reporting Summary linked to this article.

## Data availability

All resource and reagents are listed in Supplementary Data 3–4. All data generated in this study are provided in the article file, Supplementary Information, Supplementary Data 1 and Source Data files. RNA-seq data from FACS-sorted LSK obtained from the BM of 27–30-week-old Sucnr1-KO and C57BL/6 J WT male mice have been deposited in the GEO database under the accession code GSE215255: [https://www.ncbi.nlm.nih.gov/geo/query/acc.cgi?&acc=GSE215255]. Public datasets and databases used in this study in addition to the ones generated are the following: GSE14468 GSE178912 GSE42519 GSE13159 GSE17054 GSE19599 GSE11864 E-MEXP-1242 Source data are provided with this paper.

## Code availability

The code to retrieve gene expression data in GSE14468 and reproduce the multiple survival analyses presented in this manuscript have been adapted from (https://github.com/SAC-lab-UiT).

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

## Acknowledgements

We thank K. Tasken, J. Saarela and the NCMM at the University of Oslo (UiO), and S. Kanse (UiO), for access to facilities. We acknowledge L.M. Gonzalez, C. Armengol, Y. Chen, F. Ferrer, P. Utnes, E. Gargiulo, M. Ristic (MSCA-IF-EF-ST, 101031817) and other present and past members of L. Arranz group, M. Egg, N.E. Bastani, C.G. Fenton, L.D. Håland, J. Landskron, UiT Advanced Microscopy Core Facility, Flow Cytometry Core Facility at Oslo University Hospital, UiO and UiT Comparative Medicine Units, for assistance. The Pluripotent Cell Technology unit, and L.M. Criado and the Transgenesis unit, at CNIC for the generation of *Sucnr1-tdTomato* reporter mice. A. Cervera for kindly providing *Sucnr1⁻/⁻* and *Sucnr1ᶠˡ/ᶠˡ* mice. N. Cabezas-Wallscheid for helpful discussions. We would also like to thank the AML patient and healthy donor who donated biological samples. Our work was supported by Centre for Embryology and Healthy Development from the Research Council of Norway (Norwegian Centres of Excellence scheme, 332713), Young Research Talent grants from the Research Council of Norway (Stem Cell Program, 247596; FRIPRO Program, 250901), the Norwegian Cancer Society (6765150), the Northern Norway Regional Health Authority (HNF1338-17) and a joint grant from the Northern Norway Regional Health Authority, the University Hospital of Northern Norway (UNN) and UiT (Strategisk-HN06-14) to LA. NVL was supported by the EU grant HaemMetabolome H2020-MSCA-ITN-2015–675790. NVL and PG acknowledge the Wellcome Trust for supporting access to NMR instruments at the Henry Wellcome Building for Biomolecular NMR in Birmingham (208400/Z/17/Z). LS-MS measurements were supported by Fund for Strategic Fundamental Research of the Walloon Region (WELBIO-CR-2022 S-08) and Belgian Foundation Against Cancer (ATE-2022/1863) to NG.

## Author contributions

Conceptualization: L.A. Data curation: V.C., E.B., M.J.G., C.T., and L.A. Formal analysis: V.C., E.B., M.J.G., C.T., F.S.C., and L.A. Funding acquisition: L.A. Investigation: V.C., E.B., M.H., J.K., A.B., F.G., N.V.L., M.F., A Villatoro, H.T., R.H.P., T.V., N.G., P.G., J.E.S., and L.A. Methodology: V.C., D.P.P., G.G., and L.A. Project administration: L.A. Resources: G.G., C.A.H., A Vik, and L.A. Supervision: L.A. Visualization: V.C, E.B., J.K., C.T., and L.A. Writing—original draft: V.C. and L.A. Writing—review & editing: All authors.

## Competing interests

The authors declare no competing interests.

## Additional information

¹Stem Cells, Ageing and Cancer Research Group, Centre for Embryology and Healthy Development, Institute of Clinical Medicine, Faculty of Medicine, University of Oslo, Oslo, Norway. ²Stem Cells, Ageing and Cancer Research Group, Department of Microbiology, Oslo University Hospital, Oslo, Norway. ³Stem Cells, Ageing and Cancer Research Group, Department of Medical Biology, Faculty of Health Sciences, MH2 building level 10, UiT – The Arctic University of Norway, Tromsø, Norway. ⁴Centre de Recherches en Cancérologie de Toulouse, Université de Toulouse, Inserm U1037, Toulouse, France. ⁵Bioinformatics Unit, Centro Nacional de Investigaciones Cardiovasculares Carlos III (CNIC), Madrid, Spain. ⁶Institute of Cancer and Genomic Sciences, University of Birmingham, Birmingham, UK. ⁷Targeting tumors of central nervous system Project Group, Centre of Embryology and Healthy Development, Institute of Clinical Medicine, Faculty of Medicine, University of Oslo, Oslo, Norway. ⁸Genomics Support Centre Tromsø, Faculty of Health Sciences, UiT – The Arctic University of Norway, MH building level 9, Tromsø, Norway. ⁹Clinical Bioinformatics Research Group, Department of Clinical Medicine, Faculty of Health Sciences, UiT – The Arctic University of Norway, MH building level 9, Tromsø, Norway. ¹⁰Institute of Immunology, University of Münster, Münster, Germany. ¹¹Institute for Molecular Medicine Finland, FIMM, Helsinki Institute of Life Science, iCAN Digital Precision Cancer Medicine Flagship, University of Helsinki, Helsinki, Finland. ¹²Department of Hematology, University Hospital of North Norway, Tromsø, Norway. ¹³Department of Clinical Medicine, Faculty of Health Sciences, UiT – The Arctic University of Norway, MH2 building level 10, Tromsø, Norway. ¹⁴Pluripotent Cell Technology Unit, CNIC, Madrid, Spain. ¹⁵de Duve Institute, Catholic University of Louvain, Brussels, Belgium. ¹⁶Associate Investigator, Centre for Molecular Medicine Norway (NCMM), University of Oslo, Oslo, Norway. ¹⁷These authors contributed equally: Marcel Heugel, Joanna Konieczny. ✉e-mail: lorena.arranz@medisin.uio.no

