## [Transparent Peer Review file · Nature Communications]

Succinate receptor 1 restricts hematopoiesis and prevents acute myeloid leukemia progression

Corresponding Author: Dr Lorena Arranz

Version 0:

Reviewer comments:

Reviewer #1

(Remarks to the Author)

The article presents a clear and well-structured exploration of *Sucnr1*'s role in hematopoiesis, offering valuable insights into a complex and underexplored area of stem cell biology. The use of comprehensive experimental approaches—ranging from genetic manipulations to transcriptomic analysis—enhances the clarity of the conclusions and strengthens the overall impact of the study. The figures are well-organized and effectively support the narrative, providing clear visual representations of the data. Overall, the manuscript is excellently presented, with a clear focus on the novelty of the findings and their potential implications in the context of hematopoiesis. However, the manuscript would benefit by addressing certain key issues.

1) The global knockout model used may limit the interpretation of *Sucnr1*'s role in hematopoiesis, as indirect effects arising from systemic *Sucnr1* deficiency could influence hematopoietic stem and progenitor cell (HSPC) responses. To clarify whether *Sucnr1*'s effects are intrinsic to hematopoietic cells or arise from changes in the microenvironment, a lineage-specific knockout approach would be insightful. This strategy could help differentiate between direct roles of *Sucnr1* within hematopoietic cells and broader physiological or paracrine influences, thereby enhancing our understanding of its role in self-renewal and differentiation dynamics in hematopoiesis.

2) The effects of *Sucnr1* deficiency on hematopoiesis appear contradictory. *Sucnr1* deletion results in both an expansion of HSPCs and increased hematopoietic activity in the bone marrow and spleen, yet, paradoxically, it also shows reduced transcriptomic markers of stemness and self-renewal. Could the authors provide a more in-depth discussion to explain this finding? Additionally, targeted experiments investigating differential effects across specific HSPC subpopulations (e.g., long-term vs. short-term HSCs, or lineage-specific progenitors) could help clarify whether *Sucnr1*'s impact varies across these subsets, potentially reconciling the observed expansion with the reduction in stemness markers.

3) Section: Low SUCNR1 Predicts Acute Myeloid Leukemia Progression

In lines 141-143, it is mentioned that "MLL-AF9+ mice had overt AML, and the lineage-negative compartment was expanded in the bone marrow of succinic acid-treated mice compared to vehicle-treated controls (Figure 1L), showing an increase in the LK fraction but no changes in LSK cells (Supplemental Figure 2E)." While there was an observed trend toward an increase in both the LK fraction and LSK cells in succinic acid-treated mice, it is crucial to emphasize that none of these differences reached statistical significance. Adding this clarification in the text would provide a more accurate and nuanced interpretation of the data.

In line 143, the statement that "low SUCNR1 represents a prognostic marker for reduced survival in AML patients" would benefit from further statistical validation to strengthen its impact. To rigorously establish SUCNR1 as an independent prognostic factor, a Cox regression analysis should be performed. This would allow the authors to assess the effect of SUCNR1 on survival while adjusting for other known prognostic variables in AML. This addition would significantly enhance the robustness of the study's conclusions regarding the prognostic value of SUCNR1 in AML.

4) Section: *Sucnr1* is expressed in HSPC, and its activation restricts their functional response to succinate ex vivo

In line 223, the authors suggest that *Sucnr1* activation suppresses succinate's intracellular effects, supported by their observations with monomethyl succinate, which increased colony-forming potential in total bone marrow (TBM) in a dose-dependent manner. However, this conclusion is not entirely convincing. If *Sucnr1* activation indeed dampens succinate's effects, one would expect to see a reduction in CFU-C in TBM with succinate treatment (Figure 4A), but no such decrease is

observed. This raises questions about the robustness of the authors' claim. Additionally, to strengthen their argument, the authors could have tested the combination of monomethyl succinate with succinic acid to explore potential synergistic effects, which would provide more clarity on their hypothesis.

Additionally, clarification is needed on why monomethyl succinate increased CFU-C in FACS-sorted HSC from *Sucnr1*-KO mice in the first plating (Figure 4C), while this effect was not seen in cells from WT mice (Figure 4B). Could the difference be attributed to varying intracellular succinate levels between the two mouse strains? Has the possibility of altered succinate accumulation been considered? Without addressing this, the interpretation of the results remains incomplete and raises doubts about the underlying mechanisms.

5) Section: *Sucnr1* restricts HSPC and hematopoiesis via control of S100a8/S100a9

To better elucidate the role of *Sucnr1* in hematopoiesis through S100a8/S100a9 regulation, it would be valuable to conduct a direct comparison between tasquinimod-treated WT mice and tasquinimod-treated *Sucnr1*-KO mice. This comparison would clarify whether the observed hematopoietic effects, including changes in HSPC populations and lineage distribution, are specifically mediated by *Sucnr1*.

Additionally, investigating the combined effects of succinic acid treatment and tasquinimod in mouse models of pre-leukemic myelopoiesis and AML could provide a more comprehensive understanding of S100a8/S100a9 signaling in leukemogenesis and its potential modulation by *Sucnr1*. This approach would help identify the therapeutic relevance of targeting *Sucnr1* and its downstream pathways in both pre-leukemic and leukemic hematopoiesis.

6) Minor point: please correct "succinate" in Supplementary Figure 2 to "succinic acid."

Reviewer #2

(Remarks to the Author)

Starting from the observation that low succinate receptor 1 (SUCNR1) expression is associated with AML subtypes M0-M3 and correlates with poorer outcomes, Cuminetti et al. explore SUCNR1 role in hematopoiesis and its impact on AML progression. The authors utilize a *Sucnr1*-tdTomato reporter mouse to analyze its expression across hematopoietic compartments. This model reveals that SUCNR1 is predominantly expressed in LSK, HSC and MPP cells subsets, while other hematopoietic progenitors, suggesting a role for SUCNR1 in early hematopoietic differentiation.

To assess SUCNR1 function, the authors develop a KO model. In SUCNR1 KO mice, immature progenitor within LSK cells expand, but other populations increase in frequency, including mature B cells. This LSK expansion coincides with a gene expression shift toward an inflammatory profile and a reduced stem cell signature, including increase expression of S100A8/A9. Interestingly, progenitor-expanding phenotype is partly reversed by tasquinimod, an S100A8/A9 inhibitor, suggesting that S100A8/A9 signaling may be mediating these effects.

The study extends these findings into AML models (NRASG12D, MLL-AF9), where succinate treatment promotes myeloid output and accelerates disease progression. Pre-leukemic HSC/MPP subsets in Mx1-Cre NRASG12D show increased S100A8/A9 expression, mirroring the inflammatory state seen in SUCNR1 KO animals.

In human AML patient-derived xenografts (PDX) from two patients, chemoresistant cells exhibit increased inflammation and S100A8/A9 expression, while SUCNR1 expression is mainly observed in pre-treatment samples, suggesting possible downregulation in response to therapy.

Overall Comments

While the manuscript presents interesting data and some novel characterizations of SUCNR1 role in hematopoiesis, a lack of proper controls and incomplete analyses make data interpretation difficult.

- Manuscript Structure: The narrative is challenging to follow and at times counterintuitive, frequently shifting between AML models, human AML, healthy hematopoiesis, and KO mouse models. To improve readability, the structure should follow a linear narrative. This could start with SUCNR1 expression profiling in healthy mouse and human hematopoiesis (using gene expression databases, data from mouse models, and the *Sucnr1*-tdTomato reporter model). Profiling should cover both immature and mature myeloid cells, as some mature cells are also affected in the KO. Additionally, data on SUCNR1 expression in non-hematopoietic tissues (e.g., stromal cells) would help interpret findings from the KO mouse model. AML datasets could be used to differentiate SUCNR1 expression between healthy and malignant tissues. Conditional *Sucnr1* deletions in hematopoietic cells could clarify whether the observed hematopoietic phenotypes stem directly from HSPC-autonomous functions of *Sucnr1* or indirect effects via niche cells.

- Figure Titles: Figure titles often do not accurately reflect the content, making it hard for readers to follow the data being presented.

- FAB Classification: The FAB classification is mentioned but not systematically incorporated. Patient and PDX sample classifications by FAB subtype should be included for clarity.

- Inflammatory Cytokine Levels (Fig 3): The increase in mRNA for certain pro-inflammatory cytokines is not substantiated by corresponding protein levels in bone marrow fluid. Intracellular cytokine detection by flow cytometry would be more

appropriate. Given the inflammatory phenotype, a panel to detect immune cell activation in the KO model would be important.

- Focus on S100A8/A9: Among various inflammation-related factors, the rationale for focusing on S100A8/A9 as a primary mechanistic candidate is unclear.

- Colony Assay (Fig 4): The CFU assay results suggest that succinic acid has no effect on colony output in healthy BM, while monomethyl succinate (which bypasses SUCNR1) increases CFU output. Conversely, cis-epoxy succinate reduces colony formation in healthy LSK cells, with no effect in SUCNR1 KO cells. This implies that SUCNR1 activation may inhibit LSK proliferation while alternative pathways enhance progenitor potential. However, these hypotheses completely lack mechanistic insights and are somewhat counterintuitive.

- Controls in Fig 5: Healthy controls are missing in Fig 5. Without WT references, it is impossible to (1) measure the non-specific effects of tasquinimod on WT HSPC function and (2) evaluate to what extent the KO phenotype is reversed by tasquinimod.

- Functional Assays for SUCNR1 Role in HSCs: To substantiate the role of SUCNR1 in restricting HSC numbers and function, serial replating is suboptimal in mouse models. A competitive bone marrow transplant assay would be more relevant to test SUCNR1 effect on HSC engraftment potential, especially given the inflammatory gene expression signature observed, which typically impairs HSC function in WT. The *Sucnr1*-tdTomato reporter mouse could also be employed to compare the differential repopulating potential of SUCNR1-positive versus SUCNR1-negative HSCs.

- Gene Expression Levels in Fig 6A: Gene expression levels of S100A8/9 in Mx1-Cre NRASG12D/NRASG12D mice are less informative without corresponding SUCNR1 measurements in the same cell populations. Crossing this model with the reporter mouse or performing qPCR for SUCNR1 would add clarity.

- Single-Cell Data Presentation (Fig 6): The single-cell data presentation lacks depth, with SUCNR1 and S100A8/9 correlations presented visually but without quantification. Cluster annotations by sample, timepoint, or cell type are absent, making interpretation difficult. While S100A8/A9 are identified as markers of residual disease post-cytarabine treatment, it remains unclear if they actively contribute to chemoresistance or are upregulated as a stress response.

- Role in Metabolism: The manuscript suggests a role for SUCNR1 in hematopoietic metabolism, but this lacks direct experimental support.

Reviewer #3

(Remarks to the Author)

The manuscript presented by Cuminetti et al. offers a compelling study, demonstrating for the first time a direct influence of succinate and its receptor, SUCNR1, on cellular hematopoiesis, with implications for hematological malignancies. While prior research suggested that succinate/SUCNR1 might impact hematopoiesis indirectly, no studies to date have documented a direct effect, lending this study considerable novelty. Overall, the publication is well-structured and clearly written. However, for enhanced clarity to a broader audience, adding some contextual information in the results section could be beneficial. In certain areas, the reasoning behind specific experiments and the implications of the findings remain somewhat unclear (see comments below). Briefly outlining the observations could enrich the reader's experience significantly.

This study is extensive and detailed, and its results, along with the newly developed mouse model, hold promise for future mechanistic studies aimed at identifying targetable factors for potential therapeutic applications.

That said, I have several specific comments and questions that may enhance the already high quality of the manuscript:

•Figure 1A: Could you specify the type of survival displayed? It appears to be overall survival, but progression-free survival would also be informative.

•Figure 1D:

o What was the rationale for treating mice with succinic acid? Thus far, the indication for succinic acid involvement derives only from SUCNR1 expression differences in the patient cohort.

o How was succinic acid administered to the animals? Does the chosen route allow succinic acid to reach the cells directly, or might this be an indirect effect? Drawing conclusions about cellular metabolism from systemic metabolic changes can be challenging. Are elevated succinic acid levels detectable in the serum or other compartments in the mouse model?

o Considering the heterogeneity of AML, it is unfortunate that only one patient sample was used. Could this patient fall within either the SUCNR1 high or low expression groups?

•Figure 1G:

o The authors state that succinic acid injection worsens the disease. However, the administration route remains unclear (see above), and an explanation for this aggravation is lacking. Are elevated succinic acid levels observable in serum, other body fluids, organs, or cells?

o While significant, the increase in myeloid frequency is marginal. Could the authors comment on whether such a small difference holds biological significance? It appears the range in myeloid frequency in the vehicle group is wider than the mean difference between the vehicle and succinic acid groups.

- Figure 1J: It would be insightful to see corresponding protein levels.
- The concluding statement in the first Results section could benefit from added context. The link between low SUCNR1 expression and succinate as a contributor to leukemia progression is not fully clear.
- Figure 2A: Could you specify the exact mouse ages corresponding to “young adult” and “later middle-aged stage”?
- Supplemental Figure 4B: The claimed increase in live cell count is minimal and not statistically significant. Displaying actual frequencies instead of percentage changes from WT may be more illustrative. Also, the event count in the representative plots appears low (below 10,000 total events), which is below the standard for reliable flow cytometry.
- Supplemental Figure 4D: How can the knockout of SUCNR1 impact succinate levels in BMEF? As I understand it, SUCNR1 is activated by succinate but does not regulate succinate availability or metabolism. I would anticipate succinate accumulation due to receptor absence.
- Supplemental Figure 4E: Background controls for OCR and ECAR measurements, such as rotenone/antimycin A for OCR and 2-DG for ECAR, are absent. Without these, it is challenging to isolate mitochondrial respiration from other oxidative processes. Could such controls be included?
- Figure 3E: How do you explain the reduced IL-1 β protein levels in BMEF when IL-1 β gene expression is significantly increased in *Sucnr1*-KO versus WT animals? Consequently, the concluding statement of this section may require modification or expansion.
- Supplemental Figure 6E: The intent of this raw spectral ribbon plot is unclear. Without spectral unmixing, assigning signals to specific fluorochromes or fluorescent proteins is not possible.
- In the discussion, the authors note “overexpression of S100a8 and S100a9 in HSPC subsets [...]” How does this align with observations of increased S100A8/S100A9 in AML blasts in bone marrow compared to peripheral blood and the identified pro-inflammatory triggers reported in Blood Advances 2022 6(21)?

Reviewer #4

(Remarks to the Author)

The manuscript by Cuminetti et al. examines the role of Succinate receptor (*Suncr1*) in hematopoiesis. The authors show that low *Suncr1* expression correlates with poor survival in AML. Further, treatment of human and leukemia cells in vivo with succinic acid promotes leukemia progression. In the context of normal hematopoiesis *sucnr1* knockout resulted in the expansion of several stem and progenitor populations determined by cell number in aged mice. The increased in HSPC cell numbers correlated with an increase in gene expression in signatures associated with HSCs and a decreased gene expression in signatures associated with decreased differentiation and inflammation. Expression of s100a8 and s100a9 were increased in *sucnr1* knockdown mice. Treatment with s100a8/9 inhibitor results in reduced hematopoietic cells in the *sucnr1* knockdown mice. Finally, the authors show that *sucnr1* low expressing cells have high S100a8/9 in single-cell RNA-sequencing post cytarabine treatment (derived from PDX models). Overall, of the manuscript highlights several intriguing findings. However, several additional experiments/analyses should be performed to address the concerns below.

Major Concerns:

1. In Figure 2, the conclusions about changes in HSPC and mature cells are made based on cell numbers only. Most of the populations have increased numbers including total WBCs. Are there any changes in population percentages which may help to tease about if any one population is expanded upon *sucnr1* knockout?
2. The authors use the steady-state populations and gene expression analysis to conclude changes in HSPC function. These conclusions should be supported by performing competitive bone marrow transplants.
3. Figure 5 would benefit from including wild-type mice. This would allow the assessment of tasquinimod on hematopoiesis in the normal setting and enable the researchers to determine if treatment restores cell numbers back to the wild type levels.
4. Do monomethyl succinate and succinic acid have the same degree of effect on the intracellular biology of succinate? For example, to they increase oxidative phosphorylation to the same level. Differences between the compounds could be a confounding factor in the interpretation of results in figure 4.
5. Using a genetic approach like overexpressing *sucnr1* would help to solidify the conclusions from figure 4.
6. Further, overexpressing *sucnr1* and measuring the consequence on leukemia progression would enhance the study.
7. Figure 6 suggests that expression of *sucnr1* may influence the response to chemotherapy. It would strengthen the results by functionally testing this.

Minor:

1. In Figure 1D what are the patient characteristics? Specifically it would be helpful to include FAB and mutation status as the authors show these are related to *sucnr1* expression.

Reviewer #5

(Remarks to the Author)

Version 1:

Reviewer comments:

Reviewer #1

(Remarks to the Author)

The authors have addressed the reviewer's previous comments in a general and satisfactory manner, significantly improving the clarity and rigor of the manuscript. The experimental design, data presentation, and interpretation have been appropriately revised, and the manuscript is now substantially strengthened.

However, a few minor points remain that should be clarified or corrected to avoid potential confusion:

- It is not clear whether the efficiency of *Sucnr1* deletion was verified in the Mx1-Cre model, both in the bone marrow of the primary mice and in the transplanted recipients reconstituted with these BM cells. It is recommended to confirm the deletion by assessing *Sucnr1* expression levels (at the mRNA level) to ensure that the observed phenotypes are indeed due to effective loss of *Sucnr1* in hematopoietic cells.
- In the merged manuscript (lines 243–246), the authors state that the majority of *Sucnr1*⁺ HSPCs were identified as MPP3, followed by MPP4, with an average of 0.191% of *Sucnr1*⁺ cells (Figure 3B and Supplemental Figure 6C). Could the authors clarify that this order is correct, and not MPP2? In the figure, it appears that MPP2 might have a higher fraction than MPP4, so confirmation would help avoid confusion.
- The authors mention that in the first and second replating, both monomethyl succinate and cis-epoxy succinate stimulated the colony-forming potential of *Sucnr1*-KO HSCs, suggesting a loss of SUCNR1-mediated repression of this potential (Figure 3F). The Discussion ends with the sentence "Given the multi-layer intracellular functions of succinate, the question should be subject of future work" (lines 533–534), which could be refined for greater clarity and precision. The authors' results suggest that, in the absence of SUCNR1, cis-epoxy succinate may enhance colony-forming potential through receptor-independent mechanisms, possibly related to intracellular metabolic modulation. Have the authors assessed intracellular succinate levels to support this interpretation? Such data would help to substantiate the proposed mechanism and clarify whether the effects of the two compounds truly involve similar intracellular pathways.
- Please ensure that all figures include legends indicating the meaning of the colors used. In several cases (e.g., Figures 1F, 1K, 2I, and others), the color legends are missing. We recommend revising all figures to maintain consistency and clarity across the manuscript.

Reviewer #2

(Remarks to the Author)

Thanks for convincingly addressing my concerns

Reviewer #3

(Remarks to the Author)

The authors have thoroughly addressed my initial concerns, questions, and comments, which has further improved the overall already high quality of the manuscript. While most points were addressed experimentally, some were also resolved through well-reasoned justification and discussion. Well done, and thank you for this very nice piece of science.

Reviewer #4

(Remarks to the Author)

The authors have gone to great lengths to address concerns from five reviewers and significantly strengthened their manuscript. I have no additional concerns.

Version 2:

Reviewer comments:

Reviewer #1

(Remarks to the Author)

Thank you for addressing the remaining minor points. I have no further comments.

REVIEWER COMMENTS

We would like to thank all 5 Referees for the time you have spent with the revision of our work, your positive comments and constructive criticism. We believe our work is substantially improved and hope you will agree with us.

Reviewer #1 (Remarks to the Author): expert in Succinate and Sucnr1

The article presents a clear and well-structured exploration of Sucnr1's role in hematopoiesis, offering valuable insights into a complex and underexplored area of stem cell biology. The use of comprehensive experimental approaches—ranging from genetic manipulations to transcriptomic analysis—enhances the clarity of the conclusions and strengthens the overall impact of the study. The figures are well-organized and effectively support the narrative, providing clear visual representations of the data. Overall, the manuscript is excellently presented, with a clear focus on the novelty of the findings and their potential implications in the context of hematopoiesis. However, the manuscript would benefit by addressing certain key issues.

1) The global knockout model used may limit the interpretation of Sucnr1's role in hematopoiesis, as indirect effects arising from systemic Sucnr1 deficiency could influence hematopoietic stem and progenitor cell (HSPC) responses. To clarify whether Sucnr1's effects are intrinsic to hematopoietic cells or arise from changes in the microenvironment, a lineage-specific knockout approach would be insightful. This strategy could help differentiate between direct roles of Sucnr1 within hematopoietic cells and broader physiological or paracrine influences, thereby enhancing our understanding of its role in self-renewal and differentiation dynamics in hematopoiesis.

We thank the Reviewer for this comment. To dissect the role of Sucnr1 on healthy hematopoiesis via the hematopoietic compartment, we transplanted *Sucnr1^{fl/fl}* control or *Mx1-Cre Sucnr1^{fl/fl}* experimental BM cells into C57BL/6J WT mice, which were later injected with poly-inosine:poly-cytosine (plpC; please, see schematics in Figure 2I). *In vivo* deletion of Sucnr1 from hematopoietic cells resulted in early expansion of the numbers of circulating total leukocytes, i.e. myeloid cells, monocytes and lymphocytes (Figure 2J), which was maintained over the course of the experiment (Figure 2K). In BM, *in vivo* deletion of Sucnr1 from hematopoietic cells led to an expansion of MPP versus intact recipients, particularly MPP4 (Figure 2L). This means that Sucnr1 effects can be explained greatly by intrinsic effects to hematopoietic cells. These ideas have been included in the manuscript.

The end-point analysis of this experiment was performed 47 weeks (about 11 months) after the transplant, to make it a time-point comparable to the global deletion of Sucnr1. The differences compared with the results obtained in Sucnr1-KO may arise from the experimental conditions (irradiation, plpC-induced recombination, etc) and/or a potential contribution from the stem cell microenvironment and/or other systemic effects that should be subject of future work.

2) The effects of Sucnr1 deficiency on hematopoiesis appear contradictory. Sucnr1 deletion results in both an expansion of HSPCs and increased hematopoietic activity in the bone marrow and spleen, yet, paradoxically, it also shows reduced transcriptomic markers of stemness and self-renewal. Could the authors provide a more in-depth discussion to explain this finding? Additionally, targeted experiments investigating differential effects across specific HSPC subpopulations (e.g., long-term vs. short-term HSCs, or lineage-specific

progenitors) could help clarify whether *Sucnr1*'s impact varies across these subsets, potentially reconciling the observed expansion with the reduction in stemness markers.

Thank you for bringing up this interesting topic. We have performed new FACS-sorting of long-term HSC, short-term HSC and MPP obtained from the BM of WT and *Sucnr1*-KO mice, and performed ddPCR on two important genes related to stemness (*Procr*, *Hoxa9*). The expression of both genes was reduced in the gene expression profiling by RNA-Seq of *Sucnr1*-KO LSK cells versus WT cells, and their reduced expression levels were also confirmed by ddPCR in independent samples. Importantly, when analyzing different HSPC subsets, we find that the levels of expression of *Procr* and *Hoxa9* are significantly reduced only in the MPP subset, but not in the long-term and short-term HSC compartments (new Figure 4D).

We are also providing new data on the functionality of HSPC in vivo analyzed by competitive repopulation assays of BM nucleated cells isolated from WT or *Sucnr1*-KO mice (Ly5.2) into lethally irradiated Ly5.1 congenic recipients. Peripheral blood analysis from recipient mice 16 weeks after transplantation revealed a mild reduction in HSPC engraftment in recipients transplanted with *Sucnr1*-KO cells compared to recipients transplanted with WT control cells (please see new Figure 3G).

As discussed in the manuscript, these seemingly paradoxical effects of *Sucnr1* deletion on HSPC with increased numbers/survival/colony-forming potential versus reduced engraftment in competition/proliferation/gene expression markers of stemness may be important factors underlying the chronic nature of the hematopoietic abnormalities displayed by *Sucnr1*-KO mice, which become symptomatic only at the middle-age.

This paradox between numbers and function is common to other chronic conditions, like ageing and related chronic inflammatory conditions, where the HSPC pool is increased in numbers but reduced in function, representing a shift towards a biased, less effective and abnormal hematopoietic system. The potential role of *Sucnr1* in the ageing hematopoietic system should be subject of future work. We have included these ideas in the Discussion section.

Further, although *Sucnr1*-KO BM cells are less fit in competition with WT cells, as explained in the point above, primary transplants of *Sucnr1^{fl/fl}* or *Mx1-Cre Sucnr1^{fl/fl}* BM cells into C57BL/6J WT mice, induced after stable engraftment, result in expanded hematopoiesis and MPP compartment, greatly recapitulating the hematopoietic defects of *Sucnr1*-KO mice (Figure 2I-2L).

To further substantiate the role of *Sucnr1* in HSPC function in vivo, we performed competitive repopulation assays using 200 *Sucnr1*-tdTomato⁺ or *Sucnr1*-tdTomato⁻ FACS-sorted HSPC (containing a mix of 40 HSC, 86 MPP3 and 74 MPP4) isolated from Ly5.2 *Sucnr1*-tdTomato mice into lethally irradiated Ly5.1 congenic recipients. Peripheral blood analysis from recipient mice 10 weeks after the transplant showed a reduction of circulating leukocytes and donor-derived monocytes in recipients of *Sucnr1*⁺ cells compared to recipients of *Sucnr1*⁻ cells (Figure 3H).

Thus, *Sucnr1* expression marks a subset of HSPC with restricted engraftment potential in vivo.

3) Section: Low SUCNR1 Predicts Acute Myeloid Leukemia Progression

In lines 141-143, it is mentioned that "MLL-AF9+ mice had overt AML, and the lineage-negative compartment was expanded in the bone marrow of succinic acid-treated mice

compared to vehicle-treated controls (Figure 1L), showing an increase in the LK fraction but no changes in LSK cells (supplemental Figure 2E)." While there was an observed trend toward an increase in both the LK fraction and LSK cells in succinic acid-treated mice, it is crucial to emphasize that none of these differences reached statistical significance. Adding this clarification in the text would provide a more accurate and nuanced interpretation of the data.

Thanks for pointing out this error, which has now been corrected and reads "with an observed trend toward increased LK fraction and LSK cells that did not reach statistical significance".

In line 143, the statement that 'low SUCNR1 represents a prognostic marker for reduced survival in AML patients' would benefit from further statistical validation to strengthen its impact. To rigorously establish SUCNR1 as an independent prognostic factor, a Cox regression analysis should be performed. This would allow the authors to assess the effect of SUCNR1 on survival while adjusting for other known prognostic variables in AML. This addition would significantly enhance the robustness of the study's conclusions regarding the prognostic value of SUCNR1 in AML.

We have now performed Cox regression analysis, which uncovered SUCNR1 expression as an independent predictor of both overall survival and progression-free survival after adjusting for age, gender and FAB classification. Please, see results in new Table 1.

4) Section: *Sucnr1* is expressed in HSPC, and its activation restricts their functional response to succinate *ex vivo*

In line 223, the authors suggest that *Sucnr1* activation suppresses succinate's intracellular effects, supported by their observations with monomethyl succinate, which increased colony-forming potential in total bone marrow (TBM) in a dose-dependent manner. However, this conclusion is not entirely convincing. If *Sucnr1* activation indeed dampens succinate's effects, one would expect to see a reduction in CFU-C in TBM with succinate treatment (Figure 4A), but no such decrease is observed. This raises questions about the robustness of the authors' claim. Additionally, to strengthen their argument, the authors could have tested the combination of monomethyl succinate with succinic acid to explore potential synergistic effects, which would provide more clarity on their hypothesis.

We appreciate this comment. Monomethyl succinate is inactive on *Sucnr1* and enhances the colony-forming potential in total BM (TBM), in a dose-response manner. Conversely, succinic acid, which displays both *Sucnr1*-dependent and independent effects, results in unchanged colony-forming potential. Taken together, we reasoned that these results suggest that *Sucnr1* activation represses the intracellular effect of succinate. We have reformulated these results for clarity. Further, we have followed the Referee suggestion and explored the effect of succinic acid and monomethyl succinate combo, which results in a partial increase in colony-forming potential in TBM versus succinic acid alone and a partial reduction versus monomethyl succinate alone, providing robust evidence that *Sucnr1* activation represses the intracellular effect of succinate. Please, see the results in new Figure 3D.

Additionally, clarification is needed on why monomethyl succinate increased CFU-C in FACS-sorted HSC from *Sucnr1*-KO mice in the first plating (Figure 4C), while this effect was not seen in cells from WT mice (Figure 4B). Could the difference be attributed to varying intracellular succinate levels between the two mouse strains? Has the possibility of altered succinate accumulation been considered? Without addressing this, the interpretation of the results remains incomplete and raises doubts about the underlying mechanisms.

This is a legit consideration, and we had indeed taken into account the possibility of increased intracellular succinate levels in *Sucnr1*-KO mice versus WT mice. In fact, as stated in the Results section of the previous version of our manuscript, “Given that succinate is an intermediate metabolite of the TCA cycle, we studied its potentially increased cycling which would result in metabolic changes. *Sucnr1* deletion resulted in increased baseline oxygen consumption rate in lineage-negative progenitors (supplemental Figure 4G). This effect was specific and reversible because the maximal oxygen consumption rate and the glycolytic flux remained unchanged (supplemental Figure 4G). Consistent with these data, both total and mitochondrial reactive oxygen species (ROS) levels were higher in LSK cells from *Sucnr1*-KO mice (supplemental Figure 4H).” We have now established a new collaboration that has enabled us to provide direct evidence using metabolomics in HSC, which confirms increased intracellular succinate levels in *Sucnr1*-KO versus WT mice (please see new supplemental Figure 4F). The regulation of succinate localization by *Sucnr1* should be subject of future work. Whereas it could simply be a matter of increased succinate availability in the absence of *Sucnr1* binding, active regulatory mechanisms could additionally take place. We have added these ideas to the Discussion.

In parallel, we provide compelling evidence showing that activation of *Sucnr1* counterbalances the stimulatory effect of intracellular succinate in HSPC, via control of *S100a8/S100a9*. Consistent with the increased mRNA expression of *S100a8*, *S100a9* in HSPC from *Sucnr1*-KO versus WT mice, the protein levels of the alarmin complex *S100A8/S100A9* were increased in the BMEF. Notably, intracellular succinate activates metabolism and may have multiple additional effects, but treatment with tasquinimod, *S100a9* blocker, rescued the hematopoietic defects of *Sucnr1*-KO HSPC both added ex vivo and in vivo.

A remaining open question was whether *S100a9* pathway could be both repressed by *Sucnr1* signaling and activated by increased intracellular succinate. To answer this question, we have used available material from cell lines, where we show that the combination of cis-epoxy succinate but not monomethyl succinate and AraC reduces the expression of *S100A9*, versus AraC alone (new supplemental Figure 11C). Thus, *S100a9* is an alternative pathway controlled by *Sucnr1*, independent of intracellular succinate, which counterbalances the stimulatory effect of intracellular succinate.

5) Section: *Sucnr1* restricts HSPC and hematopoiesis via control of *S100a8/S100a9*

To better elucidate the role of *Sucnr1* in hematopoiesis through *S100a8/S100a9* regulation, it would be valuable to conduct a direct comparison between tasquinimod-treated WT mice and tasquinimod-treated *Sucnr1*-KO mice. This comparison would clarify whether the observed hematopoietic effects, including changes in HSPC populations and lineage distribution, are specifically mediated by *Sucnr1*.

As suggested by Reviewers #1, #2 and #5, and #4, to substantiate a selective role of *S100a8/S100a9* in the hematopoietic system via *Sucnr1* deletion, we performed an independent experiment treating both *Sucnr1*-KO and WT mice with tasquinimod or vehicle in parallel (supplemental Figure 8A). Confirming our previous results, tasquinimod treatment in *Sucnr1*-KO mice reduced the expanded numbers of circulating leukocytes to the numbers observed in WT mice, including both overall myeloid cells as well as B cells and with no significant effect observed in WT mice (supplemental Figure 8B). Tasquinimod administration reduced the expanded numbers of monocytes in BM of *Sucnr1*-KO mice to the numbers found in WT mice, whilst no effect was observed in WT mice (supplemental Figure 8C). Conversely, treatment with tasquinimod reduced the numbers of BM B cells to similar levels in both *Sucnr1*-KO and WT mice (supplemental Figure 8C). The expansion of BM HSC in

Sucnr1-KO mice was reduced by treatment with tasquinimod, but a minor reduction in HSC numbers was also observed in WT mice, bringing HSC values in both tasquinimod-treated groups to similar level (supplemental Figure 8D). In turn, the increased colony-forming potential in Sucnr1-KO mice was reduced to WT values, with no effect seen in samples obtained from WT mice (supplemental Figure 8E). Of note, this assay is optimal for the growth of primitive erythroid progenitor cells, granulocyte-macrophage progenitor cells, and multi-potential granulocyte, erythroid, macrophage, megakaryocyte progenitor cells, but not for the growth of lymphoid progenitor cells. Thus, these data suggest that tasquinimod effect originates in the HSC compartment for both lineages, and whereas the effect of tasquinimod on the myeloid lineage is fully dependent on Sucnr1 deletion, the effect on the B cell lineage is only partially dependent on the loss of Sucnr1 signaling.

Additionally, investigating the combined effects of succinic acid treatment and tasquinimod in mouse models of pre-leukemic myelopoiesis and AML could provide a more comprehensive understanding of S100a8/S100a9 signaling in leukemogenesis and its potential modulation by Sucnr1. This approach would help identify the therapeutic relevance of targeting Sucnr1 and its downstream pathways in both pre-leukemic and leukemic hematopoiesis.

Thank you for the opportunity to address this question. To study the therapeutic potential of targeting Sucnr1 signaling pathway in AML, we turned to the more aggressive Mll-AF9 mouse model, which express low levels of Sucnr1 (Figure 1L). We transplanted Mll-AF9+ BM into lethally irradiated recipients and treated them with tasquinimod, tasquinimod in combination with cis-epoxy succinate, or the corresponding vehicles, for a maximum of 24 weeks (Figure 6A). The combination of tasquinimod and cis-epoxy succinate reduced the numbers of circulating leukocytes (Figure 6B) and Gr-1^{hi} blasts in spleen (Figure 6C), compared to mice receiving the vehicle for both drugs. In the BM, we found that tasquinimod alone decreased the numbers of c-Kit⁺FcRγ⁺ GMP-like leukemic blasts versus mice treated with vehicle (Figure 6D-6E). Further, tasquinimod reduced the number of BM c-Kit⁺FcRγ⁺CD48⁺CD34⁺ lymphoid-primed multipotent progenitor (LMPP) LSC when administered alone or in combination with cis-epoxy succinate, versus the corresponding vehicle-treated mice (Figure 6D-6E). These data indicate that activating Sucnr1 signaling with a potent Sucnr1 agonist and its downstream pathway through S100A9 blocking shows therapeutic value in Mll-AF9+ mice, reducing immunophenotypically-defined LSC in the BM and peripheral leukemic burden.

6) Minor point: please correct “succinate” in Supplementary Figure 2 to “succinic acid.”

Thanks for noticing, we have corrected this mistake.

Reviewer #2 (Remarks to the Author): expert in RNA-seq and scRNA-seq analysis, AML and hematopoiesis

Starting from the observation that low succinate receptor 1 (SUCNR1) expression is associated with AML subtypes M0-M3 and correlates with poorer outcomes, Cuminetti et al. explore SUCNR1 role in hematopoiesis and its impact on AML progression. The authors utilize a Sucnr1-tdTomato reporter mouse to analyze its expression across hematopoietic compartments. This model reveals that SUCNR1 is predominantly expressed in LSK, HSC and MPP cells subsets, while other hematopoietic progenitors, suggesting a role for SUCNR1 in early hematopoietic differentiation.

To assess SUCNR1 function, the authors develop a KO model. In SUCNR1 KO mice, immature progenitor within LSK cells expand, but other populations increase in frequency,

including mature B cells. This LSK expansion coincides with a gene expression shift toward an inflammatory profile and a reduced stem cell signature, including increase expression of S100A8/A9. Interestingly, progenitor-expanding phenotype is partly reversed by tasquinimod, an S100A8/A9 inhibitor, suggesting that S100A8/A9 signaling may be mediating these effects.

The study extends these findings into AML models (NRASG12D, MLL-AF9), where succinate treatment promotes myeloid output and accelerates disease progression. Pre-leukemic HSC/MPP subsets in Mx1-Cre NRASG12D show increased S100A8/A9 expression, mirroring the inflammatory state seen in SUCNR1 KO animals.

In human AML patient-derived xenografts (PDX) from two patients, chemoresistant cells exhibit increased inflammation and S100A8/A9 expression, while SUCNR1 expression is mainly observed in pre-treatment samples, suggesting possible downregulation in response to therapy.

Overall Comments

While the manuscript presents interesting data and some novel characterizations of SUCNR1 role in hematopoiesis, a lack of proper controls and incomplete analyses make data interpretation difficult.

- Manuscript Structure: The narrative is challenging to follow and at times counterintuitive, frequently shifting between AML models, human AML, healthy hematopoiesis, and KO mouse models. To improve readability, the structure should follow a linear narrative.

We appreciate there are various ways to effectively convey a scientific report. We prefer to start with the most important and clinically relevant information to catch the attention of the reader, then continue with the body of the work where the mechanisms (cellular and molecular) are uncovered, mainly by use of biologically relevant mouse models and paralleled validations in humans where possible. In this work, we end by using the mechanistic information back in AML mouse models, patients and human cell lines.

This is the only comment referring to “challenging” structure out of four Reviewer reports, and two other Reviewers (#1 and #3) in fact refer to the manuscript as “well-structured”, “clear” and “well-written”. However, given the extensive revisions, we have made a small reorganization of the narrative from “Sucnr1 restricts HSPC and hematopoiesis under steady-state” to “Sucnr1 is expressed in HSPC and its activation restricts their functional response”, followed by “Sucnr1 preserves transcriptional programs characteristic of HSPC” and “Sucnr1 restricts HSPC and hematopoiesis via control of S100a8/S100a9”.

This could start with SUCNR1 expression profiling in healthy mouse and human hematopoiesis (using gene expression databases, data from mouse models, and the Sucnr1-tdTomato reporter model). Profiling should cover both immature and mature myeloid cells, as some mature cells are also affected in the KO.

To answer these questions, HSPC subsets, committed hematopoietic progenitors and differentiated cells obtained from the mouse BM were analyzed by spectral flow cytometry. After spectral unmixing, we quantified the Sucnr1-tdTomato+ cell fractions (Figure 3A and supplemental Figure 6A-6D). Our analysis revealed that all HSPC subsets show a varying fraction of Sucnr1+ cells and the majority of Sucnr1+ HSPC were identified as MPP3, with an average of 0.191% of Sucnr1+ cells, followed by MPP4 (Figure 3B and supplemental Figure 6C). Importantly, although Sucnr1+ cells were found across all hematopoietic cell subsets analyzed, including all lineage-positive cell types (supplemental Figure 6C-6D), the

majority of *Sucnr1*⁺ cells in the hematopoietic system were identified as lineage-negative progenitors, including all LK progenitors particularly GMP, LSK cells and CLP (supplemental Figure 6E).

In view of the expression of *Sucnr1* across lineage-positive cell in the BM, future studies should explore its potential role regulating their primary and effector functions. This idea has been included in the Discussion.

In parallel, we confirmed SUCNR1 mRNA expression across all hematopoietic cell subsets analyzed in normal human hematopoiesis, using expression data from the HemaExplorer collection at BloodSpot (GSE17054, GSE19599, GSE11864 and E-MEXP-1242), including HSC and HSPC (supplemental Figure 6F). In humans, CMP, GMP and differentiating promyelocytes and myelocytes display remarkably high levels of SUCNR1 mRNA expression. The expression of SUCNR1 in CLP was not available (supplemental Figure 6F).

Additionally, data on SUCNR1 expression in non-hematopoietic tissues (e.g., stromal cells) would help interpret findings from the KO mouse model.

Please, see answer below referring to “conditional *Sucnr1* deletions in hematopoietic cells could clarify whether the observed hematopoietic phenotypes stem directly from HSPC-autonomous functions of *Sucnr1* or indirect effects via niche cells”. *Sucnr1* effects on hematopoiesis can be explained greatly by intrinsic effects to hematopoietic cells.

As stated in the Discussion, although partial contribution of the BM stem cell microenvironment or metabolic systemic effects with potential to influence hematopoiesis will require future investigation, taken together our data provide compelling evidence that supports for a direct, selective role of *Sucnr1* on HSPC.

AML datasets could be used to differentiate SUCNR1 expression between healthy and malignant tissues.

Please, see new Figure 1D, where we are showing now data obtained from the dataset TARGET AML at BloodSpot (www.fobinf.com) confirming great heterogeneity in SUCNR1 expression in primary AML and overall lower expression values compared to controls (GSE42519 and GSE13159).

Conditional *Sucnr1* deletions in hematopoietic cells could clarify whether the observed hematopoietic phenotypes stem directly from HSPC-autonomous functions of *Sucnr1* or indirect effects via niche cells.

We thank the Reviewers for this comment. To dissect the role of *Sucnr1* on healthy hematopoiesis via the hematopoietic compartment, we transplanted *Sucnr1*^{fl/fl} control or *Mx1-Cre Sucnr1*^{fl/fl} experimental BM cells into C57BL/6J WT mice, which were later injected with poly-inosine:poly-cytosine (pIpC; please, see schematics in Figure 2I). *In vivo* deletion of *Sucnr1* from hematopoietic cells resulted in early expansion of the numbers of circulating total leukocytes, i.e. myeloid cells, monocytes, or lymphocytes (Figure 2J), which was maintained over the course of the experiment (Figure 2K). In BM, *in vivo* deletion of *Sucnr1* from hematopoietic cells led to an expansion of MPP versus intact recipients, particularly MPP4 (Figure 2L). This means that *Sucnr1* effects can be explained greatly by intrinsic effects to hematopoietic cells. The end-point analysis of this experiment was performed 47 weeks (about 11 months) after the transplant, to make it a time-point comparable to the global deletion of *Sucnr1*.

- Figure Titles: Figure titles often do not accurately reflect the content, making it hard for readers to follow the data being presented.

For clarity, the main Figure titles now match the heading of each part of the Results section. The Supplemental Figure titles have been improved accordingly.

- FAB Classification: The FAB classification is mentioned but not systematically incorporated. Patient and PDX sample classifications by FAB subtype should be included for clarity.

The patient and PDX sample were classified as M1. This information has now been included in the manuscript.

- Inflammatory Cytokine Levels (Fig 3): The increase in mRNA for certain pro-inflammatory cytokines is not substantiated by corresponding protein levels in bone marrow fluid. Intracellular cytokine detection by flow cytometry would be more appropriate. Given the inflammatory phenotype, a panel to detect immune cell activation in the KO model would be important.

Thank you for this important question. Given the increased mRNA expression of *Il1b* in HSPC from *Sucnr1*-KO versus WT mice, we analyzed the intracellular levels of IL-1 β protein and confirmed them slightly but significantly increased in *Sucnr1*-KO compared to WT mice (Figure 4E). This effect seemed to be specific of the HSPC compartment, as we observed no differences in the intracellular levels of IL-1 β in total live cells, lineage-positive cells, lineage-negative progenitors or LK progenitors (supplemental Figure 7C). This is in contrast with the levels of IL-1 β , which we found to be reduced in the BM of *Sucnr1*-KO versus WT mice. Thus, *in vivo* deletion of *Sucnr1* promotes inflammation specifically in HSPC. We have included these ideas in the Results and Discussion sections.

- Focus on S100A8/A9: Among various inflammation-related factors, the rationale for focusing on S100A8/A9 as a primary mechanistic candidate is unclear.

Despite complex changes taking place in the global cytokine status of the BM in *Sucnr1*-KO mice, S100a8 and S100a9 were consistently increased in their mRNA expression in HSPC and in their protein levels in BMEF, making it a promising mechanistic candidate to follow up. We have included this idea in the corresponding Results section.

- Colony Assay (Fig 4): The CFU assay results suggest that succinic acid has no effect on colony output in healthy BM, while monomethyl succinate (which bypasses SUCNR1) increases CFU output. Conversely, *cis*-epoxy succinate reduces colony formation in healthy LSK cells, with no effect in SUCNR1 KO cells. This implies that SUCNR1 activation may inhibit LSK proliferation while alternative pathways enhance progenitor potential. However, these hypotheses completely lack mechanistic insights and are somewhat counterintuitive.

We appreciate this comment, which is similar to comment 4) from Reviewer #1. Briefly, monomethyl succinate is inactive on *Sucnr1* and enhances the colony-forming potential in total BM (TBM), in a dose-response manner. Conversely, succinic acid, which displays both *Sucnr1*-dependent and independent effects, results in unchanged colony-forming potential. Taken together, we reasoned that these results suggest that *Sucnr1* activation represses the intracellular effect of succinate. We have reformulated these results for clarity. Further, we have followed Referee #1 suggestion and explored the effect of succinic acid and monomethyl succinate combination, which results in a partial increase in colony-forming potential in TBM versus succinic acid alone and a partial reduction versus monomethyl succinate alone, confirming that *Sucnr1* activation represses the intracellular effect of succinate. Please, see the results in new Figure 3D.

Of note, *cis*-epoxy succinate, 10- to 20-fold more potent than succinic acid on *Sucnr1*, reduced the colony-forming potential of healthy HSC (Figure 3E) whereas it stimulated the

colony-forming potential of *Sucnr1*-KO HSC (Figure 3F), suggesting loss of colony-forming potential repression by *Sucnr1* activation in the absence of *Sucnr1*. These data were present in the previous version of the manuscript.

In parallel, we provide compelling evidence showing that activation of *Sucnr1* counterbalances the stimulatory effect of intracellular succinate in HSPC, via control of S100a8/S100a9. Notably, treatment with tasquinimod, S100a9 blocker, rescued the hematopoietic defects of *Sucnr1*-KO HSPC both added *ex vivo* and *in vivo*.

A remaining open question was whether S100a9 pathway could be both repressed by *Sucnr1* signaling and activated by increased intracellular succinate. To answer this question, we have used available material from cell lines, and we show now that the combination of cis-epoxy succinate but not monomethyl succinate and AraC reduces the expression of S100A9, versus AraC alone (new supplemental Figure 11C). Thus, S100a9 seems to be an alternative pathway controlled by *Sucnr1*, independent of intracellular succinate, which counterbalances the stimulatory effect of intracellular succinate.

- Controls in Fig 5: Healthy controls are missing in Fig 5. Without WT references, it is impossible to (1) measure the non-specific effects of tasquinimod on WT HSPC function and (2) evaluate to what extent the KO phenotype is reversed by tasquinimod.

As suggested by Reviewers #1, #2 and #5, and #4, to substantiate a selective role of S100a8/S100a9 in the hematopoietic system via *Sucnr1* deletion, we performed an independent experiment treating both *Sucnr1*-KO and WT mice with tasquinimod or vehicle in parallel (supplemental Figure 8A). Confirming our previous results, tasquinimod treatment in *Sucnr1*-KO mice reduced the expanded numbers of circulating leukocytes to the numbers observed in WT mice, including both overall myeloid cells as well as B cells and with no significant effect observed in WT mice (supplemental Figure 8B). Tasquinimod administration reduced the expanded numbers of monocytes in BM of *Sucnr1*-KO mice to the numbers found in WT mice, whilst no effect was observed in WT mice (supplemental Figure 8C). Conversely, treatment with tasquinimod reduced the numbers of BM B cells to similar levels in both *Sucnr1*-KO and WT mice (supplemental Figure 8C). The expansion of BM HSC in *Sucnr1*-KO mice was reduced by treatment with tasquinimod, but a minor reduction in HSC numbers was also observed in WT mice, bringing HSC values in both tasquinimod-treated groups to similar level (supplemental Figure 8D). In turn, the increased colony-forming potential in *Sucnr1*-KO mice was reduced to WT values, with no effect seen in samples obtained from WT mice (supplemental Figure 8E). Of note, this assay is optimal for the growth of primitive erythroid progenitor cells, granulocyte-macrophage progenitor cells, and multi-potential granulocyte, erythroid, macrophage, megakaryocyte progenitor cells, but not for the growth of lymphoid progenitor cells. Thus, these data suggest that tasquinimod effect originates in the HSC compartment for both lineages, and whereas the effect of tasquinimod on the myeloid lineage is fully dependent on *Sucnr1* deletion, the effect on the B cell lineage is only partially dependent on the loss of *Sucnr1* signaling.

- Functional Assays for SUCNR1 Role in HSCs: To substantiate the role of SUCNR1 in restricting HSC numbers and function, serial replating is suboptimal in mouse models. A competitive bone marrow transplant assay would more relevant to test SUCNR1 effect on HSC engraftment potential, especially given the inflammatory gene expression signature observed, which typically impair HSC function in WT. The *Sucnr1*-tdTomato reporter mouse could also be employed to compare the differential repopulating potential of SUCNR1-positive versus SUCNR1-negative HSCs.

We agree with the Reviewers. We are now providing new data on the functionality of HSPC in vivo analyzed by competitive repopulation assays of BM nucleated cells isolated from WT or *Sucnr1*-KO mice (Ly5.2) into lethally irradiated Ly5.1 congenic recipients. Peripheral blood analysis from recipient mice 16 weeks after transplantation revealed a mild reduction in HSPC engraftment in recipients transplanted with *Sucnr1*-KO BM cells compared to recipients transplanted with WT BM control cells (please see new Figure 3G).

Although *Sucnr1*-KO BM cells seem to be less fit in competition with WT cells, as explained in the point above, primary transplants of *Sucnr1^{fl/fl}* or *Mx1-Cre Sucnr1^{fl/fl}* BM cells into C57BL/6J WT mice greatly recapitulate the hematopoietic expansion of *Sucnr1*-KO mice (Figure 2I-2L).

To further substantiate the role of *Sucnr1* in HSPC function in vivo, we then performed competitive repopulation assays using 200 *Sucnr1*-tdTomato⁺ or *Sucnr1*-tdTomato⁻ FACS-sorted HSPC (containing a mix of 40 HSC, 86 MPP3 and 74 MPP4) isolated from Ly5.2 *Sucnr1*-tdTomato mice into lethally irradiated Ly5.1 congenic recipients. Peripheral blood analysis from recipient mice 10 weeks after the transplant showed a reduction of circulating leukocytes and donor-derived monocytes in recipients of *Sucnr1*⁺ cells compared to recipients of *Sucnr1*⁻ cells (Figure 3H). Thus, *Sucnr1* expression marks a subset of HSPC with restricted engraftment potential in vivo.

The conclusions on functionality drawn from the experiments on serial replatings have also been toned down throughout the manuscript.

- Gene Expression Levels in Fig 6A: Gene expression levels of S100A8/9 in *Mx1-Cre NRASG12D/NRASG12D* mice are less informative without corresponding SUCNR1 measurements in the same cell populations. Crossing this model with the reporter mouse or performing qPCR for SUCNR1 would add clarity.

We have recently moved our lab and activities from the University of Tromsø to the University of Oslo and unfortunately, we do not have available animals of these strains to perform cell sortings at this time. However, we have been able to quantify *Sucnr1* expression in frozen total BM nucleated cell samples, and found it reduced in samples obtained from pl-pC-induced *Mx1-Cre NRAS-G12D* versus *NRAS-G12D* controls (see new Figure 1F).

- Single-Cell Data Presentation (Fig 6): The single-cell data presentation lacks depth, with SUCNR1 and S100A8/9 correlations presented visually but without quantification. Cluster annotations by sample, timepoint, or cell type are absent, making interpretation difficult. While S100A8/A9 are identified as markers of residual disease post-cytarabine treatment, it remains unclear if they actively contribute to chemoresistance or are upregulated as a stress response.

We apologize for the lack of clarity regarding the correlation and annotations in our single-cell analysis, and we thank the reviewer for the opportunity to clarify our results. Annotation by sample and timepoint is now provided in supplemental Figure 10A. The single-cell RNA-sequencing (scRNA-seq) dataset corresponds to two AML patient-derived xenografts (PDX) analyzed before and after 8 days of a 5-day cytarabine (AraC) treatment (GSE178912). After integration with the Harmony algorithm, we performed unbiased clustering using K-nearest neighbor analysis (Figure 7B), which identified seven clusters (0–6).

Following AraC treatment, clusters 0 and 1 expanded markedly. Together with cluster 5, these clusters displayed a shared transcriptional program characterized both by the residual disease signature post-AraC (AraC_UP)¹ and by an inflammatory/senescence–stress signature (CISG_UP)² (Figure 7C). Notably, S100A8 and S100A9 were highly expressed in

these persisting clusters (0, 1, 5) (Figure 7D). Their expression was significantly increased post-AraC compared to diagnosis (Figure 7E–7F) and correlated with the residual disease signature (Figure 7G).

In contrast, SUCNR1 was expressed at low levels in cluster 3, which decreased in numbers after AraC treatment and showed low expression of S100A8 and S100A9 (Figure 7D–7E). Functional enrichment analysis (RunEnrichment, SCP package) revealed that cluster 3 was enriched for mitochondrial and metabolic processes (supplemental Figure 10B). This finding is consistent with previous studies showing that AraC treatment induces a metabolic rewiring characterized by enhanced mitochondrial activity, a key driver of chemoresistance³.

Thus, the reduction of cluster 3 post-AraC suggests that SUCNR1-expressing cells represent a metabolically active population with a distinct role in shaping treatment response. The metabolic adaptation triggered by increased mitochondrial activity may sustain chemopersistence in vivo. In this context, SUCNR1 expression, as a marker of this cell subset, could represent both an important determinant of chemotherapy response and a potential therapeutic vulnerability.

- Role in Metabolism: The manuscript suggests a role for SUCNR1 in hematopoietic metabolism, but this lacks direct experimental support.

We agree with the Reviewers that our data provides evidence for a role of *Sucnr1* in HSPC metabolism. As stated in the Results section of the previous version of our manuscript, “Given that succinate is an intermediate metabolite of the TCA cycle, we studied its potentially increased cycling which would result in metabolic changes. *Sucnr1* deletion resulted in increased baseline oxygen consumption rate in lineage-negative progenitors (supplemental Figure 4G). This effect was specific and reversible because the maximal oxygen consumption rate and the glycolytic flux remained unchanged (supplemental Figure 4G). Consistent with these data, both total and mitochondrial reactive oxygen species (ROS) levels were higher in LSK cells from *Sucnr1*-KO mice (supplemental Figure 4H).” We have now established a new collaboration that has enabled us to provide direct evidence using metabolomics in HSC, which confirms increased intracellular succinate levels in *Sucnr1*-KO versus WT mice (please see new supplemental Figure 4F). The regulation of succinate localization by *Sucnr1* should be subject of future work. Whereas it could simply be a matter of increased succinate availability in the absence of *Sucnr1* binding, active regulatory mechanisms could additionally take place. We have added these ideas to the Discussion.

However, in parallel, we provide compelling evidence showing that activation of *Sucnr1* counterbalances the stimulatory effect of intracellular succinate in HSPC, via control of S100a8/S100a9. Consistent with the increased mRNA expression of *S100a8*, *S100a9* in HSPC from *Sucnr1*-KO versus WT mice, the protein levels of the alarmin complex S100A8/S100A9 were increased in the BMEF. Notably, intracellular succinate activates metabolism and may have multiple additional effects, but treatment with tasquinimod, S100a9 blocker, rescued the hematopoietic expansions of *Sucnr1*-KO HSPC both added ex vivo and in vivo.

A remaining open question was whether S100a9 pathway could be both repressed by *Sucnr1* signaling and activated by increased intracellular succinate. Using available material from cell lines, we show that the combination of cis-epoxy succinate but not monomethyl succinate and AraC reduces the expression of S100A9, versus AraC alone (new supplemental Figure 11C). Thus, S100a9 seems to be an alternative pathway controlled by *Sucnr1*, independent of intracellular succinate, which counterbalances the stimulatory effect of intracellular succinate.

The single-cell RNA-sequencing (scRNA-seq) dataset from PDX analyzed before and after 8 days of a 5-day cytarabine (AraC) treatment (GSE178912) further supported for a role of SUCNR1 in metabolism of human AML blasts. As previously stated, SUCNR1 was expressed at low levels in cluster 3, which decreased in numbers after AraC treatment and showed low expression of S100A8 and S100A9 (Figure 7D–7E). Cluster 3 was enriched for mitochondrial and metabolic processes (supplemental Figure 10B), finding consistent with previous studies showing that AraC treatment induces a metabolic rewiring characterized by enhanced mitochondrial activity, a key driver of chemoresistance³. These findings will require future in-depth exploration, but this is out of the scope of our present manuscript.

Reviewer #3 (Remarks to the Author): expert in S100A8/S100A9, AML

The manuscript presented by Cuminetti et al. offers a compelling study, demonstrating for the first time a direct influence of succinate and its receptor, SUCNR1, on cellular hematopoiesis, with implications for hematological malignancies. While prior research suggested that succinate/SUCNR1 might impact hematopoiesis indirectly, no studies to date have documented a direct effect, lending this study considerable novelty. Overall, the publication is well-structured and clearly written. However, for enhanced clarity to a broader audience, adding some contextual information in the results section could be beneficial. In certain areas, the reasoning behind specific experiments and the implications of the findings remain somewhat unclear (see comments below). Briefly outlining the observations could enrich the reader's experience significantly.

This study is extensive and detailed, and its results, along with the newly developed mouse model, hold promise for future mechanistic studies aimed at identifying targetable factors for potential therapeutic applications.

That said, I have several specific comments and questions that may enhance the already high quality of the manuscript:

- Figure 1A: Could you specify the type of survival displayed? It appears to be overall survival, but progression-free survival would also be informative.

As stated in the Methods, we reported overall survival, but we have now additionally confirmed that low levels of expression of SUCNR1 are associated with reduced progression-free survival in the same dataset (new Figure 1B). Following the recommendation of Reviewer #1, we performed subsequent Cox regression analysis, which uncovered SUCNR1 expression as an independent predictor of overall survival and progression-free survival after adjusting for age, gender and FAB classification.

- Figure 1D:

- o What was the rationale for treating mice with succinic acid? Thus far, the indication for succinic acid involvement derives only from SUCNR1 expression differences in the patient cohort.

As stated in the Introduction, "Succinate is a metabolic signal that links hypoxia to IL-1 β and cell responses in myeloid cells through intracellular mechanisms, but also autocrine/paracrine processes through *Sucnr1*. *Sucnr1* seems to have different functions depending on the context, and has been proposed as inflammatory in dendritic cells but anti-inflammatory in neural stem cells and both inflammatory or anti-inflammatory in macrophages. Cancer cells release succinate and locally polarize macrophages into tumor-associated macrophages through *Sucnr1* activation. Despite these intriguing precedents,

succinate has been scarcely explored in hematopoiesis. Human CD34+ progenitors express SUCNR1 and its activation has been suggested to induce proliferation *in vitro*. Here we investigated the *in vivo* role and the cellular and molecular mechanisms of succinate and Sucnr1 signaling in normal and malignant hematopoiesis.”

We first uncovered that low levels of expression of *SUCNR1* were associated with reduced survival in AML patients. To understand how low expression of *SUCNR1* is functionally linked to poor survival in AML patients *in vivo*, we used succinic acid injections in AML patient xenografts, as succinic acid takes the form of the anion succinate in living organisms. This clarification has been included in the manuscript (Page 5).

o How was succinic acid administered to the animals? Does the chosen route allow succinic acid to reach the cells directly, or might this be an indirect effect? Drawing conclusions about cellular metabolism from systemic metabolic changes can be challenging. Are elevated succinic acid levels detectable in the serum or other compartments in the mouse model?

As stated in the Methods, “Effect of acute succinic acid administration in human AML xenografts was studied through short-term daily *i.p.* injections of vehicle (saline) or succinic acid at 2 mg per kg neutralized with NaOH to pH 7.3, for 10 days.” Unfortunately, we did not have biological material available to answer this question for this experiment. Please, see similar comment below referred to Figure 1G, where we have been able to answer this question experimentally.

o Considering the heterogeneity of AML, it is unfortunate that only one patient sample was used. Could this patient fall within either the SUCNR1 high or low expression groups?

The patient was classified as M1, with adverse risk, and we found no detectable expression of *SUCNR1* by ddPCR. The data on *SUCNR1* expression has been included in new supplemental Figure 1C.

•Figure 1G:

o The authors state that succinic acid injection worsens the disease. However, the administration route remains unclear (see above), and an explanation for this aggravation is lacking. Are elevated succinic acid levels observable in serum, other body fluids, organs, or cells?

As stated in the Methods, “Effect of chronic succinic acid administration in leukemia models was evaluated by long-term *i.p.* injections of succinic acid.”

Succinic acid is cleared from plasma and distributed to tissues quickly after administration through various routes⁴, so we measured succinic acid levels in BM extracellular fluid and found that they were increased in succinic acid-treated versus vehicle-treated pre-leukemic NRAS-G12D⁺ mice. This result links succinate with the aggravation of disease in a direct way. The data have been included in new supplemental Figure 2B.

o While significant, the increase in myeloid frequency is marginal. Could the authors comment on whether such a small difference holds biological significance? It appears the range in myeloid frequency in the vehicle group is wider than the mean difference between the vehicle and succinic acid groups.

We agree that the increase in the already expanded myeloid cell frequency is marginal if considered as an isolated event. However, this happens coupled to further hampering of the B cell lineage and expansion of the HSPC compartment shown in the same figure. Together, it is possible to conclude aggravation of various hallmarks of NRAS-G12D-driven disease

and hence biological significance. We have reformulated these Results so that they are considered together, removing any potential emphasis on the effect on the myeloid bias alone.

•Figure 1J: It would be insightful to see corresponding protein levels.

We agree but this is challenging, given the lack of commercially available mouse *Sucnr1*-specific antibody stated in the manuscript and the limited availability of *Sucnr1*-tomato mice. However, our data in Figure 1 show that *SUCNR1* mRNA expression is biologically relevant for disease progression in AML patients, as shown by the association of low *SUCNR1* mRNA expression with reduced overall survival and progression-free survival (Figure 1A-1B).

•The concluding statement in the first Results section could benefit from added context. The link between low *SUCNR1* expression and succinate as a contributor to leukemia progression is not fully clear.

We have reformulated this statement as recommended, it reads as follows: “To sum up, AML patients show heterogeneous expression of *SUCNR1* and low *SUCNR1* represents a prognostic marker for reduced overall and progression-free survival. Conversely, in AML mouse models and patient xenografts with low *SUCNR1* expression, succinate accumulates in BM and contributes to leukemia progression”.

•Figure 2A: Could you specify the exact mouse ages corresponding to “young adult” and “later middle-aged stage”?

This information was included in the Legends of the corresponding Figures in the previous version of the manuscript. Briefly, supplemental Figure 3 “young adult” – (A-E) 10-17 weeks of age and figure 2 “middle-aged” – (A, peripheral blood) 38-41 weeks old, (B-H, bone marrow) 53-54 weeks old.

•Supplemental Figure 4B: The claimed increase in live cell count is minimal and not statistically significant. Displaying actual frequencies instead of percentage changes from WT may be more illustrative. Also, the event count in the representative plots appears low (below 10,000 total events), which is below the standard for reliable flow cytometry.

We have reformulated the Results section, which now reads: “analysis of the LSK cell compartment in the BM of *Sucnr1*-KO mice showed reduced numbers of early apoptotic BM LSK and a trend towards increased numbers of live cells”.

Both frequency and percentage of change are illustrative and accepted as valid ways to display results. In this case, we chose to express the results as percentage of change because it reduces variability in the analysis and helps place the focus in the actual changes caused by *Sucnr1* deletion, which we agree are mild as shown by the data.

Regarding event count, in these experiments around 1.5×10^6 total events from total bone marrow per sample were recorded in the flow cytometer, and from those, 1000 to 2000 gated-LSK were used to determine the proportion of live and apoptotic LSK cells per sample. This experimental setup is standard in studies involving subsets of HSPC by flow cytometry. Further, although lineage depletion or LSK sorting would increase the numbers of LSK cells analyzed, those protocols are known to reduce cell viability which could result in misleading observations.

•Supplemental Figure 4D: How can the knockout of *SUCNR1* impact succinate levels in BMEF? As I understand it, *SUCNR1* is activated by succinate but does not regulate

succinate availability or metabolism. I would anticipate succinate accumulation due to receptor absence.

As stated in the Results section of the previous version of our manuscript, succinate levels were reduced in the BM extracellular fluid (BMEF) of *Sucnr1*-KO versus WT mice (supplemental Figure 4E). Given that succinate is an intermediate metabolite of the TCA cycle, we studied its potentially increased cycling which would result in metabolic changes. *Sucnr1* deletion resulted in increased baseline oxygen consumption rate in lineage-negative progenitors (supplemental Figure 4G). This effect was specific and reversible because the maximal oxygen consumption rate and the glycolytic flux remained unchanged (supplemental Figure 4G). Consistent with these data, both total and mitochondrial reactive oxygen species (ROS) levels were higher in LSK cells from *Sucnr1*-KO mice (supplemental Figure 4H). We have now established a new collaboration that has enabled us to provide direct evidence using metabolomics in HSC, which confirms increased intracellular succinate levels in *Sucnr1*-KO versus WT mice (please see new supplemental Figure 4F). The regulation of succinate localization by *Sucnr1* should be subject of future work. Whereas it could simply be a matter of increased succinate availability in the absence of *Sucnr1* binding, in otherwise steady-state/healthy conditions, active regulatory mechanisms could additionally take place. We have added these ideas to the Discussion.

•Supplemental Figure 4E: Background controls for OCR and ECAR measurements, such as rotenone/antimycin A for OCR and 2-DG for ECAR, are absent. Without these, it is challenging to isolate mitochondrial respiration from other oxidative processes. Could such controls be included?

We agree, but it is not possible to repeat these experiments to include these controls due to mice availability. The main result of these experiments is that lineage-negative progenitors from *Sucnr1*-KO mice have more oxygen consumption in basal conditions and this is concomitant with increased ROS levels in LSK cells. Hence, we have toned down the specific participation of mitochondrial respiration in these processes.

•Figure 3E: How do you explain the reduced IL-1 β protein levels in BMEF when IL-1 β gene expression is significantly increased in *Sucnr1*-KO versus WT animals? Consequently, the concluding statement of this section may require modification or expansion.

Given the increased mRNA expression of *Il1b* in HSPC from *Sucnr1*-KO versus WT mice, we analyzed the intracellular levels of IL-1 β protein and confirmed them slightly but significantly increased in *Sucnr1*-KO compared to WT mice (Figure 4E). This effect seemed to be specific of the HSPC compartment, as we observed no differences in the intracellular levels of IL-1 β in total live cells, lineage-positive cells, lineage-negative progenitors or LK progenitors (supplemental Figure 7C). This is indeed in contrast with the levels of IL-1 β protein in BMEF, which we found to be reduced in *Sucnr1*-KO versus WT mice. Thus, *in vivo* deletion of *Sucnr1* promotes inflammation specifically in HSPC, as explained in the concluding statement of this section. We have also added that there are complex changes taking place in the global cytokine status of the BM in *Sucnr1*-KO mice. Discrepancies may be caused by multiple factors, and due to the complexity of the question, we believe that it falls outside the scope of this work and should be subject of future work. However, despite this complexity, it is important to note that we were able to uncover *S100a8* and *S100a9* as consistently increased in their mRNA expression in HSPC and in their protein levels in BMEF, and demonstrated their primary functional/mechanistic role in *Sucnr1*-KO mice. We have included these ideas in the Discussion section.

•Supplemental Figure 6E: The intent of this raw spectral ribbon plot is unclear. Without spectral unmixing, assigning signals to specific fluorochromes or fluorescent proteins is not possible.

We agree and this panel has been removed.

•In the discussion, the authors note “overexpression of S100a8 and S100a9 in HSPC subsets [...]” How does this align with observations of increased S100A8/S100A9 in AML blasts in bone marrow compared to peripheral blood and the identified pro-inflammatory triggers reported in Blood Advances 2022 6(21)?

Thank you for this comment. Our work aligns but deepens the understanding provided by previous reports showing that S100A8/A9+ AML cells display a distinct metabolic, differentiation, and chemoresistance profile, compared to S100A8/A9- AML cells, which are therapy-sensitive and found mainly in periphery. Using cell lines and in vitro cultures, the authors of this work identified stromal cell-derived IL-6 induced expression of S100A8/A9 in AML cells in a JAK/STAT3-dependent manner. We are now providing new measurements showing similar levels of IL-6 in the BMEF of WT and Sucnr1-KO mice. Thus, our data in vivo supports Sucnr1 as an alternative pathway controlling S100A8/A9 independently of IL-6. Boosting Sucnr1 and blocking S100A9 thus may open new potential therapeutic avenues in the treatment of myeloid malignancies. We have included the new results in Figure 4F and these ideas in the Discussion.

Reviewer #4 (Remarks to the Author): expert in succinate AML, hematopoiesis, mouse models

The manuscript by Cuminetti et al. examines the role of Succinate receptor (Suncr1) in hematopoiesis. The authors show that low Suncr1 expression correlates with poor survival in AML. Further, treatment of human and leukemia cells in vivo with succinic acid promotes leukemia progression. In the context of normal hematopoiesis sucnr1 knockout resulted in the expansion of several stem and progenitor populations determined by cell number in aged mice. The increased in HSPC cell numbers correlated with an increase in gene expression in signatures associated with HSCs and a decreased gene expression in signatures associated with decreased differentiation and inflammation. Expression of s100a8 and s100a9 were increased in sucnr1 knockdown mice. Treatment with s100a8/9 inhibitor results in reduced hematopoietic cells in the sucnr1 knockdown mice. Finally, the authors show that sucnr1 low expressing cells have high S100a8/9 in single-cell RNA-sequencing post cytarabine treatment (derived from PDX models). Overall, the manuscript highlights several intriguing findings. However, several additional experiments/analyses should be performed to address the concerns below.

Major Concerns:

1. In Figure 2, the conclusions about changes in HSPC and mature cells are made based on cell numbers only. Most of the populations have increased numbers including total WBCs. Are there any changes in population percentages which may help to tease about if any one population is expanded upon sucnr1 knockout?

We focused this figure on absolute numbers as these are the accurate way to understand real expansions/reductions in cell subsets. Relative frequencies are not measurements of real cell numbers and in fact an increased proportion of a cell subset can be associated to reduced absolute numbers. However, we do agree with the Referee on the value of understanding the changes in cell population percentages in the global pathophysiological

context, as they provide valuable information of which cell subset/s may be most affected by the intervention compared to others.

As requested by the Reviewer, we are now providing the population percentages of LSK cells, which was increased in the BM of *Sucnr1*-KO versus WT mice (new supplemental Figure 4B). Analysis of HSPC subsets corresponding to HSC and MPP1-MPP6 revealed increased relative frequencies of HSC, MPP1, MPP4, MPP5 and MPP6 (supplemental Figure 4B). These data further support that the stimulated hematopoiesis in the BM of *Sucnr1*-KO mice originates in the HSPC compartment. We have included these ideas in the Results and Discussion sections.

2. The authors use the steady-state populations and gene expression analysis to conclude changes in HSPC function. These conclusions should be supported by performing competitive bone marrow transplants.

We agree with the Reviewers, and we are now providing new data on the functionality of HSC in vivo analyzed by competitive repopulation assays of BM nucleated cells isolated from WT or SUCNR1-KO mice (Ly5.2) into lethally irradiated Ly5.1 congenic recipients. Peripheral blood analysis from recipient mice 16 weeks after transplantation revealed a mild reduction in engraftment in recipients transplanted with SUCNR1-KO cells compared to recipients transplanted with WT control cells (please see new Figure 3G).

Although *Sucnr1*-KO BM cells seem to be less fit in competition with WT cells, primary transplants of *Sucnr1^{fl/fl}* or *Mx1-Cre Sucnr1^{fl/fl}* BM cells into C57BL/6J WT mice, induced after stable engraftment, greatly recapitulate the hematopoietic expansions of *Sucnr1*-KO mice (Figure 2I-2L).

Thus, to further substantiate the role of *Sucnr1* in HSPC function in vivo, we performed competitive repopulation assays using 200 *Sucnr1*-tdTomato⁺ or *Sucnr1*-tdTomato⁻ FACS-sorted HSPC (containing a mix of 40 HSC, 86 MPP3 and 74 MPP4) isolated from Ly5.2 *Sucnr1*-tdTomato mice into lethally irradiated Ly5.1 congenic recipients. Peripheral blood analysis from recipient mice 10 weeks after the transplant showed a reduction of circulating leukocytes and donor-derived monocytes in recipients of *Sucnr1*⁺ cells compared to recipients of *Sucnr1*⁻ cells (Figure 3H). This experiment allowed us to conclude that *Sucnr1* expression marks a subset of HSPC with restricted engraftment potential in vivo.

3. Figure 5 would benefit from including wild-type mice. This would allow the assessment of tasquinimod on hematopoiesis in the normal setting and enable the researchers to determine if treatment restores cell numbers back to the wild type levels.

As suggested by Reviewers #1, #2 and #5, and #4, to substantiate a selective role of S100a8/S100a9 in the hematopoietic system via *Sucnr1* deletion, we performed an independent experiment treating both *Sucnr1*-KO and WT mice with tasquinimod or vehicle in parallel (supplemental Figure 8A). Confirming our previous results, tasquinimod treatment in *Sucnr1*-KO mice reduced the expanded numbers of circulating leukocytes to the numbers observed in WT mice, including both overall myeloid cells as well as B cells and with no significant effect observed in WT mice (supplemental Figure 8B). Tasquinimod administration reduced the expanded numbers of monocytes in BM of *Sucnr1*-KO mice to the numbers found in WT mice, whilst no effect was observed in WT mice (supplemental Figure 8C). Conversely, treatment with tasquinimod reduced the numbers of BM B cells to similar levels in both *Sucnr1*-KO and WT mice (supplemental Figure 8C). The expansion of BM HSC in *Sucnr1*-KO mice was reduced by treatment with tasquinimod, but a minor reduction in HSC numbers was also observed in WT mice, bringing HSC values in both tasquinimod-treated groups to similar level (supplemental Figure 8D). In turn, the increased colony-forming

potential in *Sucnr1*-KO mice was reduced to WT values, with no effect seen in samples obtained from WT mice (supplemental Figure 8E). Of note, this assay is optimal for the growth of primitive erythroid progenitor cells, granulocyte-macrophage progenitor cells, and multi-potential granulocyte, erythroid, macrophage, megakaryocyte progenitor cells, but not for the growth of lymphoid progenitor cells. Thus, these data suggest that tasquinimod effect originates in the HSC compartment for both lineages, and whereas the effect of tasquinimod on the myeloid lineage is fully dependent on *Sucnr1* deletion, the effect on the B cell lineage is only partially dependent on the loss of *Sucnr1* signaling.

4. Do monomethyl succinate and succinic acid have the same degree of effect on the intracellular biology of succinate? For example, do they increase oxidative phosphorylation to the same level. Differences between the compounds could be a confounding factor in the interpretation of results in figure 4.

Thanks for this very good question, challenging to answer in the light of the various variables to consider. Monomethyl succinate is a dicarboxylic acid monoester that is succinic acid in which one of the carboxy groups has been converted to its methyl ester (succinic acid monomethyl ester). Addition of ester moieties to molecules serves as an efficient means to help internalizing compounds across the plasma membrane. In living organisms, succinic acid takes the form of the anion succinate, which would be the naturally occurring endogenous form, and requires transporters for its internalization. Thus, monomethyl succinate is cell-permeable and it allows bypassing the permeability limitation of succinate. However, monomethyl succinate is considered a prodrug and it requires hydrolysis by esterases inside the cell that convert it to succinate, making it available for intracellular metabolic and signaling functions. In parallel, *cis*-epoxy succinate is 10-20x more potent than succinic acid on *Sucnr1*, and it did not change SDH activity in previous work at concentrations up to 500 μM ⁵. Importantly, in our work, we are using concentrations up to 10mM, and uncover a similar CFU-C output for both monomethyl succinate and *cis*-epoxy succinate in *Sucnr1*-KO HSPC (whilst their effect is opposite in WT cells in presence of *Sucnr1*). Thus, we can conclude that the same concentration of monomethyl succinate and *cis*-epoxy succinate induce the same functional output in the absence of *Sucnr1*, but we cannot say that the underlying mechanisms are exactly the same. However, we think that these results add clarity and rule out potential differences between compounds as a confounding factor for interpretation, and hope the Referee will agree with us. Given the multi-layer intracellular functions of succinate, the question raised by the Reviewer is out of the scope of the present manuscript, but we do agree that it should be subject of future work. We have added this idea in the Discussion section.

5. Using a genetic approach like overexpressing *sucnr1* would help to solidify the conclusions from figure 4.

Monomethyl succinate is inactive on *Sucnr1* and enhances the colony-forming potential in total BM (TBM), in a dose-response manner. Conversely, succinic acid, which displays both *Sucnr1*-dependent and independent effects, results in unchanged colony-forming potential. Taken together, we reasoned that these results suggest that *Sucnr1* activation represses the intracellular effect of succinate. We have reformulated these results for clarity. Further, we have followed the suggestion of Referee #1 and explored the effect of succinic acid and monomethyl succinate combination, which results in a partial increase in colony-forming potential in TBM versus succinic acid alone and a partial reduction versus monomethyl succinate alone, providing robust evidence that *Sucnr1* activation represses the intracellular effect of succinate. Please, see the results in new Figure 3D. In parallel, the previous version of our work already showed a similar CFU-C output for both monomethyl succinate and *cis*-

epoxy succinate in *Sucnr1*-KO HSPC, whilst their effect is opposite in WT cells (in presence of *Sucnr1*). We agree with the Reviewer that using a complementary strategy to the *in vivo* deletion of *Sucnr1* would add valuable information, but we think the conclusions from this figure are solid, particularly after the addition of the new experiments mentioned above. We hope the Referee will agree with us.

6. Further, overexpressing *sucnr1* and measuring the consequence on leukemia progression would enhance the study.

To answer this question, we have used the human AML cell line Kasumi. Overexpression of *SUCNR1* with a lentivector and treatment with cis-epoxy succinate increased apoptosis in comparison with the apoptotic rate of Kasumi cells transduced with an empty vector and treated with cis-epoxy succinate. Please, see the results in new supplemental Figure 11C-11D.

7. Figure 6 suggests that expression of *sucnr1* may influence the response to chemotherapy. It would strengthen the results by functionally testing this.

Using NOMO-1 and Kasumi human AML cell lines, we observed that 10mM treatment with cis-epoxy succinate improved chemotherapy induced apoptosis versus chemotherapy alone (supplemental Figure 11A-11B). The results are particularly remarkable for Kasumi (M2) versus NOMO-1 (M5).

Minor:

1. In Figure 1D what are the patient characteristics? Specifically it would be helpful to include FAB and mutation status as the authors show these are related to *sucnr1* expression.

The patient was classified as M1, with adverse risk, and we found no detectable expression of *SUCNR1* by ddPCR. The data on *SUCR1* expression has been included in supplemental Figure 1C.

Reviewer #5 (Remarks to the Author): ECR, co-reviewed with Rev#2

References

- 1 Farge, T. *et al.* Chemotherapy-Resistant Human Acute Myeloid Leukemia Cells Are Not Enriched for Leukemic Stem Cells but Require Oxidative Metabolism. *Cancer Discov* **7**, 716-735 (2017). <https://doi.org/10.1158/2159-8290.Cd-16-0441>
- 2 Duy, C. *et al.* Chemotherapy Induces Senescence-Like Resilient Cells Capable of Initiating AML Recurrence. *Cancer Discov* **11**, 1542-1561 (2021). <https://doi.org/10.1158/2159-8290.Cd-20-1375>
- 3 Aroua, N. *et al.* Extracellular ATP and CD39 Activate cAMP-Mediated Mitochondrial Stress Response to Promote Cytarabine Resistance in Acute Myeloid Leukemia. *Cancer Discovery* **10**, 1544-1565 (2020). <https://doi.org/10.1158/2159-8290.Cd-19-1008>
- 4 Jung, Y., Song, J. S. & Ahn, S. Pharmacokinetics and Tissue Distribution of (13)C-Labeled Succinic Acid in Mice. *Nutrients* **14** (2022). <https://doi.org/10.3390/nu14224757>

- 5 Geubelle, P. *et al.* Identification and pharmacological characterization of succinate receptor agonists. *Br J Pharmacol* **174**, 796-808 (2017). <https://doi.org/10.1111/bph.13738>

REVIEWER COMMENTS

Reviewer #1 (Remarks to the Author): expert in Succinate and Sucnr1

The authors have addressed the reviewer's previous comments in a general and satisfactory manner, significantly improving the clarity and rigor of the manuscript. The experimental design, data presentation, and interpretation have been appropriately revised, and the manuscript is now substantially strengthened.

However, a few minor points remain that should be clarified or corrected to avoid potential confusion:

- It is not clear whether the efficiency of Sucnr1 deletion was verified in the Mx1-Cre model, both in the bone marrow of the primary mice and in the transplanted recipients reconstituted with these BM cells. It is recommended to confirm the deletion by assessing Sucnr1 expression levels (at the mRNA level) to ensure that the observed phenotypes are indeed due to effective loss of Sucnr1 in hematopoietic cells.

The Mx1-Cre mouse model is broadly used in the hematopoietic field to induce recombination in a variety of genes in the hematopoietic system during adulthood, allowing fine-tuned controlled levels of Cre selectively in blood cells when used under transplantation.

One of the main drawbacks of Mx1-Cre is that the model is inducible with an inflammatory signal that may impact HSCs. However, bone marrow isolated from mice treated three times with similar doses of plpC (followed by 10 days recovery) was serially transplanted into irradiated recipients, showing no differences in the repopulation activity (Essers et al, 2009). Thus, transient activation of IFN α signaling does not affect the number of functional HSC, as opposed to chronic activation. The mice in this experiment were monitored for a long period of time (47 weeks after plpC induction) and the controls used were Sucnr1fl/fl with no Cre treated with plpC, to reduce any potential confounding effect derived from plpC-induced inflammation.

In addition, please note that all our mice are kept in SOPF status, to prevent spontaneous Cre activation before plpC induction.

To answer to the Reviewer question, we have now assessed Sucnr1 expression in frozen total BM nucleated cell samples from primary mice and from the transplanted mouse recipients shown in Figure 2I. As expected, BM nucleated cells from Sucnr1fl/fl and Mx1-Cre Sucnr1fl/fl primary mutant mice displayed similar expression of Sucnr1 prior to plpC induction. In turn, in vivo deletion of Sucnr1 from hematopoietic cells was confirmed by virtual absence of Sucnr1 expression in WT recipients of Mx1-Cre Sucnr1fl/fl experimental BM cells versus recipients of control Sucnr1fl/fl BM cells, injected with plpC 4 weeks after the transplant and analyzed 47 weeks after plpC induction. Of note, Sucnr1 expression in BM cells from WT recipients of control Sucnr1fl/fl BM cells injected with plpC 4 weeks after the transplant and analyzed 47 weeks after plpC induction showed no significant differences compared to Sucnr1fl/fl or Mx1-Cre Sucnr1fl/fl primary mutant mice. Please, see the results in new Supplementary Figure 4L.

- In the merged manuscript (lines 243–246), the authors state that the majority of Sucnr1⁺ HSPCs were identified as MPP3, followed by MPP4, with an average of 0.191% of Sucnr1⁺ cells (Figure 3B and Supplemental Figure 6C). Could the authors clarify that this order is correct, and not MPP2? In the figure, it appears that MPP2 might have a higher fraction than MPP4, so confirmation would help avoid confusion.

Our statement reads: “Our analysis revealed that all HSPC subsets show a varying fraction of Sucnr1+ cells and the majority of Sucnr1+ HSPC were identified as MPP3, with an average of 0.191% of Sucnr1+ cells, followed by MPP4 (Figure 3B and supplemental Figure 6C).” We confirm that this statement is correct.

The bar chart of Figure 3B (upper panel) shows the fraction of Sucnr1+ cells within each HSPC subset, as we thought it was important to provide these quantifications per cell subset. From the Source Data file:

% of Sucnr1+ cells							
LSK	HSC	MPP1	MPP2	MPP3	MPP4	MPP5	MPP6
0.120	0.000	0.100	0.090	0.200	0.060	0.090	0.070
0.080	0.000	0.000	0.000	0.110	0.030	0.000	0.150
0.060	0.000	0.000	0.000	0.130	0.000	0.000	0.000
0.150	0.200	0.130	0.000	0.210	0.060	0.000	0.130
0.120	0.000	0.000	0.080	0.170	0.000	0.000	0.000
0.039	0.000	0.000	0.000	0.195	0.071	0.000	0.000
0.190	0.000	0.000	0.000	0.429	0.118	0.000	0.000
0.033	0.000	0.000	0.000	0.198	0.000	0.000	0.000
0.059	0.000	0.000	0.000	0.272	0.069	0.000	0.000
0.027	0.568	0.000	0.568	0.000	0.000	0.000	0.127

As you can see, all HSPC subsets showed a varying fraction of Sucnr1+ cells and the average fraction of Sucnr1+ cells in MPP3 was 0.191%. Sucnr1 was expressed consistently in the MPP3 subset in 9 out of 10 animals.

Figure 3B (upper panel) shows that only 3 out of 10 animals displayed detectable % of Sucnr1+ cells within MPP2, versus 6 out of 10 within MPP4. The fraction of Sucnr1+ cells within MPP4 showed also less standard deviation across animal samples in the group compared to MPP2, although it was high in both cases. Because of these reasons, we did not provide the average fraction in the text, but the raw values are shown in the figure and provided in the Source Data file.

The pie chart of Figure 3B (lower panel) represents the distribution of Sucnr1+ cells in the total HSPC compartment. Here, you can see how the majority of Sucnr1+ HSPC were identified as MPP3, followed by MPP4. From the Source Data file:

Pie chart % distribution of Sucnr1+ events						
HSC	MPP1	MPP2	MPP3	MPP4	MPP5	MPP6
6.333	1.051	5.214	71.516	10.783	0.577	4.526

To calculate the % shown above and represented in the pie chart, we considered each fraction of Sucnr1+ HSPC subset in absolute numbers, summed all of them, and calculated the distribution of each subset in the total Sucnr1+ HSPC population, in %. Thus, the numbers in the pie chart depend on the % of Sucnr1+ cells and the absolute numbers of each HSPC subset. MPP2 is a rare population in the bone marrow, found in lower numbers than MPP4 (0.00955% versus 0.0636% of total bone marrow cells, respectively; in this experiment, 6.7 times lower). Hence, when considering all HSPC subsets together to

understand the distribution of *Sucnr1*⁺ cells, the majority of *Sucnr1*⁺ HSPC are identified as MPP3, followed by MPP4, HSC, MPP2 and MPP6, in that order.

- The authors mention that in the first and second replating, both monomethyl succinate and cis-epoxy succinate stimulated the colony-forming potential of *Sucnr1*-KO HSCs, suggesting a loss of SUCNR1-mediated repression of this potential (Figure 3F). The Discussion ends with the sentence “Given the multi-layer intracellular functions of succinate, the question should be subject of future work” (lines 533–534), which could be refined for greater clarity and precision.

The authors’ results suggest that, in the absence of SUCNR1, cis-epoxy succinate may enhance colony-forming potential through receptor-independent mechanisms, possibly related to intracellular metabolic modulation. Have the authors assessed intracellular succinate levels to support this interpretation? Such data would help to substantiate the proposed mechanism and clarify whether the effects of the two compounds truly involve similar intracellular pathways.

These data were present in the previous version of the manuscript, and Reviewer 4 formulated a similar comment in the previous round of revision (point 4: “Do monomethyl succinate and succinic acid have the same degree of effect on the intracellular biology of succinate? For example, do they increase oxidative phosphorylation to the same level. Differences between the compounds could be a confounding factor in the interpretation of results in figure 4”).

Monomethyl succinate is a dicarboxylic acid monoester that is succinic acid in which one of the carboxy groups has been converted to its methyl ester (succinic acid monomethyl ester). Addition of ester moieties to molecules serves as an efficient means to help internalizing compounds across the plasma membrane. In living organisms, succinic acid takes the form of the anion succinate, which would be the naturally occurring endogenous form, and requires transporters for its internalization. Thus, monomethyl succinate is cell-permeable and it allows bypassing the permeability limitation of succinate. However, monomethyl succinate is considered a prodrug and it requires hydrolysis by esterases inside the cell that convert it to succinate, making it available for intracellular metabolic and signaling functions. In parallel, cis-epoxy succinate is 10-20x more potent than succinic acid on *Sucnr1*, and it did not change SDH activity in previous work at concentrations up to 500 μ M (Geubelle et al., 2017). Importantly, in our work, we are using concentrations up to 10mM, and uncover a similar CFU-C output for both monomethyl succinate and cis-epoxy succinate in *Sucnr1*-KO HSPC whilst their effect is opposite in WT cells in presence of *Sucnr1*. Thus, we can conclude that the same high concentration of monomethyl succinate and cis-epoxy succinate induce the same functional output in the absence of *Sucnr1*, but we cannot say that the underlying mechanisms are exactly the same. We think that these results rule out potential differences between compounds as a confounding factor for interpretation of the functional role of *Sucnr1*, and hope the Referee will agree with us.

Further, please note the experimental conditions of the technique used in these assays, where succinate analogues are added only at the time of plating. From the Methods: “Serial colony-forming-unit assays from 300 FACS-sorted HSC from magnetically enriched bone marrow Lin⁻ progenitors were performed as described previously, colonies were scored and 10 000 cells replated after 7, 12 and 17 days, 10mM of monomethyl succinate or cis-epoxy succinate were added at each plating”. Measuring succinate in colonies may not be reflective of the initial effect of succinate analogues on HSC, but rather on the differential cell subsets present at the end of the culture procedure. The increase in intracellular succinate after a single addition of succinate analogues in vitro could also be transient and/or have

differential dynamics for different analogues because of the structural differences mentioned above.

Together, we think that this would be a time-consuming experiment that would require a considerable use of animals and other resources, whilst producing an output that would have significant caveats and will not effectively convince whether the effects of the two compounds truly involve similar intracellular pathways.

Besides, although certainly interesting, dissecting the potential intracellular differences between different succinate analogues is not the purpose of the present work and would not influence our main message, which robustly and convincingly shows that activation of *Sucnr1* counterbalances the stimulatory effect of succinate in HSPC. We hope that the Referee will agree with us.

However, as suggested by the Reviewer, we have reformulated the Discussion, which now reads: "In turn, similar concentrations of monomethyl succinate and cis-epoxy succinate induced the same increased functional output in HSC in the absence of *Sucnr1*, but it is unknown whether the underlying mechanisms are exactly the same for both compounds. Thus, in the absence of *Sucnr1*, cis-epoxy succinate enhances colony-forming potential through receptor-independent mechanisms, possibly with contribution to intracellular metabolic modulation. However, given the structural differences among different succinate analogues and the multi-layer intracellular functions of succinate, this question should be subject of future work."

- Please ensure that all figures include legends indicating the meaning of the colors used. In several cases (e.g., Figures 1F, 1K, 2I, and others), the color legends are missing. We recommend revising all figures to maintain consistency and clarity across the manuscript.

We have gone through all the figures, and all of them showed the colors used in all panels in the previous version of our manuscript, either in the corresponding panel or in its vicinity when the panel is part of a subgroup of panels that belong to a similar experimental design. This way, there is just one legend for Figure 1F, 1G and 1H (under 1G); Figure 1I, 1J and 1K (under 1I); 2A-2H (right of 1A); 2I-2L (colored mice in 2I), etc. This saves space in the figure while maintaining clarity.

Reviewer #2 (Remarks to the Author): expert in RNA-seq and scRNA-seq analysis, AML and hematopoiesis

Thanks for convincingly addressing my concerns.

We would like to thank the Reviewer again for the important points raised that have helped us strengthen our work.

Reviewer #3 (Remarks to the Author): expert in S100A8/S100A9, AML

The authors have thoroughly addressed my initial concerns, questions, and comments, which has further improved the overall already high quality of the manuscript. While most points were addressed experimentally, some were also resolved through well-reasoned justification and discussion. Well done, and thank you for this very nice piece of science.

Thank you for your kind feed-back. We would like to take the opportunity to thank you once more for the important input that we have received from you.

Reviewer #4 (Remarks to the Author): expert in succinate AML, hematopoiesis, mouse models

The authors have gone to great lengths to address concerns from five reviewers and significantly strengthened their manuscript. I have no additional concerns.

We would like to thank the Reviewer again for the constructive and important input that has helped us strengthen our manuscript.

REVIEWER COMMENTS

Reviewer #1 (Remarks to the Author): expert in Succinate and Sucnr1

Thank you for addressing the remaining minor points. I have no further comments.

We would like to thank the Reviewer again for the points raised that helped us strengthen our work.